# LET OOD FEATURE EXPLORE VAST PREDEFINED CLASSIFIERS

**Kewen Xia[1], Xiaodong Yue[1,2,3] \*, Zhipeng Wei[1], Yaxin Peng[2], Zihao Li[1], Jianxiang Zhu[2], Jie Shi[1] & Peilin Xu[1]**

[1]School of Computer Engineering and Science, Shanghai University, China
[2]School of Future Technology, Shanghai University, China
[3]Institute of Artificial Intelligence, Shanghai University, China
`kewen_xia@outlook.com, yswantfly@shu.edu.cn`

## ABSTRACT

Real-world out-of-distribution (OOD) data exhibit broad, continually evolving distributions, rendering reliance solely on in-distribution (ID) data insufficient for robust detection. Consequently, methods leveraging auxiliary Outlier Exposure (OE) data have emerged, substantially enhancing generalization by jointly fine-tuning models on ID and large-scale OE data. However, many existing approaches primarily enforce orthogonality between ID and OE features while pushing OE predictions toward near-uniform, low-confidence scores, thus overlooking the controllability of representation geometry. We propose Vast Predefined Classifiers (VPC), which constructs a pre-specified Orthogonal Equiangular Feature Space (OEFS) to explicitly separate ID and OOD representations while capturing the rich variability of OOD features. We employ evidential priors to align ID features with their class-specific Equiangular Basic Vectors (EBVs), thereby preserving ID performance. In parallel, a new VEBV loss encourages OE features to explore the subspace spanned by Vast EBVs (VEBVs), enabling a rich characterization of diverse OOD patterns. This dual optimization, coupled with the prescribed geometric representation space, promotes optimal orthogonality between ID and OOD representations. Furthermore, we introduce the VPC Score, a discriminative metric based on the L2 activation intensity of features over the predefined classifiers. Extensive experiments across diverse OOD settings and training paradigms on benchmarks including CIFAR-10/100 and the ImageNet-1k, demonstrate strong and robust performance, validating VPC's effectiveness. Code is available at `https://github.com/eightnight2049/VPC`.

## 1 INTRODUCTION

In open-world scenarios, deep neural networks (DNNs) must not only accurately recognize in-distribution (ID) samples encountered during training, but also robustly distinguish and detect unseen out-of-distribution (OOD) data (Hendrycks & Gimpel, 2016; Liu et al., 2020; 2021). Because real-world OOD data are often diverse and continuously evolving (Ye et al., 2025), relying solely on ID data during training is no longer sufficient to achieve desirable robustness. A recent and effective strategy is therefore to leverage large-scale auxiliary outlier-exposure (OE) data and fine-tune the model jointly on ID and OE samples, which markedly improves generalization to OOD (Hendrycks et al., 2018; Katz-Samuels et al., 2022).

However, typical outlier-exposure (OE)-based pipelines either encourage the model to produce uniform, low-confidence predictions on OE data (Hendrycks et al., 2018), or regularize post-hoc scores such as maximum softmax probability (MSP) and energy values(Hendrycks & Gimpel, 2016; Liu et al., 2020). Beyond logit-based, gradient or neighborhood-based detectors provide complementary signals (Huang et al., 2021; Sun et al., 2022), but these methods fail to explicitly construct a controllable representation geometry during training (Liang et al., 2017; Sun et al., 2021; Wang et al., 2022a; Djurisic et al., 2022). This leads to two issues: (i) unstable and incomplete separation between in-distribution (ID) features and OE features, which may bias predictions toward ID classes at test time (Wu et al., 2024); and (ii) limited representational capacity for the rich and

Figure 1: Overview of the Proposed VPC OOD Detector. During joint ID/OE fine-tuning or one-stage training, the normalized feature space is aligned to an Orthogonal Equiangular Feature Space (OEFS) with predefined EBVs/VEBVs; $\mathcal{L}_{\text{ENC}}$ aligns ID to EBVs, $\mathcal{L}_{\text{VEBV}}$ attracts OOD into the VEBV subspace, $\mathcal{L}_{\text{OC}}$ enforces separation, and OODness is scored by the L2-based VPC Score, defined as the absolute difference between the L2 activation magnitudes of the two subspaces.

evolving spectrum of OOD patterns, a problem that becomes more prominent especially when the semantic space scales up (Huang & Li, 2021).

To overcome these limitations, we propose Vast Predefined Classifiers (VPC) , a framework that explicitly represents OOD variability by allocating a large set of predefined optimal classifiers distinct from ID classifiers. Our design is inspired by the phenomenon of Neural Collapse (NC)(Papyan et al., 2020), where the class means of features and classifiers approach a Simplex Equiangular Tight Frame (ETF) (Han et al., 2021), as well as by the equiangular basis vectors (EBVs) based feature alignment optimization to induce this phenomenon (Shen et al., 2025; Markou et al., 2024). The central intuition is that, to preserve the ID decision structure while accommodating the diversity of real OOD under joint ID/OE fine-tuning, the representation space must embed a controllable geometric prior with scalable capacity. Naive uniformization of OE outputs may fail to ensure stable separability and lacks sufficient and directional coverage for complex OOD modes (Yang et al., 2025; Tang et al., 2025).

To this end, we anchor OOD discrimination on predefined EBVs as class prototypes, defining an Orthogonal Equiangular Feature Space (OEFS) on a high-dimensional unit hypersphere. In OEFS, EBVs enforce within-class collapse for ID features, while abundant VEBVs span a broad, activatable OOD subspace. Guided by evidential priors, we align ID features toward EBVs; an OOD-oriented attraction with orthogonality-based isolation encourages activation in the VEBV subspace, maintaining geometric separation from the ID subspace. This induces an interpretable pattern: ID activates EBVs, whereas OOD activates VEBVs. Building on this, the VPC Score computes L2 activation intensity over the predefined classifier space as a class-agnostic OOD measure, decoupled from any ID classifier head. Our contributions are summarized as follows:

- We propose Vast Predefined Classifiers (VPC) framework: using an Orthogonal Equiangular Feature Space (OEFS) to structurally decouple ID and OOD representations, and introducing a controllable, expandable geometric subspace for OOD without sacrificing ID performance.

- We propose an evidential prior guided Neural Collapse (ENC) loss that stabilizes ID representations, and an OOD-oriented VEBV subspace attraction loss with an orthogonality constraint regularization loss to maintain consistent ID–OOD separation.

- We introduce the VPC Score as a class-agnostic metric, achieving robust discrimination across diverse OOD modes and outperforming MSP/uncertainty scores.

- We obtain consistent performance gains on representative OOD detection settings, validating the synergistic benefits of geometric priors and evidential modeling.

## 2 RELATED WORK

**Auxiliary OE Data based OOD Detection.** With partial OOD data, Outlier Exposure (OE) improves detection by co-training on ID and external samples. Classic OE (Hendrycks et al., 2018) drives OOD outputs toward uniformity; follow-ups either design loss, energy, or metric learning objectives (Liu et al., 2020; Bai et al., 2023; Ying et al., 2017) or mine/synthesize harder outliers to sharpen the boundary (Ming et al., 2022a; Chen et al., 2021; Wang et al., 2024a; 2023; Zheng et al., 2023). A parallel line of research focuses on post-hoc models to further enhance separability, such as methods

based on feature processing (Sun et al., 2021; Liang et al., 2017; Wang et al., 2022a;b), activation shaping (Djurisic et al., 2022), density estimation (Peng et al., 2024), or energy-based scoring (Zhang et al., 2022). However, most OE methods emphasize output-level uniformization and provide little control over the geometry of representations. Our VPC remedies this by imposing a geometric prior and allocating OOD features to a predefined, expandable subspace, trained jointly with OE.

**Evidential Deep Learning.** Evidential deep learning (EDL) characterizes classification uncertainty via a Dirichlet evidence formulation (Sensoy et al., 2018), drawing on Dempster Shafer theory and subjective logic (Sentz & Ferson, 2002; Jøsang, 2016). It has been widely adopted in various tasks (Gao et al., 2024), ranging from open-set recognition (Bao et al., 2021; Wang et al., 2024b; Yu et al., 2024) and continual learning (Aguilar et al., 2023) to multi-view classification (Liu et al., 2022; 2025a;b;c), while also being generalized to regression to model both aleatoric and epistemic uncertainty (Amini et al., 2020). Departing from the common view of EDL as an output-layer calibration tool, we embed evidential prior into feature–prototype alignment: controlling the rate of evidence generation to ensure stable convergence of ID features to EBVs while mitigating overbias caused by optimization interference from OE samples and hard outliers. Coupled with our geometric constraints, VEBV subspace attraction and orthogonal separation from the ID subspace, evidential prior driven alignment produces a consistent train–test discrimination behavior.

**Neural Collapse.** Neural Collapse (NC) captures late-stage geometry within-class collapse, simplex ETF means, and classifier means alignment(Papyan et al., 2020; Han et al., 2021) which has been exploited in incremental/continual, few-shot, and large-scale settings(Yang et al., 2023; Seo et al., 2024; Shen et al., 2025), but is sensitive to imbalance and drift (Markou et al., 2024; Yan et al., 2024; Fang et al., 2021). For OOD, NC inspired two main approaches: one leverages the emergent phenomenon, such as in PFS (Wu et al., 2024), by applying constraints to the learned ID classifier weights. The second approach leverages the ideal geometry as prototypes (Shen et al., 2025). Recently, Zou et al. (2025) utilized EBVs as a foundational framework for OOD detection and corrected the angular misalignment between EBV prototypes and data distributions. Furthermore, this framework relates to Prototype Learning (PL), but with critical differences: most PL methods (Peng et al., 2025; Lu et al., 2024) rely on dynamic prototypes that are learned and updated, with OOD detection dependent on these dynamic ID representations. LPO (Zhou et al., 2021) introduces learnable OOD prototypes via manifold mixup. However, the limited diversity of internally synthesized features risks prototype collapse due to insufficient semantic separation from ID classes. While Bojanowski & Joulin (2017) and Saadabadi et al. (2024) also exploit fixed target structures for unsupervised learning or dynamic assignment, our method adopts a fixed prototype strategy which based on the ideal NC geometry and keep them fixed. This avoids the instability of dynamic prototypes while providing a more active geometric constraint than Wu et al. (2024). Crucially, we expand the ID-presetted EBVs into our OEFS to simultaneously represent both ID and OOD, yielding interpretable separation and a straightforward L2 activation-strength scoring rule.

## 3 METHOD

### 3.1 PRELIMINARY

**Formalizing the OOD Detection Paradigm.** Detecting inputs outside the training distribution is critical for deploying reliable models in open environments (Hendrycks & Gimpel, 2016; Liu et al., 2020). We consider a classification task with $C$ classes over input space $\mathcal{X}$ and a corresponding ID label space $\mathcal{Y} = \{1, \ldots, C\}$. The in-distribution (ID) data $(x, y)$ is drawn from a joint distribution $D_{in}(\mathcal{X}, \mathcal{Y})$. Out-of-distribution (OOD) data $x_{ood}$ is drawn from a marginal distribution $D_{out}(\mathcal{X})$, which originates from a different generative process $D_{out}(\mathcal{X}, \mathcal{Y}_{out})$. The core assumption of OOD detection is that the OOD label space $\mathcal{Y}_{out}$ is disjoint from the ID label space, i.e., $\mathcal{Y} \cap \mathcal{Y}_{out} = \emptyset$. The objective is to design a decision function $G_\lambda : \mathcal{X} \mapsto \{\text{ID}, \text{OOD}\}$ based on a scoring function $S$:

$$G_\lambda(x) = \begin{cases} \text{ID} & \text{if } S(x, f) \geq \lambda \\ \text{OOD} & \text{if } S(x, f) < \lambda \end{cases}$$

where $\lambda$ is calibrated on ID data to achieve a target true positive rate. In practice, $S$ can be instantiated by MSP (Hendrycks & Gimpel, 2016), energy (Liu et al., 2020), Mahalanobis distance (Lee et al., 2018), or $k$-NN distances (Sun et al., 2022).

**Revisiting the Outlier Exposure Paradigm.** Recent advances demonstrate that exposing models to auxiliary OOD data during training significantly enhances OOD detection robustness (Hendrycks et al., 2018; Liu et al., 2020; Wang et al., 2024a). Let $D_{\text{out}}^{\text{aux}}$ denote an auxiliary OOD dataset satisfying $\text{supp}(D_{\text{out}}^{\text{aux}}) \cap \text{supp}(D_{in}) = \emptyset$ and $\text{supp}(D_{\text{out}}^{\text{aux}}) \neq \text{supp}(D_{\text{out}})$. The seminal Outlier Exposure (OE) framework (Hendrycks et al., 2018) regularizes the feature space by minimizing the KL divergence between OOD predictions and a uniform distribution:

$$\mathcal{L}_{\text{OE}}(x) = -\frac{1}{C}\sum_{j=1}^{C} \log f_j(x) = \mathcal{H}(f(x), \mathbf{u}_C), \tag{1}$$

where $\mathcal{H}$ denotes cross-entropy, $f_j(x)$ is the $j$-th element of the model output, and $\mathbf{U}_C$ is the uniform distribution over $C$ classes. The composite optimization objective becomes:

$$\min_f \underbrace{\mathbb{E}_{(x,y)\sim D_{\text{in}}}\mathcal{L}_{\text{CE}}(x, y)}_{\text{ID classification}} + \lambda \underbrace{\mathbb{E}_{x\sim D_{\text{out}}^{\text{aux}}}\mathcal{L}_{\text{OE}}(x)}_{\text{OOD regularization}} \tag{2}$$

This formulation establishes the foundational paradigm for OE-based OOD-aware training (Ndiour et al., 2020; Du et al., 2022; Wu et al., 2023). However, this paradigm suffers from fundamental limitations. First, it forces a capacity-limited ID classifier (designed for $K$ classes) into a conflicting-objective dilemma: it must simultaneously perform fine-grained ID discrimination while rejecting massive, diverse OOD data via uniform, low-confidence predictions.

More critically, the persistent OOD regularization severely impedes the alignment of ID features. While ID features should ideally converge to a Neural Collapse (NC) state for maximal separability, the OE objective interferes with this convergence by forcing the classifier to also manage OOD inputs. The resulting feature space is thus a compromise, sacrificing the compactness and stability of ID representations. This analysis reveals a critical need for a solution beyond mere output regularization that can structurally decouple ID and OOD representations. This would protect the ID feature geometry's convergence while providing a dedicated OOD space. Overcoming this limitation is the primary motivation for our VPC framework.

## 3.2 ORTHOGONAL EQUIANGULAR FEATURE SPACE

The core intuition of VPC is to explicitly engineer the feature space geometry, creating a structural separation between ID and OOD representations. We move beyond OOD detection paradigms that rely solely on ID class weights/classifiers. Instead, we propose the Orthogonal Equiangular Feature Space (OEFS), a pre-specified representation space that is conceptually partitioned a prior.

This space is mathematically constructed from a single set of $K + V$ total prototype vectors, $W = \{w_i\}_{i=1}^{K+V}$, generated on the unit hypersphere. To ensure maximal and uniform separation between all prototypes, these vectors are structured as a simplex Equiangular Tight Frame (ETF) (Papyan et al., 2020; Yang et al., 2023; Markou et al., 2024; Shen et al., 2025). This construction provides the geometric foundation for the OEFS and is defined by the relation:

$$w_{k_1}^{\top} w_{k_2} = \frac{K+V}{K+V-1}\delta_{k_1,k_2} - \frac{1}{K+V-1}, \quad \forall k_1, k_2 \in \{1, \dots, K+V\}, \tag{3}$$

where $\delta_{k_1,k_2} = 1$ if $k_1 = k_2$ and 0 otherwise. This guarantees all vectors share the same $\ell_2$ norm and any two distinct vectors have an inner product of $-1/(K+V-1)$.

The key to our method is the functional partitioning of this unified set of prototypes. The first $K$ vectors are designated as Equiangular Basic Vectors (EBVs); they form the ID subspace and serve as the stable, fixed prototypes for the $K$ in-distribution classes. The remaining $V$ vectors are designated as Vast EBVs (VEBVs), establishing a dedicated OOD subspace. Crucially, this subspace is geometrically orthogonal to the ID boundaries to ensure non-interference with ID tasks, while simultaneously offering a vast array of distinct geometric anchors to resolve the fine-grained semantic variations inherent in OOD data.

Building on this preset geometry, we adopt a dual feature-alignment strategy (Sec. 3.3, 3.4) to induce distinct activation patterns on EBVs and VEBVs for ID and OOD samples, respectively. This drives ID features toward Neural Collapse geometry while granting OOD features sufficient directional resolution to differentiate complex patterns; theoretical discussion of Neural Collapse is provided in the appendix A.1. This geometric separation is measured by our VPC Score 3.5, a discriminative metric designed to quantify the activation of a feature within these distinct subspaces.

### 3.3 EVIDENTIAL PRIOR GUIDED FEATURE ALIGNMENT

In guiding ID features toward the target subspace, a central challenge is that OE features with rich variability can interfere with the model at any time and destabilize the representation space. To counter this, we propose an Evidential Neural Collapse framework that reformulates feature alignment as a geometry-driven evidence accumulation process.

Unlike standard evidential learning (Sensoy et al., 2018; Jøsang, 2016; Sentz & Ferson, 2002) where evidence stems from learnable logits $g(x \mid \theta)$, we instantiate a geometry-driven evidence metric derived directly from the cosine proximity between features and the predefined EBVs. This couples the accumulation of evidence with the rigor of geometric alignment:

$$e_{i,j} = \exp\Big( \underbrace{g_j(x_i^{\mathrm{id}}; \theta)}_{\substack{\text{Learnable} \\ \text{Classifier Logits}}} \Big) \xLeftrightarrow{\text{Predefined VPC}} e_{i,j}^* = \exp\Big( \underbrace{\hat{m}_i^{\mathrm{id}\top} \hat{w}_j^{\mathrm{ebv}}/\tau}_{\substack{\text{Geometric} \\ \text{Alignment Logits}}} \Big). \tag{4}$$

Here, the evidence magnitude $e_{i,j}^*$ reflects the angular alignment intensity of the normalized feature $\hat{m}_i^{\mathrm{id}}$ towards the $j$-th prototype $\hat{w}_j^{\mathrm{ebv}}$, scaled by a temperature $\tau$.

Crucially, we fuse this geometric evidence with a uniform prior to form the Dirichlet parameters $\alpha_{i,j}^* = e_{i,j}^* + 1$. This unit prior acts as an angular regulator: it injects a uniform geometric buffer that prevents the model from collapsing onto specific directions too abruptly due to optimization interference from OE samples. By maximizing the following ENC likelihood, we achieve a calibrated convergence where evidence growth is strictly governed by reliable geometric support:

$$\mathcal{L}_{\mathrm{ENC}}(x_i^{\mathrm{id}}) = \sum_{j=1}^{K} y_{i,j} \left( \log S_i^* - \log \alpha_{i,j}^* \right), \tag{5}$$

where $S_i^* = \sum_{k=1}^{K} \alpha_{i,k}^*$ is the total evidential strength and $y_{i,j}$ is the one-hot label. This objective effectively mitigates over-confidence and ensures stable ID-EBVs alignment, which we further theoretically justify via optimality and stability analyses in Appendix A.2.1 A.2.2.

### 3.4 DUAL-SUBSPACE FEATURE ATTRACTION AND ORTHOGONALITY CONSTRAINTS

To fully exploit the benefits of abundant auxiliary OE samples for OOD detection, we move beyond the traditional OE loss that only enforces uniformization over ID classifier weights, and establish a dual mechanism of subspace attraction and orthogonality constraints. On the one hand, OE features are guided to enter the VEBV subspace so as to enrich OOD representations; on the other hand, they are enforced to be orthogonal to ID EBVs to preserve ID classification performance. For the subspace-attraction objective, we design the VEBV loss by quantifying the Euclidean distance between the normalized OE feature $\hat{m}_i^{\mathrm{oe}}$ and the subspace spanned by the VEBVs:

$$\mathcal{L}_{\mathrm{VEBV}}(x_i^{\mathrm{oe}}) = -\sqrt{\sum_{j=1}^{V} \big(\hat{m}_i^{\mathrm{oe}\top} \hat{w}_j^{\mathrm{vebv}}\big)^2}, \tag{6}$$

where $\hat{w}_j^{\mathrm{vebv}}$ denotes the $j$-th VEBV in OEFS. The summand is the squared cosine similarity between the normalized OE feature and each VEBV; The square root (an $\ell_2$ norm) measures the activation strength within the VEBV subspace. Minimizing the negative of this norm maximizes the activation of outlier features in the VEBV subspace, forcing them to distribute activation over multiple VEBVs rather than collapsing onto a single direction, thereby capturing the diversity of OOD modes.

In addition, we introduce an orthogonality constraint (OC) loss to strictly regulate OE feature alignment within the ID subspace. Formally, it minimizes the KL divergence between the predicted distribution over ID EBVs and a uniform distribution:

$$\mathcal{L}_{\mathrm{OC}}(x_i^{\mathrm{oe}}) = -\frac{1}{K} \sum_{j=1}^{K} \log(\mathbf{p}_j^{\mathrm{oe}\to\mathrm{ebv}}). \tag{7}$$

Here, $\mathbf{p}_j^{\mathrm{oe}\to\mathrm{ebv}}$ is computed by applying Softmax to the scaled cosine similarities $s_j = (\hat{m}_i^{\mathrm{oe}\top} \hat{w}_j^{\mathrm{ebv}})/\tau$. By virtue of the Simplex ETF structure, this uniformity objective effectively translates into a geometric orthogonality constraint (proof provided in Appendix A.3). This ensures that OE features reside in the null space of ID prototypes, safeguarding ID convergence and creating rigorous decision boundary.

## 3.5 DISCRIMINATIVE SCORING VIA ACTIVATION INTENSITY

The geometric partitioning induced by the OEFS enables a principled and discriminative scoring function. This function quantifies the distinct activation patterns of ID and OOD features across the predefined classifier subspaces. We define two metrics based on the L2 norm of a feature's projection onto the ID and OOD subspaces, respectively:

$$
\begin{aligned}
\ell_2^{\text{ebv}}(x_i) &= \left\| \left[ \hat{m}_i^\top \hat{w}_1^{\text{ebv}}, \ldots, \hat{m}_i^\top \hat{w}_K^{\text{ebv}} \right] \right\|_2, \\
\ell_2^{\text{vebv}}(x_i) &= \left\| \left[ \hat{m}_i^\top \hat{w}_1^{\text{vebv}}, \ldots, \hat{m}_i^\top \hat{w}_V^{\text{vebv}} \right] \right\|_2.
\end{aligned}
\tag{8}
$$

Here, $\ell_2^{\text{ebv}}(x_i)$ and $\ell_2^{\text{vebv}}(x_i)$ measure the activation intensity of a feature $\hat{m}_i$ within the ID (EBV) and OOD (VEBV) subspaces, respectively.

Following optimization with our proposed loss, ID features align with their EBVs, yielding high $\ell_2^{\text{ebv}}$ and negligible $\ell_2^{\text{vebv}}$ activations. OOD features are conversely guided into the VEBV subspace, producing the opposite pattern. This resulting dichotomy motivates the VPC Score:

$$
S_{\text{VPC}}(x_i) = -\alpha \cdot \ell_2^{\text{ebv}}(x_i) + \beta \cdot \ell_2^{\text{vebv}}(x_i),
\tag{9}
$$

Consequently, the VPC score serves as a continuous measure of OOD likelihood, allowing for the separation of ID and OOD samples via a simple threshold. In contrast to methods that rely on the statistical properties of logits implicitly learned, the VPC Score is derived directly from the prior geometric structure of the feature space. By measuring the relative activation intensity across orthogonal subspaces, it provides a robust and interpretable measure of distributional shift, grounded in the model's explicit geometric constraints.

## 4 EXPERIMENTS

In this section, we first evaluate our method on the large-scale ImageNet-1k benchmark and the widely-used CIFAR-10/100 benchmarks (Krizhevsky et al., 2009) to assess its performance (Sec. 4.1). We then examine a variety of model architectures to further verify its effectiveness (Secs. 4.2, 4.3, and 4.4). Additional ablation studies are reported in Sec. 4.5. More theoretical analysis, results appear in appendix A B. We begin by detailing the experimental setup.

**OOD Datasets.** For CIFAR benchmarks, we use 300K auxiliary samples from 80 Million Tiny Images (Torralba et al., 2008) and evaluate on five standard test datasets with disjoint categories: SVHN (Netzer et al., 2011), LSUN (Yu et al., 2015), iSUN (Xu et al., 2015), Texture (Cimpoi et al., 2014), and Places365 (Zhou et al., 2017). For ImageNet, we utilize the ImageNet-21k-p validation subset as auxiliary data and test on four widely-recognized datasets: iNaturalist (Van Horn et al., 2018), SUN (Xiao et al., 2010), Places (Zhou et al., 2017), and Textures (Cimpoi et al., 2014).

**Pre-training Setups.** We employ Wide ResNet-40-2, ResNet-18, and DenseNet-121 (Zagoruyko & Komodakis, 2016; He et al., 2016; Huang et al., 2017) as backbones for the CIFAR benchmarks, training for 200 epochs with a batch size of 128, initial learning rate 0.1, momentum 0.9, weight decay 0.0005, and a cosine learning-rate schedule. For the ImageNet-1k benchmark, in the comparative experimental methods, we use standard pre-trained ResNet-50 (He et al., 2016) and ViT-B-16 models as backbones. In the VPC experiments, we follow the standard practice for EBV-based training (Shen et al., 2025). Unlike the conventional cross-entropy + learnable classifier head, our method directly maximizes the similarity between features and predefined classifiers. Accordingly, we remove the final classification layer and add an extra projection layer to match the dimensional requirements of OEFS; see the Appendix A.4 B.3 for details.

**Two-stage Training Setting.** Fine-tuning setup. For the CIFAR-10 and CIFAR-100 benchmarks, we initialize the network with the best-performing checkpoint from the pretraining stage and fine-tune for 50 epochs with auxiliary OE data (Hendrycks et al., 2018) (ID batch size 128; OOD batch size 256; initial learning rate 0.07; momentum 0.9; weight decay 0.0005; cosine annealing schedule). For the ImageNet benchmark, we use pre-trained models from Pytorch and pre-trained models following the (Shen et al., 2025) training setup as the initial networks, respectively, and then fine-tune for 5 epochs with ID/OOD batch size 64, initial learning rate 1e-4, momentum 0.9, weight decay 0.0005, and a cosine schedule. For our VPC method, the temperature $\tau$ used in all loss functions is set to 0.1. All other settings follow the original paper's setup.

Table 1: Results on ImageNet-1k benchmark with auxiliary OOD data. The best result is in **bold**.

| Model | Method | Far-OOD Datasets | | | | Near-OOD Datasets | | | | Average | | ID Acc↑ |
|---|---|---|---|---|---|---|---|---|---|---|---|---|
| | | iNaturalist | | Textures | | SUN | | Places | | | | |
| | | FPR95↓ | AUROC↑ | FPR95↓ | AUROC↑ | FPR95↓ | AUROC↑ | FPR95↓ | AUROC↑ | FPR95↓ | AUROC↑ | |
| ResNet50 | OE Hendrycks et al. (2018) | 48.60 | 88.72 | 58.85 | 82.60 | 61.75 | 82.90 | 70.70 | 80.55 | 59.98 | 83.69 | 76.00 |
| | Energy-OE Liu et al. (2020) | 49.40 | 88.40 | 59.60 | 82.25 | 62.40 | 82.70 | 71.30 | 80.25 | 60.68 | 83.40 | 75.75 |
| | DAL Wang et al. (2024a) | 48.00 | 89.05 | 58.00 | 82.95 | 61.30 | 83.15 | 67.82 | 80.75 | 58.78 | 83.98 | 75.90 |
| | PFS Wu et al. (2024) | 46.40 | 89.20 | 56.50 | **83.10** | 61.00 | **83.25** | 67.50 | 80.95 | 57.85 | 84.13 | 76.02 |
| | **Ours: VPC** | **43.50** | **91.20** | **56.00** | 83.00 | **60.20** | 83.10 | **66.30** | **81.50** | **56.50** | **84.70** | **76.11** |
| ViT-B-16 | OE Hendrycks et al. (2018) | 42.15 | 90.38 | 52.45 | 85.85 | 65.80 | 82.20 | 70.35 | 80.85 | 57.69 | 84.82 | 80.02 |
| | Energy-OE Liu et al. (2020) | 42.75 | 90.05 | 53.15 | 85.50 | 66.25 | 81.95 | 70.95 | 80.50 | 58.28 | 84.50 | 79.85 |
| | DAL Wang et al. (2024a) | 40.65 | 90.86 | **51.15** | **86.10** | 65.07 | 82.30 | 70.25 | 80.90 | 56.78 | 85.04 | 80.09 |
| | PFS Wu et al. (2024) | 40.85 | 90.80 | 51.30 | 86.05 | 65.35 | 82.25 | 70.20 | 80.95 | 56.93 | 85.01 | 80.13 |
| | **Ours: VPC** | **40.00** | **91.10** | 51.30 | 86.00 | 65.20 | **82.30** | **68.30** | **81.40** | **56.20** | **85.20** | **80.32** |

Table 2: Results on CIFAR-10 and CIFAR-100 with WideResNet-40-2. The best result is in **bold**.

| Method | Far-OOD Datasets | | | | | | | | | | Average | |
|---|---|---|---|---|---|---|---|---|---|---|---|---|
| | SVHN | | LSUN | | iSUN | | Textures | | Places365 | | | |
| | FPR95↓ | AUROC↑ | FPR95↓ | AUROC↑ | FPR95↓ | AUROC↑ | FPR95↓ | AUROC↑ | FPR95↓ | AUROC↑ | FPR95↓ | AUROC↑ |
| CIFAR-10 | | | | | | | | | | | | |
| With vanilla training | | | | | | | | | | | | |
| MSP Hendrycks & Gimpel (2016) | 44.22 | 93.61 | 27.56 | 96.12 | 69.62 | 85.29 | 60.02 | 88.53 | 65.68 | 86.25 | 53.42 | 89.96 |
| Energy Liu et al. (2020) | 31.81 | 94.65 | 4.60 | 98.96 | 50.06 | 89.75 | 49.68 | 90.09 | 42.28 | 90.82 | 35.69 | 92.85 |
| Maha Lee et al. (2018) | 42.67 | 90.71 | 18.96 | 96.46 | 28.86 | 93.76 | 26.22 | 92.81 | 86.78 | 69.14 | 40.70 | 88.58 |
| KNN Sun et al. (2022) | 44.76 | 92.55 | 27.38 | 95.34 | 43.84 | 91.24 | 37.64 | 92.82 | 49.23 | 87.89 | 40.57 | 91.97 |
| With contrastive learning | | | | | | | | | | | | |
| CSI Tack et al. (2020) | 17.37 | 97.69 | 6.75 | 98.46 | 12.58 | 97.95 | 25.65 | 94.70 | 40.00 | 92.05 | 20.47 | 96.17 |
| CIDER Ming et al. (2022b) | 6.76 | 98.44 | 7.45 | 98.76 | 26.03 | 95.93 | 22.85 | 95.75 | 43.70 | 91.94 | 21.36 | 96.16 |
| KNN+ Sun et al. (2022) | 3.28 | 99.33 | 2.24 | 98.90 | 17.85 | 97.65 | 10.87 | 97.92 | 30.63 | 94.98 | 12.97 | 97.32 |
| With auxiliary OOD data | | | | | | | | | | | | |
| OE Hendrycks et al. (2018) | 1.95 | 99.23 | 0.80 | 99.67 | 1.95 | 99.36 | 3.70 | 99.23 | 8.80 | 97.76 | 3.44 | 99.05 |
| Energy-OE Liu et al. (2020) | 1.90 | 99.32 | 0.95 | 98.99 | 3.35 | 98.72 | 4.00 | 98.85 | 8.55 | 97.42 | 3.75 | 98.66 |
| DAL Wang et al. (2024a) | 1.40 | 99.36 | 0.95 | 99.53 | 1.35 | 99.02 | 3.50 | 98.99 | 8.65 | 97.39 | 3.17 | 98.84 |
| PFS Wu et al. (2024) | 1.10 | 98.74 | 0.35 | 99.61 | 1.35 | 99.20 | 2.85 | 98.58 | 7.75 | 97.17 | 2.68 | 98.66 |
| **Ours: VPC** | 0.85 | 99.62 | 0.45 | 99.50 | 1.10 | 99.50 | 2.25 | 99.38 | 6.70 | 97.91 | **2.27** | **99.18** |
| CIFAR-100 | | | | | | | | | | | | |
| With vanilla training | | | | | | | | | | | | |
| MSP Hendrycks & Gimpel (2016) | 74.79 | 79.64 | 54.72 | 86.46 | 93.85 | 56.92 | 88.76 | 68.48 | 83.24 | 71.95 | 79.07 | 72.69 |
| Energy Liu et al. (2020) | 70.18 | 87.15 | 17.15 | 97.05 | 91.37 | 65.50 | 84.77 | 76.72 | 78.91 | 75.77 | 62.75 | 80.44 |
| Maha Lee et al. (2018) | 77.73 | 78.01 | 98.46 | 63.44 | 47.74 | 88.76 | 54.93 | 82.53 | 97.22 | 54.11 | 75.22 | 73.37 |
| KNN Sun et al. (2022) | 71.86 | 83.31 | 78.89 | 70.09 | 79.60 | 70.86 | 72.89 | 80.05 | 80.91 | 71.33 | 76.83 | 75.13 |
| With contrastive learning | | | | | | | | | | | | |
| CSI* Tack et al. (2020) | 64.50 | 84.62 | 25.88 | 95.93 | 70.62 | 80.83 | 61.50 | 86.74 | 83.08 | 77.11 | 61.12 | 85.05 |
| CIDER Ming et al. (2022b) | 16.47 | 96.23 | 45.45 | 81.64 | 66.01 | 82.21 | 49.79 | 87.48 | 82.66 | 68.39 | 52.08 | 83.19 |
| KNN+* Sun et al. (2022) | 32.50 | 93.86 | 47.41 | 84.93 | 39.82 | 91.12 | 43.05 | 88.55 | 63.26 | 79.28 | 45.20 | 87.55 |
| With auxiliary OOD data | | | | | | | | | | | | |
| OE Hendrycks et al. (2018) | 28.95 | 95.08 | 10.95 | 97.98 | 49.55 | 89.29 | 41.50 | 91.57 | 49.75 | 89.87 | 36.14 | 92.76 |
| Energy-OE Liu et al. (2020) | 23.80 | 96.18 | 31.90 | 94.88 | 41.40 | 91.67 | 48.10 | 88.09 | 56.50 | 87.66 | 40.34 | 91.69 |
| DAL Wang et al. (2024a) | 19.30 | 95.75 | 16.20 | 96.71 | 30.70 | 93.85 | 43.15 | 91.36 | 55.10 | 88.39 | 32.89 | 93.21 |
| PFS Wu et al. (2024) | 24.70 | 95.81 | 12.65 | 97.78 | 38.35 | 91.44 | 44.20 | 91.32 | 51.85 | 90.33 | 34.35 | 93.33 |
| **Ours: VPC** | 9.95 | 97.98 | 26.25 | 95.75 | 26.50 | 94.97 | 45.05 | 89.67 | 52.45 | 89.88 | **32.04** | **93.65** |

**One-stage Training Setting.** Beyond the stepwise pretraining fine-tuning paradigm for OOD models, we additionally investigate a one-stage training scheme to explore stable separation between ID and OOD samples under a more entangled optimization setting. On the CIFAR-10 benchmark, we jointly train on ID and OE data for 150 epochs; on CIFAR-100, we train for 200 epochs. All other settings follow the two-stage experiments.

**Scoring functions.** We compare the classic MSP (Maximum Softmax Probability)(Hendrycks & Gimpel, 2016), the EDL (Evidential Deep Learning) Uncertainty Score (Sensoy et al., 2018), and our proposed VPC Score. In the default configuration, VPC Score uses $\alpha = -1$ and $\beta = 100$. We also evaluate single-subspace variants: VPC Score* ($\alpha = -1$, $\beta = 0$, ID-EBVs subspace) and VPC Score† ($\alpha = 0$, $\beta = 100$, OOD-VEBVs subspace).

**Compared Methods.** We compare our method with post-hoc approaches, contrastive learning based methods, and auxiliary OOD data based methods. The post-hoc methods include MSP (Hendrycks & Gimpel, 2016), Energy (Liu et al., 2020), Maha (Lee et al., 2018), and KNN (Sun et al., 2022). The contrastive learning based methods include CSI (Tack et al., 2020), CIDER (Ming et al., 2022b), and KNN+ (Sun et al., 2022). The auxiliary OOD data based methods include OE (Hendrycks et al., 2018), Energy-OE (Liu et al., 2020), DAL (Wang et al., 2024a) and PFS (Wu et al., 2024). For other methods, we adopt their suggested setups for fairness.

**Evaluation Metrics.** We report three classic metrics commonly used in OOD detection (Hendrycks & Gimpel, 2016; Liang et al., 2017): (i) FPR95 (False Positive Rate at 95% True Positive Rate): the FPR for OOD samples when the TPR for ID samples reaches 95%. (ii) AUROC (Area Under the Receiver

Operating Characteristic Curve): the area under the TPR-FPR curve over all thresholds. (iii) AUPR (Area Under the Precision-Recall Curve): the area under the precision-recall curve, emphasizing the balance between OOD detection accuracy and coverage.

## 4.1 Main Results

Tables 1 and 2 present the primary results on the large-scale ImageNet-1k benchmark and the widely-used CIFAR-10/100 benchmarks, respectively. Compared with conventional supervised or contrastive learning methods that rely solely on ID data, incorporating auxiliary outlier exposure (OE) data during training significantly reduces FPR95 and enhances AUROC, clearly highlighting the value of OE-based training in OOD detection research. We compare our approach against several state-of-the-art OE-based methods, including the original OE method (Hendrycks et al., 2018), the Energy-based OE method (Energy-OE (Liu et al., 2020)), the Distributional Adversarial Learning (DAL (Wang et al., 2024a)), and the PFS method leveraging Neural Collapse (NC) (Papyan et al., 2020) properties. Our proposed VPC achieves superior performance across all benchmarks. On ImageNet-1k (Table 1), VPC demonstrates robust advantages. With the ResNet50 backbone, VPC achieves an average FPR95 of 56.50%, significantly outperforming the next-best baseline (PFS) by 1.35% while also attaining the highest AUROC. This advantage is maintained on the ViT-B-16 backbone, where VPC (56.20%) surpasses the strongest competitors (DAL at 56.78% and PFS at 56.93%) while also achieving the highest AUROC (85.20%) and maintaining competitive ID accuracy. This strong performance is mirrored on the CIFAR benchmarks (Table 2). On CIFAR-10, VPC reduces the average FPR95 to 2.27%, clearly surpassing the best-performing baseline (PFS at 2.68%). On CIFAR-100, VPC achieves an average FPR95 of 32.04%, outperforming both DAL (32.89%) and PFS (34.35%), and again reaches the highest AUROC score of 93.65% among the compared methods.

## 4.2 two-stage Training

Under the two-stage pretraining fine-tuning setup (Table 3), our method achieves overall superiority on CIFAR-10 with WideResNet-40-2 and DenseNet-121, and attains the best FPR95 on ResNet-18 while its AUROC/AUPR are slightly below DAL. On CIFAR-100, it shows robust advantages on WideResNet-40-2 (best FPR95 and AUROC, with AUPR tied with PFS) and on DenseNet-121 (best on all three metrics), whereas on ResNet-18 it minimizes FPR95 but trails PFS in overall ranking and PR area. Collectively, these results indicate that our approach effectively suppresses OOD false acceptance while improving overall separability.

Table 3: Two-stage Training Results on CIFAR-10 and CIFAR-100 with WideResNet-40-2, ResNet-18, DenseNet-121. Metrics are reported as FPR95↓/AUROC↑/AUPR↑.

| Method | CIFAR-10 | | | CIFAR-100 | | |
|---|---|---|---|---|---|---|
| | WideResNet-40-2 | ResNet-18 | DenseNet-121 | WideResNet-40-2 | ResNet-18 | DenseNet121 |
| OE Hendrycks et al. (2018) | 3.44/99.05/99.79 | 3.46/98.36/99.67 | 2.84/**98.86**/99.75 | 36.14/92.76/98.38 | 48.75/89.36/97.54 | 36.46/93.23/98.53 |
| Energy-OE Liu et al. (2020) | 3.75/98.66/99.69 | 3.88/98.26/99.62 | 3.45/98.71/ 99.71 | 40.34/91.69/98.00 | 46.34/90.70/97.91 | 44.87/91.82/98.21 |
| DAL Wang et al. (2024a) | 3.17/98.84/99.74 | 3.02/ **98.96/99.77** | 2.58/98.72/99.71 | 32.89/93.21/98.44 | 44.57/90.87/97.99 | 36.75/90.66/97.68 |
| PFS Wu et al. (2024) | 2.68/98.66/99.65 | 3.05/98.72/99.72 | 2.87/98.47/99.66 | 34.35/93.33/**98.53** | 40.15/ **92.64/98.41** | 43.80/90.96/97.99 |
| Ours | **2.27/99.18/99.81** | **2.84**/98.32/99.67 | **2.10/98.86/99.77** | **32.04/93.65/98.53** | **38.96**/92.03/98.20 | **31.17/94.01/98.71** |

## 4.3 one-stage Training

As reported in Table 4, we further investigate a single-stage training paradigm, which differs from the prior two-stage setup. The goal is to train OOD detectors directly with fewer steps and hyperparameters, this direct setup has been underexplored because it exacerbates the difficulty of controlling representation geometry. When jointly optimizing ID and OE data without stage separation, our method attains highly competitive results across the board. This corroborates the effectiveness of imposing the OEFS geometric prior on the representation space and, via a dual-optimization strategy, strengthens VPC's ability to suppress representation drift under noisy gradients while maintaining stable ID/OOD decision boundaries.

Table 4: One-stage Training Results on CIFAR-10 and CIFAR-100 with WideResNet-40-2, ResNet-18, DenseNet-121. Metrics are reported as FPR95↓/AUROC↑/AUPR↑.

| Method | CIFAR-10 | | | CIFAR-100 | | |
|---|---|---|---|---|---|---|
| | WideResNet-40-2 | ResNet-18 | DenseNet-121 | WideResNet-40-2 | ResNet-18 | DenseNet121 |
| OE Hendrycks et al. (2018) | 2.74/99.01/99.79 | 3.86/98.43/99.68 | 3.12/98.61/99.70 | 35.43/92.67/98.33 | 45.83/91.80/98.26 | 37.77/92.68/98.37 |
| Energy-OE Liu et al. (2020) | 2.29/98.79/99.72 | 3.66/98.21/99.63 | 3.09/98.76/99.73 | 36.08/92.76/98.28 | 41.79/91.90/98.18 | 34.12/93.30/98.44 |
| DAL Wang et al. (2024a) | 2.95/98.88/99.75 | 3.71/98.44/99.67 | 2.73/97.74/99.53 | 32.34/92.28/98.24 | 43.10/91.76/98.16 | 36.55/92.63/98.34 |
| PFS Wu et al. (2024) | 2.44/98.87/99.68 | 3.64/98.72/99.69 | 2.42/**98.95/99.75** | 32.84/93.22/98.48 | 41.14/91.70/98.14 | 39.91/91.76/98.15 |
| Ours | **2.01/99.19/99.82** | **3.45/98.73/99.73** | **2.29**/98.74/99.74 | **32.15/93.65/98.50** | **38.41/92.69/98.42** | **32.31/93.41/98.54** |

## 4.4 DIFFERENT SCORE FUNCTIONS

This section provides a detailed comparison between the proposed VPC Score and a range of scoring functions across multiple network architectures (see Table 5). Built upon the twin-subspace representation of OEFS, VPC employs a class-agnostic L2 activation magnitude as the separation signal to enhance OOD discriminability. On CIFAR-100, VPC Score achieves the best AUROC and AUPR on all three backbones and reaches, or closely approaches, the lowest FPR95. By contrast, MSP suffers from bias because OOD features cannot remain orthogonal to the ID classifier weights; EDL Prob introduces evidence priors that partially alleviate this bias yet remains suboptimal; and Uncertainty aggregates evidence strength without imposing explicit geometric constraints on the representation space. Leveraging the orthogonal subspace–induced, class-agnostic L2 activation, VPC Score delivers consistent gains across nearly all architectures while retaining interpretability and strong generalization. Additional discussion of VPC Score is provided in Appendix A.5.

Table 5: Different score functions on CIFAR-10 and CIFAR-100 with WideResNet-40-2, ResNet-18, DenseNet-121. Metrics are reported as FPR95↓/AUROC↑/AUPR↑.

| Score Function | CIFAR-10 | | | CIFAR-100 | | |
|---|---|---|---|---|---|---|
| | WideResNet-40-2 | ResNet-18 | DenseNet-121 | WideResNet-40-2 | ResNet-18 | DenseNet-121 |
| MSP | 2.74/99.18/**99.81** | 3.13/98.81/**99.75** | 2.46/98.99/99.79 | 33.95/92.36/98.11 | 44.75/89.79/97.44 | 33.65/92.79/98.36 |
| EDL Prob | 2.55/**99.19/99.81** | 3.07/**98.82/99.75** | 2.39/99.00/99.79 | 33.87/92.57/98.16 | 44.47/90.08/97.53 | 33.42/93.00/98.42 |
| Uncertainty | 2.30/99.13/99.80 | 2.91/98.72/99.74 | **2.06/99.08/99.80** | **31.20**/93.26/98.32 | 39.58/91.23/97.83 | 33.42/93.00/98.42 |
| VPC Score | **2.27/99.19/99.81** | **2.84**/98.32/99.67 | 2.10/98.86/99.77 | 32.04/**93.65/98.53** | **38.96/92.03/98.20** | **31.17/94.01/98.71** |

## 4.5 ABLATION RESULTS

### 4.5.1 SCALE OF ORTHOGONAL EQUIANGULAR FEATURE SPACE

In a fixed ID dimensionality (CIFAR-10 with $K=10$, CIFAR-100 with $K=100$), we conduct an ablation on the VEBV subspace size $V$ (see Table 6). On CIFAR-10, as $V$ increases from 10 to 2000, performance improves monotonically, and indicates that a larger VEBV subspace provides more fine-grained geometric anchors, enabling more precise resolution of diverse OOD modes and strengthening the subspace's L2 activation. On CIFAR-100, performance peaks at $V=1000$, suggesting that when $K$ is large, excessively increasing $V$ causes subspace dispersion and more diffuse gradients, thereby weakening optimization pressure toward that subspace.

Table 6: Ablation on OEFS subspace size $V$ (number of VEBVs) with WideResNet-40-2 backbone. Metrics are reported as FPR95↓/AUROC↑/AUPR↑.

| OEFS | CIFAR-10 | OEFS | CIFAR-100 |
|---|---|---|---|
| 10 | 2.68/98.96/99.75 | 100 | 34.89/92.78/98.39 |
| 500 | 2.61/99.11/99.79 | 500 | 34.65/93.01/98.41 |
| 1000 | 2.27/99.18/99.81 | 1000 | **32.04/93.65/98.53** |
| 2000 | **2.21/99.21/99.85** | 2000 | 32.56/93.31/98.49 |

### 4.5.2 ABLATION ON TRAINING LOSS

This section analyzes different loss combinations to validate the geometric logic of OEFS. As shown in Table 7, we compare: (i) Baseline (CE + OE); (ii) Neural Collapse + orthogonality constraint (see

Appendix §16); (iii) Evidential Neural Collapse ($\mathcal{L}_{\mathrm{ENC}}$) + orthogonality constraint ($\mathcal{L}_{\mathrm{OC}}$); and (iv) the full method adding $\mathcal{L}_{\mathrm{VEBV}}$. The progressive improvements across these configurations clearly indicate that OEFS's core value lies in constructing two orthogonal subspaces, which in turn relies on the synergy of the loss: $\mathcal{L}_{\mathrm{ENC}}$ stabilizes the convergence of ID features toward EBV prototypes, $\mathcal{L}_{\mathrm{OC}}$ strengthens subspace orthogonality, and $\mathcal{L}_{\mathrm{VEBV}}$ guides OOD features to activate the orthogonal VEBVs, thereby explicitly characterizing their complex intrinsic variability. Each component is indispensable. Notably, $\mathcal{L}_{\mathrm{VEBV}}$ is pivotal for breaking the performance bottleneck: its introduction elevates the OEFS geometric separation from ID-side only optimization to a dual-sided ID-OOD constraint, ultimately yielding superior OOD detection performance.

Table 7: Ablation on Training Loss. Metrics are reported as FPR95↓/AUROC↑/AUPR↑.

| Training Loss | CIFAR-10 | CIFAR-100 |
|---|---|---|
| $\mathcal{L}_{\mathrm{CE}} + \mathcal{L}_{\mathrm{OE}}$ | 3.44/99.05/99.79 | 36.14/92.76/98.38 |
| $\mathcal{L}_{\mathrm{NC}} + \mathcal{L}_{\mathrm{OC}}$ | 2.71/98.97/99.74 | 34.65/93.05/98.41 |
| $\mathcal{L}_{\mathrm{ENC}} + \mathcal{L}_{\mathrm{OC}}$ | 2.68/98.96/99.75 | 34.89/92.78/98.39 |
| $\mathcal{L}_{\mathrm{ENC}} + \mathcal{L}_{\mathrm{OC}} + \mathcal{L}_{\mathrm{VEBV}}$ | **2.27/99.18/99.81** | **32.04/93.65/98.53** |

## 4.6 VISUALIZATION

We visualize ID/OOD activations on a unified VPC model under three scoring functions: VPC Score, Uncertainty, and MSP (Figure 2); additional examples appear in Appendix B.4. On CIFAR-10/100, VPC Score produces two sharply separated modes: ID samples primarily excite EBVs, whereas OOD samples activate VEBVs. When challenging OOD data partially align with an ID EBV, MSP is readily confounded. Uncertainty and our class-agnostic VPC Score suppress this bias, with VPC Score achieving superior discriminability.

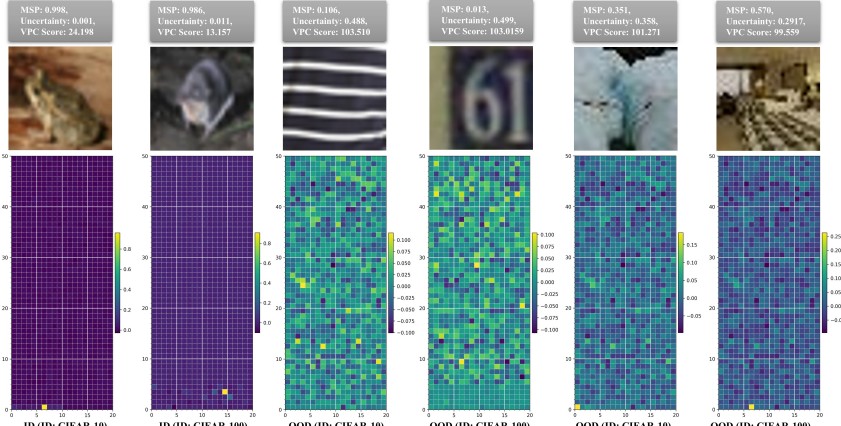

Figure 2: Visualization of ID and OOD detection using different scoring functions in CIFAR-10. VPC Score maintains distinguishability even when MSP and Uncertainty perform poorly.

## 5 CONCLUSION

In this paper, we present VPC, a geometry-driven framework that predefines an OEFS to explicitly separate ID and OOD representations. Beyond ID-only classifiers, VPC injects a training-time prior using an ENC loss for ID–EBV alignment, an orthogonality loss to shield the ID subspace from OOD, and a VEBV exploration loss to enrich the OOD subspace. The resulting VPC Score measures subspace L2 activation, offering class-agnostic, interpretable, and robust discrimination. On CIFAR-10/100 and large scale ImageNet-1k benchmark, and multiple architectures, VPC achieves strong and consistent performance. These findings highlight geometric priors plus evidential modeling as an effective path to feature-separation OOD detection with auxiliary OOD data.

ACKNOWLEDGMENTS

This work was supported by the National Natural Science Foundation of China (Nos. 62476165, 62472315, 62406182), Shanghai Science and Technology Innovation Action Plan (No. 25511102102).

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

# A THEORETICAL ANALYSIS AND METHODOLOGICAL DISCUSSIONS

## A.1 NEURAL COLLAPSE

Recent studies have shown that, in the late stage of training, classification networks exhibit the Neural Collapse geometry: within-class collapse, inter-class equiangular separation, and alignment between class means and classifier prototypes. This geometry achieves simultaneous intra-class compactness and maximal inter-class separation, and is regarded as the natural convergent state of classification models (Papyan et al., 2020; Seo et al., 2024; Yang et al., 2023; Markou et al., 2024).

In OE-based OOD detection, relying only on output-layer uniformization or simple orthogonality constraints is insufficient to maintain a controllable and stable representation geometry during joint ID and OE fine-tuning. We therefore treat NC as an explicit alignment target. We fix the ID subspace to a simplex ETF using preset Equiangular Basic Vectors (EBVs) and expand a bank of Vast EBVs (VEBVs) to characterize diverse OOD features, forming an Orthogonal Equiangular Feature Space (OEFS). We next introduce the simplex ETF induced by EBVs, summarize the core properties of NC, and present the corresponding optimization.

**A Simplex Equiangular Tight Frame** Let $K \geq 2$ and $d \geq K-1$, where $d$ is the feature embedding dimension. A simplex equiangular tight frame (ETF) is a matrix $E = [w_1, \ldots, w_K] \in \mathbb{R}^{d \times K}$ whose columns satisfy

$$w_{k_1}^\top w_{k_2} = \frac{K}{K-1} \delta_{k_1,k_2} - \frac{1}{K-1}, \qquad 1 \leq k_1, k_2 \leq K, \tag{10}$$

so that all columns have the same squared norm $K/(K-1)$ and any two distinct columns have inner product $-1/(K-1)$. A constructive form is

$$E = \sqrt{\frac{K}{K-1}} \, U \left( I_K - \frac{1}{K} \mathbf{1}_K \mathbf{1}_K^\top \right), \qquad U^\top U = I_K, \tag{11}$$

which yields zero-sum columns $E \mathbf{1}_K = \mathbf{0}$ and achieves the minimum possible cosine similarity among $K$ equiangular vectors in $\mathbb{R}^d$. In our setting, these $K$ columns serve as preset EBVs that fix the ID subspace. For OOD representation we append a large bank of VEBVs constructed analogously in the same ambient space with $d \geq K + V - 1$, which together instantiate OEFS and provide preset classifiers for both ID and OOD.

**Basic properties of Neural Collapse** Neural Collapse at the end of training is characterized by four coupled properties.

**NC1** (within-class collapse):

$$\Sigma_W^{(k)} = \mathrm{Avg}_i \big[ (\mu_{k,i} - \mu_k)(\mu_{k,i} - \mu_k)^\top \big] \to 0, \tag{12}$$

where $\mu_{k,i}$ is the penultimate feature of sample $i$ in class $k$ and $\mu_k$ is the class mean.

**NC2** (simplex-ETF class means):

$$\hat{\mu}_k = \frac{\mu_k - \mu_G}{\|\mu_k - \mu_G\|}, \qquad 1 \leq k \leq K, \tag{13}$$

where $\mu_G$ is the global mean. The centered and normalized means $\{\hat{\mu}_k\}$ approach the vertices of a simplex ETF, that is, their pairwise inner products converge to $-1/(K-1)$.

**NC3** (mean–prototype alignment):

$$\hat{\mu}_k = \frac{w_k}{\|w_k\|}, \qquad 1 \leq k \leq K, \tag{14}$$

namely the centered class means align with their classifier prototypes.

**NC4** (nearest-center decision):

$$\arg\max_k \langle \mu, w_k \rangle = \arg\min_k \|\mu - \mu_k\|, \tag{15}$$

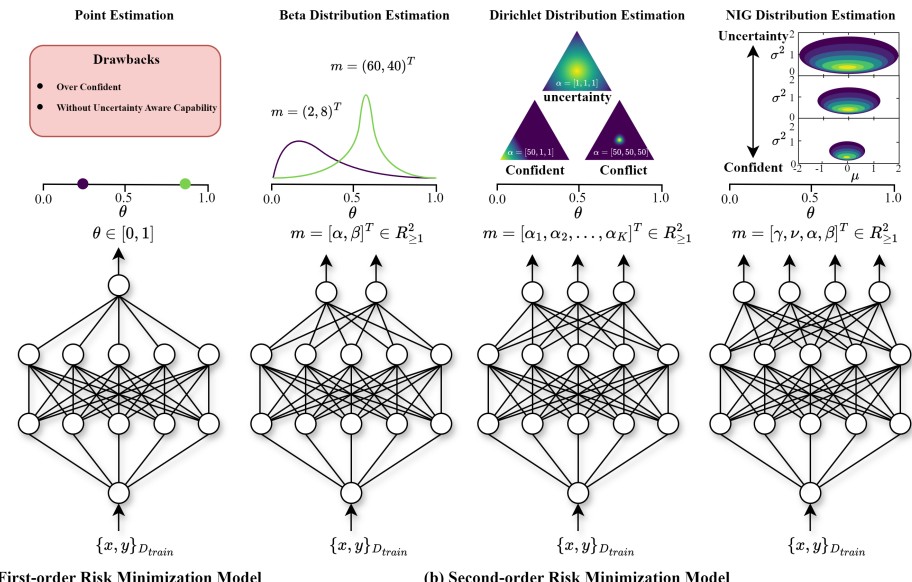

Figure 3: Overview of First-Order Risk Minimization Models and Evidential Deep Learning Based on Second-Order Risk Minimization.

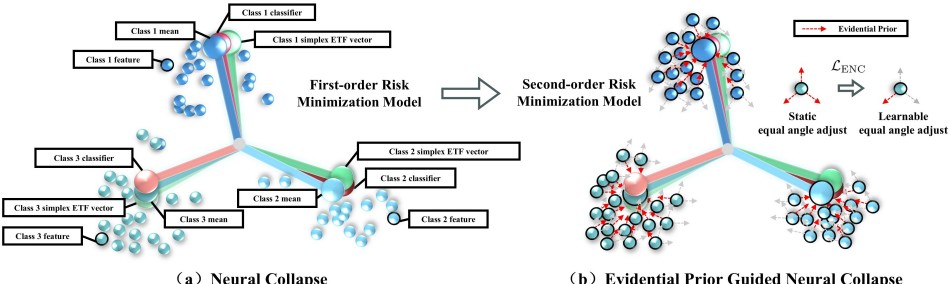

Figure 4: Overview of Evidential Prior Guided Neural Collapse. (a) Training drives final-layer features, class means, and classifier vectors towards a Simplex ETF. (b) Our Evidential Prior Guided Neural Collapse further regularises this collapse.

which holds when the equal-norm alignment in **NC2** and **NC3** is satisfied. Together, **NC1–NC4** describe an optimally discriminative geometry with minimal within-class dispersion and maximal inter-class separation.

**Neural Collapse Loss** To operationalize this geometry, Neural Collapse Loss loss fix the ID subspace to a simplex ETF via preset EBVs and optimize features to align angularly with these EBVs:

$$\mathcal{L}_{\mathrm{NC}}(x_i^{\mathrm{id}}) = -\log \frac{\exp\left(\hat{m}_i^{\mathrm{id}\top}\hat{w}_j^{\mathrm{ebv}}/\tau\right)}{\sum_{j=1}^{K}\exp\left(\hat{m}_i^{\mathrm{id}\top}\hat{w}_j^{\mathrm{ebv}}/\tau\right)}. \tag{16}$$

This objective is the discriminative counterpart of our ENC likelihood, both share the same cosine-based targets; ENC further fuses a uniform evidential prior to temper over-confidence and stabilize alignment.

### A.2 EVIDENTIAL PRIOR GUIDED NEURAL COLLAPSE

Traditional first order risk minimization fits a point estimate of $p(y \mid x)$ and directly learns the parameters $\theta$ of the data generating distribution according to the task, for example the Bernoulli parameter for binary classification, the multinomial parameter for multiclass classification, or the

Gaussian parameter for regression. This paradigm struggles to provide reliable uncertainty under distribution shift and hard examples. In contrast, second order models predict the parameters of conjugate priors, such as Beta for Bernoulli, Dirichlet for multinomial, or Normal Inverse Gamma (NIG) for Gaussian, and compute the final predictive probability or regression value from these distributional parameters. Evidential Deep Learning (EDL) (Sensoy et al., 2018; Jøsang, 2016; Sentz & Ferson, 2002) adopts the second order view. The network first predicts nonnegative evidence $e_k$ and maps it to Dirichlet parameters $\alpha_k = e_k + 1$, which defines a prior posterior family $\mathrm{Dir}(\boldsymbol{\alpha})$ over class probabilities $\boldsymbol{\pi} \in \Delta^{K-1}$. The Dirichlet parameters $\alpha_k$ can be interpreted as pseudocounts for class $k$, and the total evidence $S = \sum_k \alpha_k$ quantifies overall confidence. When $S$ is small, the model should remain unconfident. Figure 3 summarizes the differences between first order risk minimization and the second order evidential formulation.

Our ENC loss builds an explicit bridge between NC's geometric alignment and EDL's second-order evidence modeling. Rather than deriving evidence from learnable classifier activations, ENC computes cosine similarities between features and OEFS EBVs, applies the evidence activation (Eq. (4)), and injects a uniform evidence prior, forming a unified angular adjustment that is further made learnable via Bayesian inference. In joint ID+OE training, ENC concurrently regulates direction and scale, stabilizes class-wise collapse, suppresses unsupported high evidence, and prevents hard OE samples from eliciting large, non-uniform responses on the ID classifier. Figure 4 compares ENC with standard Neural Collapse.

### A.2.1 OPTIMALITY ANALYSIS

We first analyze the optimality of the ID subspace optimization. We adopt the unconstrained feature model (Papyan et al., 2020), omitting the backbone architecture to focus on the convergence properties of the final-layer features.

**Problem Setup.** Consider $N$ ID samples distributed across $K$ classes, with $n_k$ samples per class $k$. Let $\hat{W}_{\mathrm{ETF}} = [\hat{w}_1, \ldots, \hat{w}_K] \in \mathbb{R}^{d \times K}$ denote the fixed, predefined EBVs forming a Simplex ETF. The optimization objective for the feature matrix $M$ under the ENC loss is defined as:

$$
\min_{M} \quad \mathcal{L}_{\mathrm{ENC}} = \frac{1}{N} \sum_{k=1}^{K} \sum_{i=1}^{n_k} \left( \log S_i^* - \log \alpha_{k,i}^* \right),
$$

$$
\text{s.t.} \quad \|m_{k,i}\|^2 \leq 1, \quad \forall k \in \{1, \ldots, K\}, i \in \{1, \ldots, n_k\},
$$

(17)

where $e_{j,i}^* = \exp(\hat{w}_j^\top m_{k,i}/\tau)$, $\alpha_{j,i}^* = e_{j,i}^* + 1$, and $S_i^* = \sum_{j=1}^{K} \alpha_{j,i}^*$.

**Theorem 1** (Global Optimality of ENC). *For any class $k \in \{1, \ldots, K\}$ and sample $i$, the global minimizer $\hat{m}_{k,i}$ of the optimization problem in Eq. (17) satisfies $\hat{m}_{k,i} = \hat{w}_k$. This implies that minimizing $\mathcal{L}_{\mathrm{ENC}}$ strictly enforces the Neural Collapse geometry where features collapse to their corresponding class prototypes.*

*Proof.* Based on the definition of the EBVs ($\hat{W}_{\mathrm{ETF}}$), we have:

$$
\hat{w}_k^\top \hat{w}_{k'} = \frac{K}{K-1} \delta_{k,k'} - \frac{1}{K-1}, \quad \forall k, k' \in [1, K].
$$

(18)

Furthermore, $\hat{W}_{\mathrm{ETF}} \cdot \mathbf{1}_K = \mathbf{0}_d$, which implies:

$$
\sum_{j=1}^{K} \hat{w}_j = \mathbf{0}_d, \quad \sum_{j \neq k}^{K} \hat{w}_j = -\hat{w}_k.
$$

(19)

When $\hat{W}_{\mathrm{ETF}}$ is fixed, $\mathcal{L}_{\mathrm{ENC}}$ is convex with respect to $m_{k,i}$, and the constraint $\|m_{k,i}\|^2 \leq 1$ is also convex. Therefore, we can use the Kuhn-Tucker conditions (KKT) conditions to determine its global

optimality. The Lagrangian function is constructed as:

$$\mathcal{L} = \frac{1}{N} \sum_{k=1}^{K} \sum_{i=1}^{n_k} \Big[ \log\big(\sum_{j=1}^{K} \exp(\hat{w}_j^\top m_{k,i}/\tau) + K\big) $$
$$- \log\big(\exp(\hat{w}_k^\top m_{k,i}/\tau) + 1\big) \Big] \tag{20}$$
$$+ \sum_{k=1}^{K} \sum_{i=1}^{n_k} \lambda_{k,i} \left( \|m_{k,i}\|^2 - 1 \right).$$

The gradient form with respect to $m_{k,i}$ is (using the definitions of $S_i^*$ and $\alpha_{k,i}^*$):

$$\nabla_{m_{k,i}} \mathcal{L} = \frac{1}{N\tau} \left( \frac{\sum_{j=1}^{K} e_{j,i}^* \hat{w}_j}{S_i^*} - \frac{e_{k,i}^* \hat{w}_k}{\alpha_{k,i}^*} \right) + 2\lambda_{k,i} m_{k,i}. \tag{21}$$

When $\lambda_{k,i} = 0$, the gradient equation simplifies to:

$$\frac{\sum_{j=1}^{K} e_{j,i}^* \hat{w}_j}{S_i^*} = \frac{e_{k,i}^* \hat{w}_k}{\alpha_{k,i}^*}. \tag{22}$$

That is:

$$\frac{\sum_{j\neq k}^{K} e^{\hat{\mathbf{w}}_j^T m_{k,i}/\tau} \widehat{\mathbf{w}}_j}{S_i^*} = \left( \frac{1}{\alpha_{k,i}^*} - \frac{1}{S_i^*} \right) e^{\hat{w}_k^T m_{k,i}/\tau} \widehat{\mathbf{w}}_k. \tag{23}$$

Multiplying both sides by $\widehat{\mathbf{w}}_k^\top$, we obtain:

$$-\frac{1}{K-1} \frac{\sum_{j\neq k}^{K} e^{\hat{\mathbf{w}}_j^T m_{k,i}/\tau}}{S_i^*} = \left( \frac{1}{\alpha_{k,i}^*} - \frac{1}{S_i^*} \right) e^{\hat{w}_k^T m_{k,i}/\tau}. \tag{24}$$

The left-hand side is always $\leq 0$. Since $S_i^* > \alpha_{k,i}^*$, the term $\left( \frac{1}{\alpha_{k,i}^*} - \frac{1}{S_i^*} \right)$ is $> 0$, making the right-hand side $> 0$. Therefore, the condition $\nabla \mathcal{L} = 0$ cannot be satisfied when $\lambda_{k,i} = 0$.

When $\lambda_{k,i} > 0$, by KKT conditions, the global optimal solution $\widehat{\mathbf{m}}_{k,i}$ satisfies the active constraint:

$$\|\widehat{\mathbf{m}}_{k,i}\|^2 = 1. \tag{25}$$

The gradient equation $\nabla_{m_{k,i}} \mathcal{L} = 0$ is then:

$$\left( \frac{\sum_{j\neq k}^{K} e_{j,i}^* \widehat{\mathbf{w}}_j}{S_i^*} + \left( \frac{1}{S_i^*} - \frac{1}{\alpha_{k,i}^*} \right) e_{k,i}^* \widehat{\mathbf{w}}_k \right) + 2N\tau \lambda_{k,i} m_{k,i} = 0. \tag{26}$$

Multiplying both sides by $\widehat{w}_j^\top$ ($j \neq k$) gives:

$$\left( \frac{\sum_{j\neq k}^{K} e_{j,i}^*}{S_i^*} + \left( \frac{1}{\alpha_{k,i}^*} - \frac{1}{S_i^*} \right) \frac{1}{K-1} e_{k,i}^* \right) + 2N\tau \lambda_{k,i} m_{k,i}^\top \hat{w}_j = 0. \tag{27}$$

Based on the definition $S_i^* = \sum_{j=1}^{K}(e_{j,i}^*+1) = (\sum_{j=1}^{K} e_{j,i}^*)+K$, we have $\sum_{j\neq k}^{K} e_{j,i}^* = S_i^* - K - e_{k,i}^*$. Therefore:

$$\left( \frac{S_i^* - K - e_{k,i}^*}{S_i^*} + \left( \frac{1}{\alpha_{k,i}^*} - \frac{1}{S_i^*} \right) \frac{1}{K-1} e_{k,i}^* \right)$$
$$+ 2N\tau \lambda_{k,i} m_{k,i}^\top \widehat{w}_j = 0. \tag{28}$$

Multiplying the gradient equatio by $\widehat{\mathbf{w}}_k^\top$ gives:

$$\left( \frac{S_i^* - K - e_{k,i}^*}{S_i^*} + \left( \frac{1}{\alpha_{k,i}^*} - \frac{1}{S_i^*} \right) \frac{1}{K-1} e_{k,i}^* \right) \widehat{\mathbf{w}}_k$$
$$= \frac{1}{K-1} 2N\tau \lambda_{k,i} m_{k,i}. \tag{29}$$

Let $\beta = \frac{(K-1)}{2N\lambda_{k,i}} \left( \frac{S_i^* - K - e_{k,i}^*}{S_i^*} + \left( \frac{1}{\alpha_{k,i}^*} - \frac{1}{S_i^*} \right) \frac{1}{K-1} e_{k,i}^* \right)$. Since $\lambda_{k,i} > 0$ and $S_i^* > \alpha_{k,i}^* > e_{k,i}^* > 0$, we get $\beta > 0$:

$$m_{k,i} = \beta \widehat{w}_k. \tag{30}$$

Introducing the feature unit norm constraint:

$$\|m_{k,i}\| = 1. \tag{31}$$

And the ETF column vectors (EBVs) satisfy:

$$\|\widehat{w}_k\| = 1. \tag{32}$$

Substituting $\|m_{k,i}\| = 1$ into $m_{k,i} = \beta \widehat{w}_k$ gives:

$$1 = \|m_{k,i}\| = \|\beta \widehat{w}_k\| = |\beta| \, \|\widehat{w}_k\| = \beta. \tag{33}$$

Thus:

$$\beta = 1, \quad \hat{m}_{k,i} = \hat{w}_k. \tag{34}$$

This shows that the ID features are optimally aligned with the corresponding class prototypes EBVs, reaching the theoretical Neural Collapse state at the global minimum. $\qquad\square$

### A.2.2 STABILITY ANALYSIS

We explicitly analyze the gradient properties of ENC to demonstrate its robustness against interference from optimization interference arising from the massive auxiliary data. We formulate this as a bound on the sensitivity of the evidential prior mechanism.

Consider a single sample $m$ with true class $k$. We define the logits $s_j = \hat{w}_j^\top m/\tau$, the evidence $e_j^* = \exp(s_j)$, and the Dirichlet parameters $\alpha_j^* = e_j^* + 1$. The total strength is $S^* = \sum_{j=1}^K \alpha_j^*$. The per-sample $\mathcal{L}_{\text{ENC}}$ term (denoted as $\ell_{\text{ENC}}$) is defined as:

$$\ell_{\text{ENC}} = \log S^* - \log \alpha_k^*. \tag{35}$$

We formally state the smoothing mechanism in the following proposition.

**Proposition 1** (Evidential Prior Smoothing). *The evidential prior mechanism imposes a strict bound on the gradient sensitivity. Specifically, the magnitude of the gradient of the log Dirichlet parameter ratio $R_{k,j}$ with respect to the evidence $e_k^*$ is strictly bounded by 1, acting as a geometric rate-limiter.*

*Proof.* We prove this mechanism by analyzing the partial derivatives. First, the partial derivative of the loss $\ell_{\text{ENC}}$ with respect to the evidence $e_j^*$ is derived as:

$$\frac{\partial \ell_{\text{ENC}}}{\partial e_j^*} = \begin{cases} \frac{1}{S^*}, & j \neq k, \\ \frac{1}{S^*} - \frac{1}{\alpha_k^*}, & j = k. \end{cases} \tag{36}$$

Next, to quantify the angular separation stability, for any $j \neq k$, we define the log Dirichlet parameter ratio:

$$R_{k,j} = \log \frac{\alpha_k^*}{\alpha_j^*} = \log \frac{e^{s_k} + 1}{e^{s_j} + 1}. \tag{37}$$

We examine how this ratio changes with respect to the target evidence. Because $R_{k,j}$ depends on $e_k^*$ only through $\alpha_k^*$, the partial derivative is:

$$\frac{\partial R_{k,j}}{\partial e_k^*} = \frac{\partial}{\partial e_k^*} \left( \log \alpha_k^* - \log \alpha_j^* \right) = \frac{1}{\alpha_k^*} = \frac{1}{e_k^* + 1}. \tag{38}$$

Since the evidence is non-negative ($e_k^* \geq 0$), we have $\alpha_k^* \geq 1$. Consequently:

$$\frac{\partial R_{k,j}}{\partial e_k^*} \leq 1. \tag{39}$$

This bound ($\leq 1$) represents the smoothing effect. It proves that the gradient step's effect on the angular separation ($R_{k,j}$) is strictly limited. This prevents uncontrolled, rapidly diverging attraction of $m$ toward an EBV, rendering the alignment process resilient to optimization interference induced by massive OE samples. $\qquad\square$

A.3 THEORETICAL JUSTIFICATION OF ORTHOGONALITY CONSTRAINT

In Section 3.4, we introduced the Orthogonality Constraint loss $\mathcal{L}_{OC}$ to regulate the alignment of OE features. Formally, this loss minimizes the KL divergence between the predicted probability distribution $\mathbf{p}^{oe \to ebv}$ and the uniform distribution. In this section, we provide a rigorous proof demonstrating that, governed by the zero-sum property of the Simplex Equiangular Tight Frame (ETF), achieving the global minimum of this loss is mathematically equivalent to imposing a hard orthogonality constraint between the OE feature and all ID prototypes.

**Proposition 2** (Geometric Equivalence of Uniformity). *Let the ID prototypes $\{\hat{w}_j^{ebv}\}_{j=1}^{K}$ form a centered Simplex ETF in $\mathbb{R}^d$. An OE feature vector $\hat{m}^{oe} \in \mathbb{R}^d$ yields a exactly uniform probability distribution $\mathbf{p}^{oe \to ebv}$ over these prototypes if and only if it is orthogonal to every prototype, i.e., $\hat{m}^{oe\top}\hat{w}_j^{ebv} = 0, \forall j \in \{1, \ldots, K\}$.*

*Proof.* Let $\hat{m}^{oe}$ denote the normalized feature vector of an OE sample, and $\{\hat{w}_j^{ebv}\}_{j=1}^{K}$ be the set of normalized Equiangular Basic Vectors (EBVs) representing the $K$ ID classes. By definition, these EBVs constitute a Simplex ETF, which inherently satisfies the structural zero-sum property:

$$\sum_{j=1}^{K} \hat{w}_j^{ebv} = \mathbf{0}_d. \tag{40}$$

The objective of $\mathcal{L}_{OC}$ is to induce a uniform probability distribution over the $K$ classes. The probability for the $j$-th class, denoted as $(\mathbf{p}^{oe \to ebv})_j$, is derived via the Softmax function applied to the scaled cosine similarities. To achieve a exactly uniform distribution (i.e., $(\mathbf{p}^{oe \to ebv})_j = 1/K, \forall j$), the logits (cosine similarities) must be invariant across all classes. Let $c \in \mathbb{R}$ denote this constant scalar:

$$\hat{m}^{oe\top}\hat{w}_j^{ebv} = c, \quad \forall j \in \{1, \ldots, K\}. \tag{41}$$

We establish the proof by evaluating the aggregate projection of $\hat{m}^{oe}$ onto the set of ID prototypes, denoted as $\mathcal{S} = \sum_{j=1}^{K}(\hat{m}^{oe\top}\hat{w}_j^{ebv})$.

Imposing the uniformity condition from Eq. (41), the summation algebraically becomes:

$$\mathcal{S} = \sum_{j=1}^{K} c = K \cdot c. \tag{42}$$

Conversely, invoking the linearity of the inner product and the ETF zero-sum property (Eq. 40), the summation must geometrically satisfy:

$$\mathcal{S} = \hat{m}^{oe\top}\left(\sum_{j=1}^{K} \hat{w}_j^{ebv}\right) = \hat{m}^{oe\top} \cdot \mathbf{0}_d = 0. \tag{43}$$

Equating Eq. (42) and Eq. (43) yields $K \cdot c = 0$. Since $K > 0$, it follows that $c = 0$. Substituting $c = 0$ back into Eq. (41), we arrive at the necessary condition for optimality:

$$\hat{m}^{oe\top}\hat{w}_j^{ebv} = 0, \quad \forall j \in \{1, \ldots, K\}. \tag{44}$$

This concludes the proof that within the OEFS geometry, the only state in which an OE feature can minimize $\mathcal{L}_{OC}$ is when it lies in the null space of the ID prototypes. Thus, the uniformity loss effectively functions as a geometric orthogonality constraint. □

A.4 FROM LEARNABLE CLASSIFIERS TO VAST PREDEFINED CLASSIFIERS

To elucidate the design motivations and advantages of the VPC framework, we review the architectural evolution of OOD detection paradigms (Figure 5). Vanilla DNN-based OOD detection (Figure 5a) employs a learnable ID classifier optimized via first-order logits. Training with $\mathcal{L}_{OE}$ enforces output uniformity, while detection relies on post-hoc confidence-based measures like MSP. Limitations include: (1) representation drift from optimization interference of OE data during joint ID/OE optimization; (2) non-uniform OOD probabilities due to ID-centric classification. PFS exploits Neural Collapse (ETF formation between ID features and classifiers) to orthogonalize OOD features

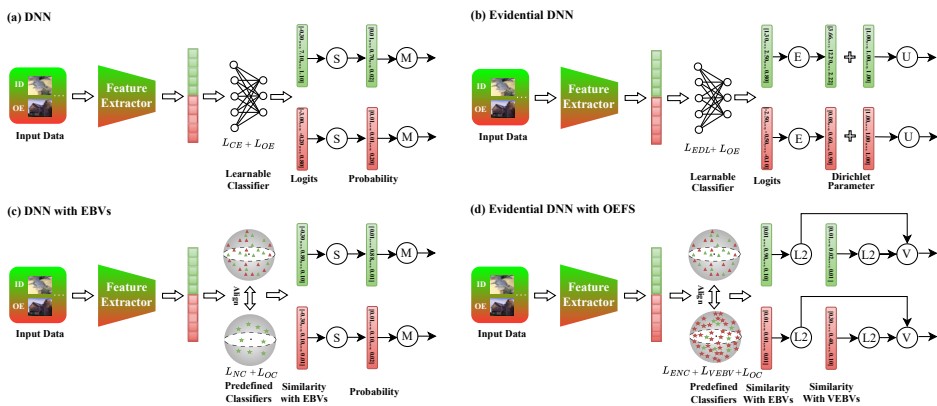

Figure 5: The architectural evolution from vanilla DNNs to our VPC. (a) Vanilla DNN: relies on learnable classifiers and first-order logits, using Softmax ($S$) and Maximum Softmax Probability ($M$) for OOD detection under $\mathcal{L}_{\text{CE}} + \mathcal{L}_{\text{OE}}$. (b) Evidential DNN: introduces second-order uncertainty ($U$) based on geometric evidence ($E$) and Dirichlet priors. (c) DNN with EBVs: aligns features with predefined Equiangular Basic Vectors (EBVs) to achieve Neural Collapse. (d) Proposed VPC: constructs an Orthogonal Equiangular Feature Space (OEFS) where ID and OE features are jointly aligned with EBVs and Vast EBVs (VEBVs), respectively. The $L_2$ symbol denotes activation intensity in each subspace, and $V$ represents the final discriminative VPC Score.

against ID weights, yet the classifier's limited capacity hinders adaptation to diverse OOD patterns. Evidential DNNs (Figure 5b) substitute logits with second-order evidence, mitigating over-confidence and enabling explicit OOD uncertainty quantification, thus avoiding non-uniform outputs. The uniform evidence prior reduces drift in ID/OE optimization but fails to ensure low OOD evidence activation via the ID classifier. EBV-based DNNs (Figure 5c) replace learnable classifiers with fixed, predefined EBVs Shen et al. (2025), optimizing Neural Collapse's ETF prior via first-order $\mathcal{L}_{NC}$ to align ID features with EBVs. This stabilizes ID representations but remains ID-centric.

VPC (Figure 5d) integrates these advances: adopting EBV priors from (c), incorporating second-order evidence from (b) via $\mathcal{L}_{ENC}$ (replacing $\mathcal{L}_{NC}$), which ensures stable alignment of ID features with EBVs under OOD perturbations. VPC extends to a bilateral framework with predefined VEBVs, creating an orthogonal subspace provided with distinct geometric anchors to resolve diverse OOD patterns. $\mathcal{L}_{OC}$ and $\mathcal{L}_{VEBV}$ guide OOD activation therein while minimizing ID interference.

We next compare VPC's parameter and computational overhead against learnable classifier DNNs. Parameter-wise, Vanilla DNNs (Figure 5a, b) incur $O(d_{\text{feat}} \times K)$ for the final classifier. VPC eliminates this, adding a projection layer $O(d_{\text{feat}} \times d_{\text{oefs}})$ to meet dimensional constraints Shen et al. (2025); Papyan et al. (2020). The $(K + V)$ EBVs/VEBVs are fixed constants, excluding them from trainable parameters.

As quantified in Table 8, this change is modest. For instance, on CIFAR-100 ($K = 100$) with a ResNet-18 ($d_{\text{feat}} = 512$), VPC replaces the $\approx 0.051M$ parameter classifier (the vanilla head) with a $\approx 0.524M$ parameter projection layer. This results in a net increase of only $\approx 0.47M$ parameters, which constitutes just $\approx 4.2\%$ of the $11.2M$ total model parameters. This modest overhead is the necessary cost to instantiate the structured geometric space. We provide fair comparisons by adding this same projection layer to baselines in Appendix B.3.

Computationally, VPC decouples the loss functions: $\mathcal{L}_{\text{ENC}}$ and $\mathcal{L}_{\text{OC}}$ are computed in the low-dimensional $K$-space, while $\mathcal{L}_{\text{VEBV}}$ operates in the $V$-space. The gradients for $\mathcal{L}_{\text{VEBV}}$ are diagonal, leading to a backpropagation complexity of $O(V)$. In contrast, naive frameworks with $(K + V)$-dimensional outputs require dense softmax Jacobians, resulting in a computational cost that scales as $O(K + V)^2$, which becomes prohibitive for large $V$.

At inference time, VPC is equally efficient. Our VPC score computes similarities against all prototypes (EBVs and VEBVs) at once by merging them into a single weight matrix and performing one unified matrix-vector multiplication ($O(d_{\text{oefs}} \times (K + V))$). By comparison, baseline methods like MSP/Uncertainty perform a smaller $O(d_{\text{feat}} \times K)$ multiplication, but due to the parallel nature of GPUs, the practical wall-clock time difference between the two is negligible. This efficiency stands in

Table 8: Comparison of learnable parameter overhead. All units are in Millions (M). $d_{\text{feat}}$ is the penultimate feature dimension. The projection dimension $d_{\text{oefs}}$ for CIFAR is 1024; for ImageNet-1k, it is set to 2048 per analysis.

| Dataset | Model | Total (M) | $d_{\text{feat}}$ | Vanilla Head (M) | $d_{\text{oefs}}$ | VPC Proj. (M) | Added Params (M) |
|---------|-------|-----------|-------|------------------|-------|---------------|------------------|
| CIFAR-10 | WRN-40-2 | 2.20 | 128 | 0.00129 | 1024 | 0.13107 | +0.13 |
| | ResNet-18 | 11.20 | 512 | 0.00513 | 1024 | 0.52429 | +0.52 |
| | DenseNet-121 | 8.00 | 1024 | 0.01025 | 1024 | 1.04858 | +1.04 |
| CIFAR-100 | WRN-40-2 | 2.20 | 128 | 0.01290 | 1024 | 0.13107 | +0.12 |
| | ResNet-18 | 11.20 | 512 | 0.05130 | 1024 | 0.52429 | +0.47 |
| | DenseNet-121 | 8.00 | 1024 | 0.10250 | 1024 | 1.04858 | +0.95 |
| ImageNet-1k | ResNet-50 | 25.56 | 2048 | 2.049 | 2048 | 4.194 | +2.15 |
| | ViT-B-16 | 86.56 | 768 | 0.769 | 2048 | 1.573 | +0.80 |

sharp contrast to neighborhood-based methods like KNN (Sun et al., 2022), which require computing distances against the entire training set ($N$ samples) at inference time ($O(N \times d_{\text{feat}})$) and are thus prohibitively slow on large datasets. Therefore, VPC exchanges a modest, linear computational overhead for the significant performance benefit of explicitly modeling the OOD space, a capability the baselines lack.

## A.5 Why does the VPC Score yield additional discriminative power?

Prior methods are confined to ID-class features and the learnable ID-class classifier, building various scoring functions on top of that setup. Our approach steps outside this limitation. To our knowledge, we are the first to explicitly expand a large preset classifier for OOD features. We analyze the effect through cosine similarity, Euclidean distance, and L2 activation strength.

Note that all our metrics are computed between normalized features and the preset EBVs/VEBVs. By contrast, the metrics for PFS and OE are computed between normalized features and the normalized classifier weights of the ID classes. For Euclidean distance and cosine similarity, we apply a unified MSP (max over classes) reduction and then average the per-sample values. The final aggregated results under different settings are reported in Table 9. If evaluation is restricted to metrics based solely on the ID classifiers or EBVs, our gains are comparable to those of the state-of-the-art PFS, as both methods impose orthogonality on OOD features, while the latter emphasizes uniform output. The additional benefit of our approach is the use of predefined VEBVs to generate supplementary L2 activation differences, which constitutes a distinctive advantage over competing methods.

In addition to the above findings, Table 10 provides a comprehensive comparison of the VPC Score under L1 (Manhattan) and L2 (Euclidean) norms, ablating the influence of the EBV subspace coefficient ($\alpha$) and the VEBV subspace coefficient ($\beta$). The results first indicate that the L2 norm generally yields superior or more stable performance than the L1 norm. This superiority likely arises because the L2 norm, which squares activations, more faithfully captures the variations in activation magnitude that are highly informative for complex OOD samples. Under the L1 norm, performance is highly sensitive to the coefficient choice. Notably, relying solely on the VEBV subspace ($\alpha = 0, \beta = 100$) consistently and significantly outperforms relying only on the EBV subspace ($\alpha = -1, \beta = 0$), especially on the more complex CIFAR-100 dataset. This demonstrates the critical role of the VEBV subspace in capturing OOD variability.

The most striking finding, however, comes from the L2 norm results. Performance is remarkably stable and remains near-optimal across all tested combinations of $\alpha$ and $\beta$. Whether using only the ID-EBV subspace ($\alpha = -1, \beta = 0$), only the OOD-VEBV subspace ($\alpha = 0, \beta = 100$), or our main combined score ($\alpha = -1, \beta = 100$), the detection metrics are nearly indistinguishable. This strongly implies that our OEFS training successfully creates a geometric separation where the L2 activation strength in either subspace alone becomes a sufficient and robust signal for OOD detection, a capability rarely observed in previous methods.

Table 9: Statistical results for ID/OOD under different metrics on CIFAR-10, cosine similarity, Euclidean distance, and L2 activation strength. Our method obtains additional discriminative scores by using the L2 activation strength within the subspaces spanned by EBVs and VEBVs. We report ID/OOD averages and the gap (Diff↑).

| Method | Euclidean Distance | | | Cosine Similarity | | | L2 (logits/weight/EBVs) | | | L2 (VEBVs) | | |
|---|---|---|---|---|---|---|---|---|---|---|---|---|
| | ID | OOD | Diff↑ | ID | OOD | Diff↑ | ID | OOD | Diff↑ | ID | OOD | Diff↑ |
| OE | 0.62 | 1.08 | 0.46 | 0.80 | 0.40 | 0.40 | 0.87 | 0.55 | 0.32 | – | – | – |
| PFS | 0.67 | 1.38 | **0.71** | 0.75 | 0.05 | **0.70** | 0.87 | 0.06 | **0.82** | – | – | – |
| Ours: VPC | 0.70 | 1.39 | 0.69 | 0.73 | 0.03 | **0.70** | 0.77 | 0.04 | 0.72 | 0.63 | 1.03 | **0.40** |

Table 10: Norm-aware comparison of VPC scores on CIFAR-10 and CIFAR-100 across backbones (WideResNet-40-2, ResNet-18, DenseNet-121). Metrics are reported as FPR95↓/AUROC↑/AUPR↑.

| Norm | VPC Score | | CIFAR-10 | | | CIFAR-100 | | |
|---|---|---|---|---|---|---|---|---|
| | $\alpha$ | $\beta$ | WRN-40-2 | ResNet-18 | DenseNet-121 | WRN-40-2 | ResNet-18 | DenseNet-121 |
| L1 | -1 | 0 | 2.41/99.04/99.71 | 2.95/**98.24**/99.62 | **2.01/98.81/99.70** | 39.57/88.01/96.39 | 40.01/89.26/97.03 | 38.63/89.08/96.84 |
| | 0 | 100 | 2.29/99.00/99.78 | **2.86**/97.29/99.50 | 2.10/96.55/99.38 | 32.43/**93.52/98.52** | 38.71/92.08/**98.21** | **31.01/94.00/98.70** |
| | -1 | 1 | **2.25/99.10/99.79** | 2.88/98.16/**99.64** | 2.08/98.34/99.68 | 38.60/91.12/97.59 | **38.39/92.10**/98.20 | 32.41/93.78/98.65 |
| | -1 | 10 | 2.28/99.02/99.78 | 2.87/97.58/99.54 | 2.10/96.84/99.43 | 34.90/93.26/98.46 | 38.48/92.09/**98.21** | 31.22/93.98/**98.70** |
| | -1 | 100 | 2.29/99.00/99.78 | **2.86**/97.32/99.50 | 2.10/96.58/99.38 | **32.40**/93.50/**98.52** | 38.74/92.08/**98.21** | 31.03/**94.00/98.70** |
| L2 | -1 | 0 | **2.27/99.19/99.81** | 2.85/**98.50/99.70** | **2.08/98.95/99.78** | 32.04/**93.65/98.55** | 39.10/**92.03/98.20** | 31.19/**94.01/98.71** |
| | 0 | 100 | **2.27/99.19/99.81** | 2.84/98.32/99.67 | 2.10/98.70/99.74 | **32.03/93.65/98.53** | 38.98/**92.03/98.20** | **31.17/94.01/98.71** |
| | -1 | 1 | **2.27/99.19/99.81** | 2.85/**98.50/99.70** | 2.10/**98.95/99.78** | 32.07/**93.65**/98.53 | 39.11/**92.03/98.20** | 31.19/**94.01/98.71** |
| | -1 | 10 | **2.27**/99.18/**99.81** | 2.84/98.49/**99.70** | 2.10/98.94/**99.78** | 32.05/**93.65**/98.53 | 38.98/**92.03/98.20** | **31.17/94.01/98.71** |
| | -1 | 100 | **2.27**/99.18/**99.81** | 2.84/98.32/99.67 | 2.10/98.86/99.77 | 32.04/**93.65**/98.53 | 38.96/**92.03/98.20** | **31.17/94.01/98.71** |

## A.6 ACTIVATION STRENGTH IS A BETTER OOD SCORING FUNCTION

Building on the previous section, we introduce a new L2 activation-strength score for both PFS and OE. Specifically, for PFS the activation matrix is the cosine similarity between normalized features and the normalized classifier weights; for OE we directly use the model's activation outputs. As shown in Table 11, incorporating the L2 strength yields substantial improvements, further validating the benefit of class-agnostic score design for OOD detection.

Prior approaches typically rely on constraining OOD features to be orthogonal to the ID classifier or enforcing near-uniform outputs. However, MSP tends to exaggerate the non-uniform outputs of hard samples, often leading to inseparable cases. In contrast, computing the overall activation strength (L2) naturally benefits from both orthogonality and uniformity regularization, producing a more discriminative score.

Table 11: L2 Activation Strength Driven Improvements in OOD Detection

| Score Function | Method | SVHN | | LSUN | | Far-OOD Datasets iSUN | | Textures | | Places365 | | Average | |
|---|---|---|---|---|---|---|---|---|---|---|---|---|---|
| | | FPR95↓ | AUROC↑ | FPR95↓ | AUROC↑ | FPR95↓ | AUROC↑ | FPR95↓ | AUROC↑ | FPR95↓ | AUROC↑ | FPR95↓ | AUROC↑ |
| CIFAR-10 | | | | | | | | | | | | | |
| MSP | OE | 1.95 | 99.23 | 0.80 | 99.67 | 1.95 | 99.36 | 3.70 | 99.23 | 8.80 | 97.76 | 3.44 | 99.05 |
| PFS Score | PFS | 0.75 | 98.95 | 0.35 | 99.51 | 1.30 | 99.26 | 2.85 | 98.49 | 7.25 | 97.02 | 2.50 | 98.65 |
| L2 Score | OE | 1.45 | 99.26 | 0.35 | 99.70 | 0.90 | 99.43 | 2.75 | 99.26 | 7.20 | 97.90 | 2.53 (↓ 0.91) | **99.11** (↑ 0.06) |
| | PFS | 0.70 | 98.58 | 0.35 | 99.43 | 1.30 | 99.23 | 2.90 | 98.26 | 7.10 | 96.91 | **2.47** (↓ 0.03) | 98.48 (↓ 0.17) |
| CIFAR-100 | | | | | | | | | | | | | |
| MSP | OE | 34.95 | 93.75 | 14.90 | 97.23 | 49.50 | 88.16 | 43.35 | 90.63 | 52.50 | 87.68 | 39.04 | 91.49 |
| PFS Score | PFS | 24.75 | 95.81 | 12.65 | 97.78 | 38.40 | 91.44 | 44.20 | 91.32 | 51.85 | 90.33 | 34.37 | 93.33 |
| L2 Score | OE | 28.60 | 94.97 | 10.10 | 98.08 | 53.50 | 88.03 | 43.05 | 90.45 | 49.40 | 90.64 | 36.93 (↓ 2.11) | 92.43 (↑ 0.94) |
| | PFS | 25.00 | 95.78 | 13.80 | 97.69 | 36.60 | 91.92 | 42.50 | 91.35 | 52.25 | 90.27 | **34.03** (↓ 0.34) | **93.40** (↑ 0.07) |

## A.7 CONVERGENCE VS. DIVERGENCE OF OOD FEATURES

PFS is the first to leverage the properties of Neural Collapse to explore the subspace spanned by ID-class classifier weights as a way to represent richly varying OOD features. However, it has several limitations. It aligns ID features with ID classifiers in stages and then imposes an orthogonality constraint that encourages OOD features to explore the representation space formed by the ID classifiers. This staged scheme cannot guarantee a stable representation space while ID/OE data are being jointly fine-tuned: ID features and ID classifiers may not be well aligned, and OOD outputs may

remain non-uniform. In such a dynamically changing and unstable space, enforcing orthogonality on OOD features is unlikely to deliver the desired effect.

Our method effectively addresses this issue by directly presetting an approximately orthogonal, optimal classifier. It requires no staged constraints and can even achieve strong ID/OOD separability with a single training run. Building on this well-behaved geometric constraint, we investigate whether OOD features should diverge or converge. Beyond the divergence loss proposed in the main text, which induces OOD features to explore the VEBV subspace, we additionally propose a VEBV convergence loss that encourages OOD features to collapse onto any one VEBV, defined as follows:

$$\mathcal{L}_{\text{VEBV}}^{\text{con}}(x_i^{\text{oe}}) = 1 - \max_{1 \le j \le V} \hat{m}_i^{\text{oe}\top} \hat{w}_j^{\text{vebv}}. \tag{45}$$

All other settings follow the main experiments. As shown in Table 12, under our OEFS space constraint, even forcing OOD features to collapse onto a single VEBV yields reasonably good OOD detection, though the performance weakens as the number of ID classes increases. This is because when all OOD features collapse onto one VEBV, maintaining orthogonality to all ID classes becomes increasingly difficult.

Table 12: Divergence vs. Convergence Loss on the VEBVs Subspace.

| Optimization Loss | SVHN | | LSUN | | iSUN | | Textures | | Places365 | | Average | |
|---|---|---|---|---|---|---|---|---|---|---|---|---|
| | FPR95↓ | AUROC↑ | FPR95↓ | AUROC↑ | FPR95↓ | AUROC↑ | FPR95↓ | AUROC↑ | FPR95↓ | AUROC↑ | FPR95↓ | AUROC↑ |
| **CIFAR-10** | | | | | | | | | | | | |
| $\mathcal{L}_{\text{VEBV}}^{con}$ | 1.45 | 99.45 | 0.65 | 99.53 | 0.90 | 99.46 | 7.00 | 97.87 | 2.15 | 99.30 | 2.43 | 99.12 |
| $\mathcal{L}_{\text{VEBV}}^{div}$ | 0.85 | 99.62 | 0.45 | 99.50 | 1.10 | 99.50 | 2.25 | 99.38 | 6.70 | 97.91 | **2.27** | **99.18** |
| **CIFAR-100** | | | | | | | | | | | | |
| $\mathcal{L}_{\text{VEBV}}^{con}$ | 24.10 | 95.89 | 18.60 | 96.79 | 55.35 | 88.01 | 52.90 | 89.70 | 39.40 | 91.96 | 38.07 | 92.47 |
| $\mathcal{L}_{\text{VEBV}}^{div}$ | 9.95 | 97.98 | 26.25 | 95.75 | 26.50 | 94.97 | 45.05 | 89.67 | 52.45 | 89.88 | **32.04** | **93.65** |

We further provide visualizations that clearly depict the starkly different activation patterns produced by the two optimization regimes. As shown in Figure 6 and Figure 7, see Section B.4.1 for detailed analysis.

# B   ADDITIONAL EXPERIMENTAL RESULTS AND VISUALIZATION

## B.1   HARD OOD DETECTION

We further assess the generalization of VPC in a Hard OOD setting. We follow (Tack et al., 2020; Sun et al., 2022; Wang et al., 2024a) and use CIFAR-10 as ID while evaluating on LSUN-Fix, ImageNet-Resize, CIFAR-100, and Tiny-ImageNet, comparing VPC with OE, Energy-OE, DAL, and PFS (13). Across different architectures, VPC consistently improves OOD detection. Notably, the mean FPR95 on WideResNet-40-2 and DenseNet-121 is 11.97% and 10.65% (lowest among baselines), and DenseNet-121 achieves the best mean AUROC = 97.55%, with leading subset performance on LSUN-Fix (99.52%) and ImageNet-Resize (99.35%). These outcomes are attributable to VPC's explicit OEFS design and enforced geometric separation, which jointly curtail false positives and preserve discriminative power in near-OOD regimes.

## B.2   MORE RESULTS OF ONE-STAGE TRAINING AND TWO-STAGE TRAINING

Most existing OOD detection methods follow a two-stage training paradigm: pretrain on ID data, then jointly fine-tune on ID and OOD (OE) data. Although this paradigm appears closer to real-world deployment, in essence it relies on a large OE distribution that is mismatched with both the ID distribution and the target test OOD distribution to cue the model to output discriminatively low-confidence predictions whenever inputs deviate from ID. Prior work has leveraged auxiliary OE datasets at tens-of-millions scale to push models away from non-ID samples, spurring a line of subsequent research. However, this approach entails substantial engineering complexity and reproducibility cost, and high sensitivity to numerous hyperparameters (e.g., OE ratio, loss weights, thresholding, staged learning rates/schedules) and dependence on OE data selection bias and coverage.

Table 13: Hard OOD detection on CIFAR-10 benchmark.

| Model | Method | Near-OOD Datasets | | | | | | | | Average | |
| | | LSUN-Fix | | ImageNet-Resize | | CIFAR-100 | | Tiny-ImageNet | | | |
| | | FPR95↓ | AUROC↑ | FPR95↓ | AUROC↑ | FPR95↓ | AUROC↑ | FPR95↓ | AUROC↑ | FPR95↓ | AUROC↑ |
| CIFAR-10 | | | | | | | | | | | |
| With auxiliary OOD data | | | | | | | | | | | |
| WRNet-40-2 | OE | 1.10 | 99.49 | 7.10 | 98.48 | 24.80 | 94.74 | 18.15 | 95.53 | 12.79 | **97.06** |
| | Energy-OE | 2.15 | 99.11 | 8.75 | 97.34 | 32.50 | 91.57 | 21.75 | 94.03 | 16.29 | 95.51 |
| | DAL | 1.25 | 99.41 | 4.55 | 98.33 | 27.00 | 93.94 | 19.40 | 95.08 | 13.05 | 96.69 |
| | PFS | 0.70 | 98.59 | 5.35 | 98.20 | 25.15 | 88.88 | 17.45 | 92.41 | 12.16 | 94.52 |
| | **Ours: VPC** | 1.20 | 99.46 | 5.40 | 98.59 | 25.25 | 94.24 | 16.05 | 95.59 | **11.97** | 96.97 |
| ResNet-18 | OE | 1.45 | 99.26 | 4.45 | 98.29 | 24.30 | 94.78 | 17.70 | 95.61 | **11.97** | 96.99 |
| | Energy-OE | 1.55 | 98.08 | 9.05 | 97.96 | 30.35 | 92.44 | 21.10 | 94.64 | 15.51 | 95.78 |
| | DAL | 1.10 | 99.08 | 5.65 | 98.43 | 25.20 | 95.07 | 16.95 | 95.90 | 12.22 | 97.12 |
| | PFS | 1.08 | 99.09 | 5.62 | 98.44 | 25.15 | 95.09 | 16.93 | 95.92 | 12.20 | 97.14 |
| | **Ours: VPC** | 1.20 | 99.45 | 6.40 | 98.40 | 23.65 | 94.72 | 17.10 | 95.95 | 12.09 | **97.15** |
| DenseNet-121 | OE | 1.35 | 99.31 | 1.85 | 98.99 | 24.30 | 94.75 | 15.65 | 95.95 | 10.79 | 97.25 |
| | Energy-OE | 1.60 | 98.95 | 7.25 | 98.29 | 31.70 | 92.05 | 21.95 | 93.94 | 15.62 | 95.81 |
| | DAL | 0.70 | 98.88 | 2.35 | 99.06 | 25.65 | 94.13 | 16.75 | 95.46 | 11.36 | 96.88 |
| | PFS | 0.95 | 99.21 | 3.55 | 98.47 | 25.15 | 92.21 | 16.75 | 94.02 | 11.60 | 95.98 |
| | **Ours: VPC** | 0.90 | 99.52 | 2.30 | 99.35 | 23.90 | 95.08 | 15.50 | 96.26 | **10.65** | **97.55** |

To address these issues, we propose a one-stage joint training scheme that uses ID and OOD data within a single optimization process under a unified objective, thereby markedly reducing hyperparameter burden and training overhead. In the main text we systematically compare two-stage and one-stage training: in most settings, the one-stage approach achieves superior or comparable OOD performance. Moreover, Table 14 reports ID classification accuracy (Acc), showing that one-stage training does not incur a significant drop in ID accuracy, further corroborating its effectiveness and practical deployability in improving OOD detection robustness while preserving ID performance.

Table 14: one-stage training vs. two-stage training on CIFAR-10 with WideResNet-40-2. We report **average** FPR95↓, AUROC↑, AUPR↑, and ID Acc↑.

| Method | one-stage (ID + OE 150 epoch) | | | | two-stage (ID 200 epoch + OE 50 epoch) | | | |
| | FPR95↓ | AUROC↑ | AUPR↑ | Acc↑ | FPR95↓ | AUROC↑ | AUPR↑ | Acc↑ |
| OE | 2.74 | 99.01 | 99.79 | **95.39** | 3.44 | 99.05 | 99.79 | 95.67 |
| Energy-OE | 2.29 | 98.79 | 99.72 | 93.55 | 3.75 | 98.66 | 99.69 | 90.85 |
| DAL | 2.95 | 98.88 | 99.75 | 94.99 | 3.17 | 98.84 | 99.74 | 94.96 |
| PFS | 2.44 | 98.87 | 99.68 | 94.68 | 2.68 | 98.66 | 99.65 | 94.65 |
| **Ours: VPC** | **2.01** | **99.19** | **99.82** | 95.32 | **2.27** | **99.18** | **99.81** | **95.74** |

## B.3 FAIRNESS EVALUATION UNDER OEFS DIMENSIONAL CONSTRAINTS

OEFS requires scaling to a large bank of predefined classifiers (VEBVs) to sufficiently capture OOD variability. Since the number of EBVs is limited by dimensionality, the OEFS dimensionality must be greater than the combined count of EBVs and VEBVs. We therefore replace the final linear head in WideResNet-40-2, ResNet-18, and DenseNet-121 with a learnable projector that lifts features to a higher-dimensional space. For OEFS sizes 500, 1000, 2000, the projector outputs 512, 1024, 2048; its input equals each backbone's penultimate feature dimension. This design increases trainable parameters. To maintain fairness, we mirror the same projector in all compared baselines and also report the original backbones' detailed results (See Table 15 and Table 16 ).

Results reveal that projector-augmented baselines despite having strictly more trainable parameters than ours exhibit inconsistent performance across methods/backbones. We hypothesize that the extra dimensional transform exacerbates classifier learning, especially for PFS whose objective hinges on enforcing weight orthogonality, thereby amplifying sensitivity to optimization and data idiosyncrasies. Our approach circumvents this fragility: classifiers are pre-instantiated as optimal prototypes and remain fixed; training focuses solely on aligning features to these prototypes. This avoids the instability induced by learning classifier weights after projection and, consistently, delivers superior OOD detection. The use of a projector is principled in VPC: it provides the necessary ambient dimensionality to host a large set of equiangular prototypes without compromising ID geometry.

Table 15: Fair Comparison by adding Unified Projection layer with different architectures.

| Model | Method | SVHN | | LSUN | | iSUN (Far-OOD Datasets) | | Textures | | Places365 | | Average | |
|---|---|---|---|---|---|---|---|---|---|---|---|---|---|
| | | FPR95 | AUROC | FPR95 | AUROC | FPR95 | AUROC | FPR95 | AUROC | FPR95 | AUROC | FPR95 | AUROC |
| **CIFAR-10** | | | | | | | | | | | | | |
| *With auxiliary OOD data* | | | | | | | | | | | | | |
| WideResNet-40-2 + Projection Layer | OE | 0.70 | 99.71 | 0.80 | 99.64 | 3.60 | 99.03 | 9.30 | 97.65 | 2.60 | 99.25 | 3.40 | 99.06 |
| | Energy-OE | 0.60 | 99.58 | 0.60 | 99.10 | 4.30 | 98.83 | 7.10 | 97.85 | 2.20 | 99.16 | 2.96 | 98.90 |
| | DAL | 0.90 | 99.56 | 0.50 | 99.50 | 3.05 | 98.98 | 2.85 | 99.21 | 8.30 | 97.58 | 3.12 | 98.97 |
| | PFS | 0.75 | 99.48 | 0.70 | 99.42 | 3.40 | 99.03 | 8.00 | 99.03 | 2.55 | 98.74 | 3.08 | 98.70 |
| | **Ours: VPC** | 0.85 | 99.62 | 0.45 | 99.50 | 1.10 | 99.50 | 2.25 | 99.38 | 6.70 | 97.91 | **2.27** | **99.18** |
| ResNet-18 + Projection Layer | OE | 1.35 | 99.14 | 1.50 | 98.93 | 6.80 | 98.29 | 10.55 | 97.30 | 2.60 | 98.77 | 4.56 | **98.48** |
| | Energy-OE | 0.65 | 99.18 | 2.30 | 97.90 | 5.80 | 98.20 | 9.40 | 97.38 | 2.75 | 98.80 | 4.18 | 98.29 |
| | DAL | 0.50 | 99.02 | 1.20 | 98.92 | 3.95 | 98.53 | 2.60 | 98.56 | 9.65 | 97.08 | 3.58 | 98.42 |
| | PFS | 0.85 | 98.30 | 1.10 | 98.60 | 6.75 | 97.54 | 8.00 | 96.78 | 3.35 | 97.72 | 4.01 | 97.79 |
| | **Ours: VPC** | 0.95 | 98.56 | 1.60 | 98.81 | 2.00 | 98.61 | 2.55 | 98.30 | 7.10 | 97.34 | **2.84** | 98.32 |
| DenseNet-121 + Projection Layer | OE | 1.40 | 99.33 | 0.65 | 99.52 | 1.80 | 99.16 | 8.10 | 97.51 | 2.20 | 99.06 | 2.83 | 98.91 |
| | Energy-OE | 0.85 | 99.52 | 0.45 | 99.18 | 1.75 | 98.89 | 8.00 | 97.58 | 3.55 | 98.70 | 2.92 | 98.77 |
| | DAL | 1.10 | 99.32 | 0.40 | 99.55 | 0.60 | 98.98 | 2.45 | 98.98 | 7.30 | 97.65 | 2.37 | **98.92** |
| | PFS | 1.15 | 98.76 | 0.35 | 99.50 | 0.20 | 99.24 | 7.55 | 96.87 | 1.70 | 98.68 | 2.19 | 98.61 |
| | Ours: VPC | 0.65 | 99.33 | 0.45 | 99.25 | 0.55 | 98.98 | 2.05 | 98.87 | 6.80 | 97.89 | **2.10** | 98.86 |
| **CIFAR-100** | | | | | | | | | | | | | |
| *With auxiliary OOD data* | | | | | | | | | | | | | |
| WideResNet-40-2 + Projection Layer | OE | 42.25 | 92.80 | 14.40 | 97.40 | 53.00 | 85.80 | 51.80 | 87.55 | 43.90 | 90.47 | 41.07 | 90.80 |
| | Energy-OE | 32.75 | 95.05 | 18.05 | 96.96 | 61.90 | 85.38 | 50.75 | 89.70 | 42.95 | 91.27 | 41.28 | 91.67 |
| | DAL | 14.45 | 97.23 | 13.30 | 97.38 | 36.80 | 92.49 | 40.05 | 91.41 | 49.60 | 88.51 | **30.84** | 93.48 |
| | PFS | 21.45 | 96.17 | 17.45 | 96.89 | 48.05 | 89.27 | 50.45 | 90.11 | 39.15 | 92.09 | 35.31 | 92.91 |
| | **Ours: VPC** | 9.95 | 97.98 | 26.25 | 95.75 | 26.50 | 94.97 | 45.05 | 89.67 | 52.45 | 89.88 | 32.04 | **93.65** |
| ResNet-18 + Projection Layer | OE | 47.55 | 91.28 | 37.05 | 92.81 | 43.55 | 91.71 | 56.85 | 85.96 | 53.85 | 87.48 | 47.77 | 89.85 |
| | Energy-OE | 29.95 | 95.03 | 30.85 | 94.92 | 25.80 | 95.00 | 54.90 | 88.06 | 45.35 | 90.36 | **37.37** | **92.68** |
| | DAL | 49.95 | 89.17 | 27.00 | 94.95 | 30.45 | 94.42 | 48.55 | 89.32 | 54.85 | 87.36 | 42.16 | 91.05 |
| | PFS | 48.15 | 91.71 | 30.10 | 94.57 | 37.65 | 93.75 | 58.10 | 88.01 | 58.05 | 86.66 | 46.41 | 90.94 |
| | **Ours: VPC** | 22.30 | 95.89 | 30.80 | 94.09 | 28.85 | 95.03 | 51.40 | 88.52 | 61.45 | 86.62 | 38.96 | 92.03 |
| DenseNet-121 + Projection Layer | OE | 20.75 | 96.24 | 24.10 | 95.50 | 23.55 | 95.78 | 60.30 | 86.28 | 55.80 | 86.59 | 36.90 | 92.08 |
| | Energy-OE | 17.40 | 97.03 | 18.35 | 96.75 | 56.40 | 89.37 | 56.50 | 89.34 | 45.05 | 92.08 | 38.74 | 92.92 |
| | DAL | 16.85 | 96.62 | 16.25 | 96.92 | 49.80 | 88.28 | 41.40 | 90.52 | 59.50 | 86.72 | 36.76 | 91.81 |
| | PFS | 22.45 | 96.33 | 18.50 | 96.89 | 56.80 | 88.73 | 56.20 | 88.69 | 45.60 | 91.06 | 39.91 | 92.34 |
| | **Ours: VPC** | 12.50 | 97.71 | 19.65 | 96.67 | 22.70 | 95.98 | 42.50 | 91.20 | 58.50 | 88.49 | **31.17** | **94.01** |

Table 16: Detailed Results with different architectures. The best result is in bold.

| Model | Method | SVHN | | LSUN | | iSUN (Far-OOD Datasets) | | Textures | | Places365 | | Average | |
|---|---|---|---|---|---|---|---|---|---|---|---|---|---|
| | | FPR95↓ | AUROC↑ | FPR95↓ | AUROC↑ | FPR95↓ | AUROC↑ | FPR95↓ | AUROC↑ | FPR95↓ | AUROC↑ | FPR95↓ | AUROC↑ |
| **CIFAR-10** | | | | | | | | | | | | | |
| *With auxiliary OOD data* | | | | | | | | | | | | | |
| WideResNet-40-2 | OE | 1.95 | 99.23 | 0.80 | 99.67 | 1.95 | 99.36 | 3.70 | 99.23 | 8.80 | 97.76 | 3.44 | 99.05 |
| | Energy-OE | 1.90 | 99.32 | 0.95 | 98.99 | 3.35 | 98.72 | 4.00 | 98.85 | 8.55 | 97.42 | 3.75 | 98.66 |
| | DAL | 1.40 | 99.36 | 0.95 | 99.53 | 1.35 | 99.02 | 3.50 | 98.99 | 8.65 | 97.39 | 3.17 | 98.84 |
| | PFS | 1.10 | 98.74 | 0.35 | 99.61 | 1.35 | 99.20 | 2.85 | 98.58 | 7.75 | 97.17 | 2.68 | 98.66 |
| | **Ours: VPC** | 0.85 | 99.62 | 0.45 | 99.50 | 1.10 | 99.50 | 2.25 | 99.38 | 6.70 | 97.91 | **2.27** | **99.18** |
| ResNet-18 | OE | 3.25 | 98.37 | 1.25 | 98.97 | 1.30 | 98.78 | 3.05 | 98.35 | 8.45 | 97.34 | 3.46 | 98.36 |
| | Energy-OE | 1.40 | 98.30 | 2.55 | 98.44 | 2.90 | 98.86 | 3.40 | 98.55 | 9.15 | 97.17 | 3.88 | 98.26 |
| | DAL | 1.00 | 99.70 | 1.15 | 99.69 | 1.90 | 98.98 | 2.70 | 99.11 | 8.35 | 97.31 | 3.02 | **98.96** |
| | PFS | 1.45 | 99.29 | 0.60 | 99.31 | 1.95 | 98.88 | 2.45 | 98.88 | 8.80 | 97.26 | 3.05 | 98.72 |
| | **Ours: VPC** | 0.95 | 98.56 | 1.60 | 98.81 | 2.00 | 98.61 | 2.55 | 98.30 | 7.10 | 97.34 | **2.84** | 98.32 |
| DenseNet-121 | OE | 2.05 | 99.18 | 0.85 | 99.44 | 0.45 | 99.33 | 3.20 | 98.77 | 7.65 | 97.56 | 2.84 | **98.86** |
| | Energy-OE | 1.55 | 99.18 | 1.05 | 99.11 | 2.05 | 99.12 | 2.85 | 98.94 | 9.75 | 97.18 | 3.45 | 98.71 |
| | DAL | 1.40 | 99.16 | 0.40 | 99.03 | 0.45 | 99.31 | 2.85 | 98.79 | 7.80 | 97.30 | 2.58 | 98.72 |
| | PFS | 1.40 | 98.25 | 0.60 | 99.32 | 1.15 | 99.11 | 3.35 | 98.44 | 7.85 | 97.24 | 2.87 | 98.47 |
| | Ours: VPC | 0.65 | 99.33 | 0.45 | 99.25 | 0.55 | 98.98 | 2.05 | 98.87 | 6.80 | 97.89 | **2.10** | **98.86** |
| **CIFAR-100** | | | | | | | | | | | | | |
| *With auxiliary OOD data* | | | | | | | | | | | | | |
| WideResNet-40-2 | OE | 28.95 | 95.08 | 10.95 | 97.98 | 49.55 | 89.29 | 41.50 | 91.57 | 49.75 | 89.87 | 36.14 | 92.76 |
| | Energy-OE | 23.80 | 96.18 | 31.90 | 94.88 | 41.40 | 91.67 | 48.10 | 88.09 | 56.50 | 87.66 | 40.34 | 91.69 |
| | DAL | 19.30 | 95.75 | 16.20 | 96.71 | 30.70 | 93.85 | 43.15 | 91.36 | 55.10 | 88.39 | 32.89 | 93.21 |
| | PFS | 24.70 | 95.81 | 12.65 | 97.78 | 38.35 | 91.44 | 44.20 | 91.32 | 51.85 | 90.33 | 34.35 | 93.33 |
| | **Ours: VPC** | 9.95 | 97.98 | 26.25 | 95.75 | 26.50 | 94.97 | 45.05 | 89.67 | 52.45 | 89.88 | **32.04** | **93.65** |
| ResNet-18 | OE | 52.40 | 90.28 | 34.90 | 93.25 | 45.00 | 90.92 | 52.75 | 86.47 | 58.70 | 85.91 | 48.75 | 89.36 |
| | Energy-OE | 35.50 | 94.09 | 41.50 | 93.12 | 56.30 | 88.58 | 45.60 | 89.46 | 52.80 | 88.28 | 46.34 | 90.70 |
| | DAL | 36.50 | 93.94 | 33.85 | 93.39 | 49.80 | 90.15 | 43.95 | 90.10 | 58.75 | 86.74 | 44.57 | 90.87 |
| | PFS | 24.30 | 95.33 | 19.35 | 96.79 | 46.55 | 91.70 | 51.10 | 90.88 | 59.45 | 88.51 | 40.15 | **92.64** |
| | **Ours: VPC** | 22.30 | 95.89 | 30.80 | 94.09 | 28.85 | 95.03 | 51.40 | 88.52 | 61.45 | 86.62 | **38.96** | 92.03 |
| DenseNet-121 | OE | 23.80 | 95.89 | 28.05 | 95.11 | 31.00 | 95.07 | 46.95 | 91.07 | 52.50 | 88.99 | 36.46 | 93.23 |
| | Energy-OE | 17.60 | 96.75 | 35.60 | 93.97 | 54.90 | 91.35 | 57.25 | 89.40 | 59.00 | 87.60 | 44.87 | 91.82 |
| | DAL | 25.00 | 94.94 | 28.25 | 94.32 | 17.60 | 95.80 | 45.35 | 88.16 | 67.55 | 80.08 | 36.75 | 90.66 |
| | PFS | 21.50 | 96.43 | 29.10 | 94.75 | 35.95 | 94.24 | 68.60 | 85.55 | 63.85 | 83.84 | 43.80 | 90.96 |
| | **Ours: VPC** | 12.50 | 97.71 | 19.65 | 96.67 | 22.70 | 95.98 | 42.50 | 91.20 | 58.50 | 88.49 | **31.17** | **94.01** |

## B.4 Visualization Results and Analysis

This section presents a suite of visualizations to illustrate how VPC improves OOD detection on CIFAR-10 and CIFAR-100. The figures highlight OEFS activations, feature orthogonality, ID/OOD

score distributions, cross-metric comparisons, and failure analyses, providing an intuitive view of the advantages of our approach.

### B.4.1 OEFS ACTIVATION ANALYSIS

Figure 6 and 7 show the mean OEFS activations after optimizing VPC with the divergence loss and the convergence loss on CIFAR-10 and CIFAR-100, respectively. The divergence loss induces a canonical activation pattern across EBVs (i.e., activation shape is consistent across EBVs), reflecting the scale and coverage of the OE data. Despite this, OOD samples still exhibit distinct L2 activation magnitudes that our VPC Score can capture. Note that the seemingly higher ID activations along EBVs arise because per-sample activations are dispersed and then averaged to approximately $1/C$ over classes. In contrast, the convergence loss collapses OOD features toward a single EBV and also influences the ID features.

### B.4.2 ORTHOGONALITY OF OOD FEATURES

Figure 8 visualizes, on CIFAR-10, the projections of the first three class features, OOD features, and the corresponding classifier weights/EBVs. Both PFS and our method enforce feature–weight/EBV orthogonality, which leads to stronger OOD separability. We also observe class-wise drift under PFS, whereas our ENC loss via a uniform evidence prior that suppresses such drift.

### B.4.3 IMPACT OF SCORING FUNCTIONS ON ID/OOD DISTRIBUTIONS

Figure 9 and 10 compare ID/OOD score distributions under different scoring functions on CIFAR-10 and CIFAR-100. The VPC Score consistently reduces the overlap between ID and OOD distributions and produces a sharper OOD peak, which is crucial for effective detection.

### B.4.4 VISUAL PERFORMANCE ACROSS SCORING FUNCTIONS

Figures 11 and 12 illustrate that, on CIFAR-10/CIFAR-100, when MSP and Uncertainty perform poorly, the VPC Score remains discriminative; the figures show paired ID/OOD examples. Additional qualitative results on CIFAR-100 are given in Figures 13–18 (ID and OOD datasets including SVHN, LSUN, iSUN, Texture, and Places365), and the corresponding CIFAR-10 results appear in Figures 19–24. In each figure, the left three panels depict cases where all metrics perform well, while the right three highlight challenging cases where the VPC Score performs better.

### B.4.5 FAILURE CASE ANALYSIS

Figure 25 focuses on failures in ID detection: the left panels show representative failures on CIFAR-100, and the right panels show those on CIFAR-10, helping reveal model weaknesses and guide improvements. Figure 26 presents failures in OOD detection, again split by dataset (left: CIFAR-100, right: CIFAR-10), exposing the limitations of the methods under edge conditions.

## C LIMITATION

Our method, Vast Predefined Classifiers (VPC), is founded on a pre-specified Orthogonal Equiangular Feature Space (OEFS), which is populated using a training paradigm guided by our proposed ENC, VEBV, and OC losses. While this approach yields strong empirical results, its foundational design choices introduce several key limitations.

(i) Dependence on Dimensional Constraints. The reliance on a predefined OEFS imposes an architectural prerequisite rooted in the properties of ETFs. Specifically, the existence of a Simplex ETF dictates that the feature dimension must satisfy a strict lower bound determined by the total number of prototypes. This structural coupling introduces a scalability constraint, often necessitating higher-dimensional projections for large-scale applications.

(ii) The enforcement of a rigid equiangular structure for in-distribution classes prioritizes maximal separability at the expense of potentially valuable information regarding inter-class semantic similarity. While our focus is on leveraging OE data for stable separation, incorporating hierarchical prototype structures into the OEFS framework remains a promising avenue for future research.

(iii) The multi-objective training paradigm presents its own set of challenges. The stable convergence required to achieve an ideal Neural Collapse state for in-distribution data can be disrupted by the competing objectives of attracting OOD features to their subspace while enforcing orthogonality. This inherent tension makes the full materialization of the ideal geometry difficult.

# D LLM USAGE STATEMENT

We utilized LLMs to assist in the drafting and editing process, aiming to enhance the clarity, coherence, and quality of the text and technical presentations. All content generated or refined by LLMs has been thoroughly reviewed, verified, and revised by the authors, who assume full responsibility for all scientific content.

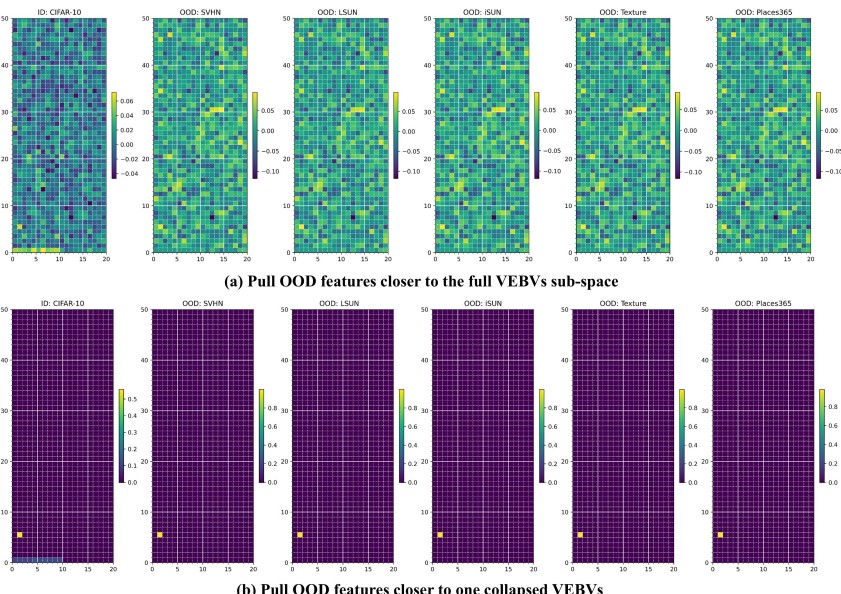

Figure 6: Mean activation after optimizing VPC with divergence and convergence loss on CIFAR-10.

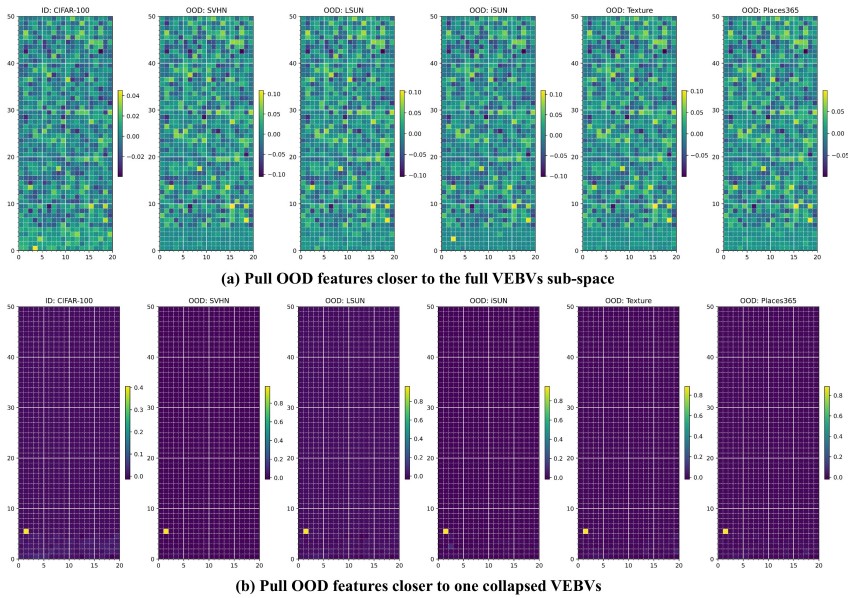

Figure 7: Mean activation after optimizing VPC with divergence and convergence loss on CIFAR-100.

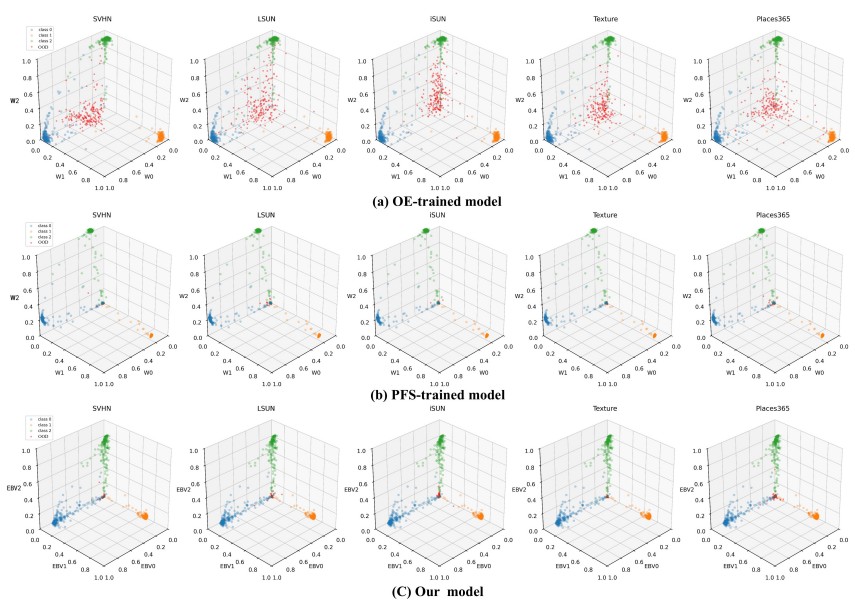

Figure 8: Orthogonality visualization of OOD features and top three class features on CIFAR-10, comparing projections of the top three class features, OOD features with classifier weights/EBVs. PFS and our method optimize OOD detection via orthogonal constraints.

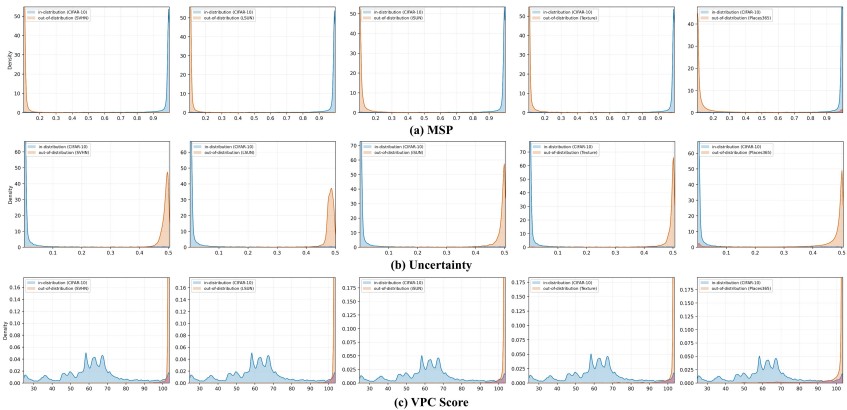

Figure 9: ID/OOD distribution plot on CIFAR-10 with various scoring functions for OOD detection. VPC Score minimizes overlap and boosts OOD distribution peak sharpness.

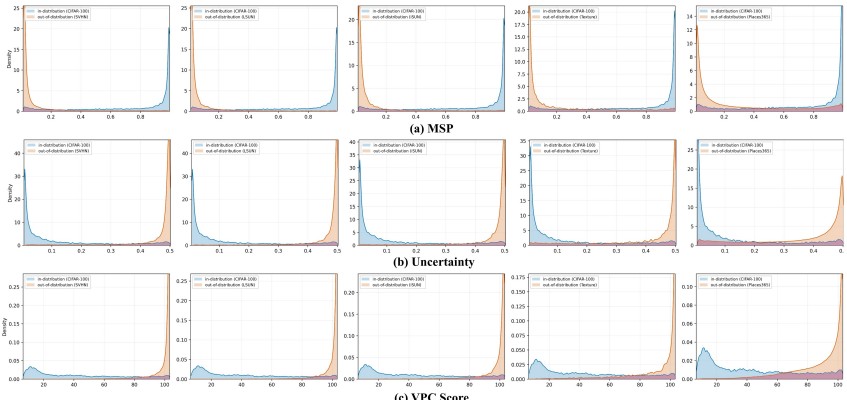

Figure 10: ID/OOD distribution plot on CIFAR-100 with various scoring functions for OOD detection. VPC Score minimizes overlap and boosts OOD distribution peak sharpness.

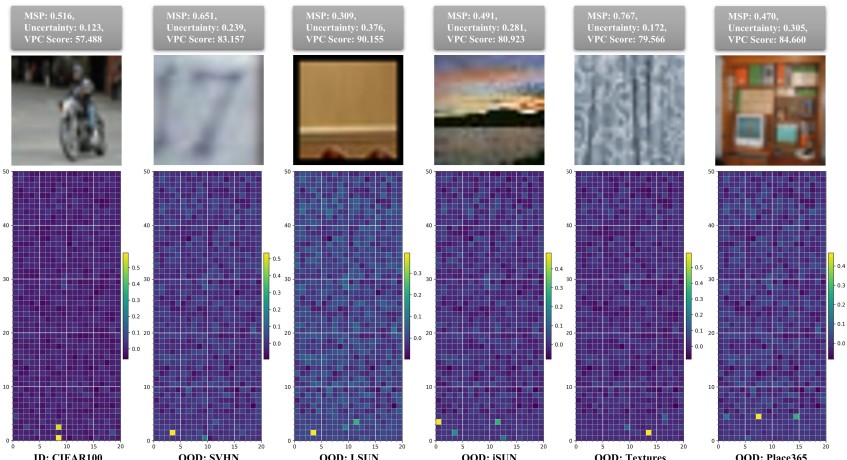

Figure 11: Visualization of ID and OOD detection using different scoring functions in CIFAR-100. VPC Score maintains distinguishability even when MSP and Uncertainty perform poorly.

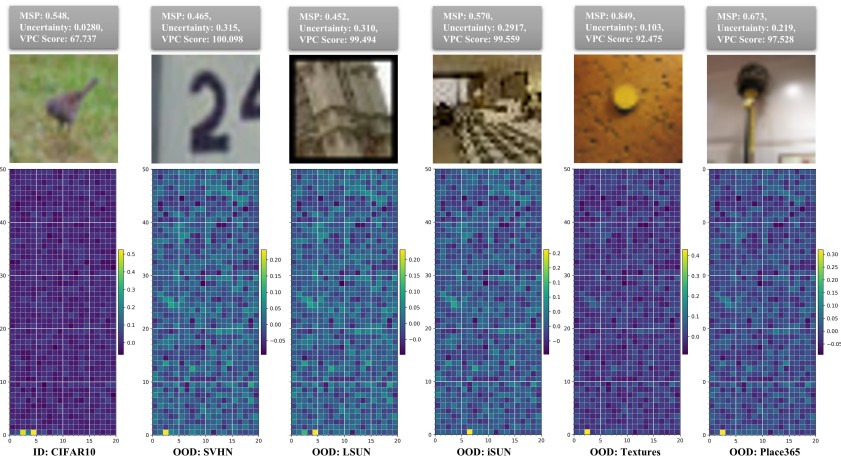

Figure 12: Visualization of ID and OOD detection using different scoring functions in CIFAR-10. VPC Score maintains distinguishability even when MSP and Uncertainty perform poorly.

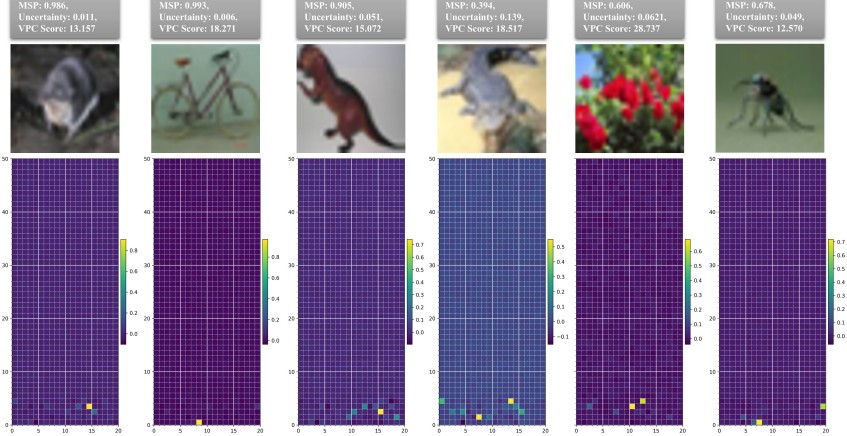

Figure 13: Visualization of ID detection using different scoring functions in CIFAR-100.

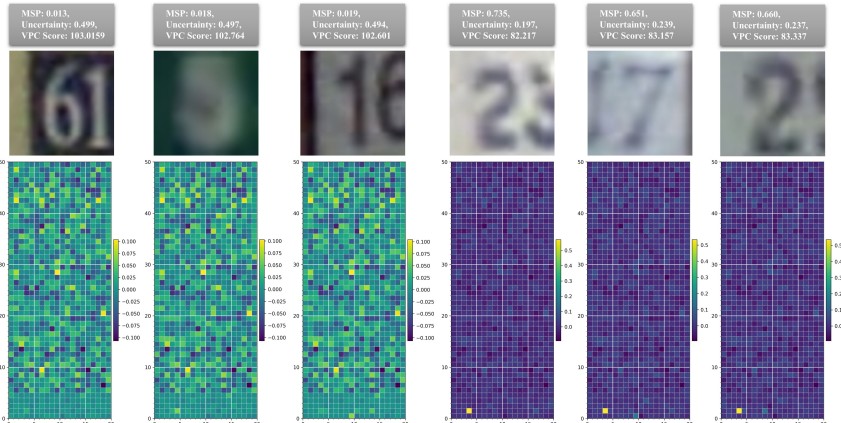

Figure 14: Visualization of OOD:SVHN detection using different scoring functions in CIFAR-100.

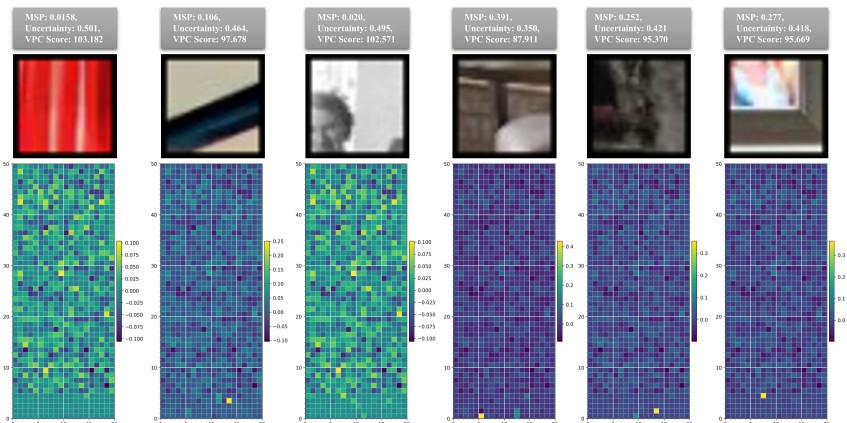

Figure 15: Visualization of OOD:LSUN detection using different scoring functions in CIFAR-100.

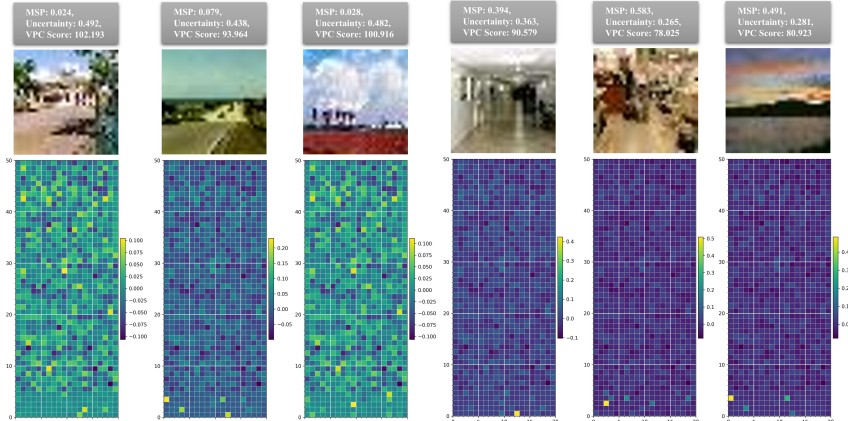

Figure 16: Visualization of OOD:iSUN detection using different scoring functions in CIFAR-100.

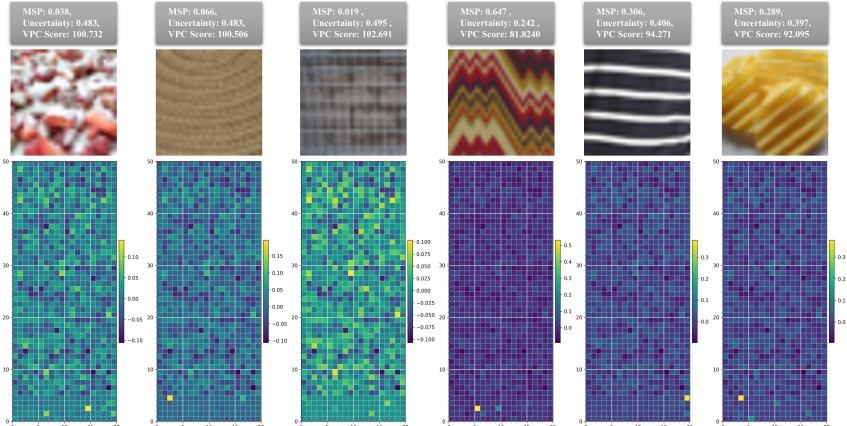

Figure 17: Visualization of OOD:Texture detection using different scoring functions in CIFAR-100.

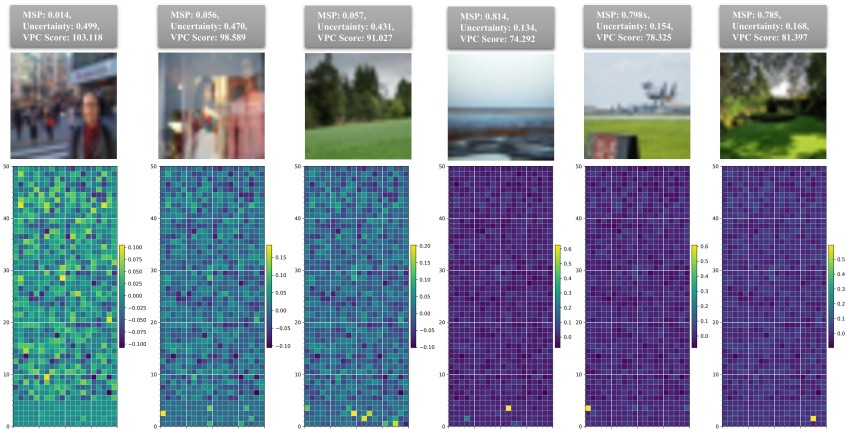

Figure 18: Visualization of OOD:Place365 detection using different scoring functions in CIFAR-100.

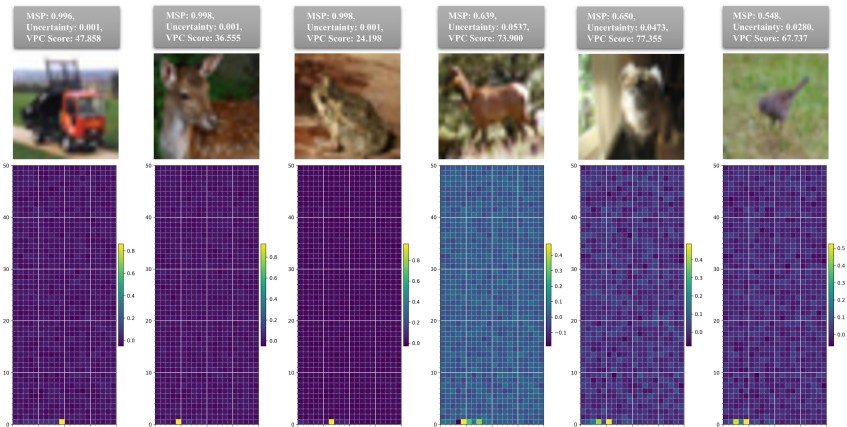

Figure 19: Visualization of ID detection using different scoring functions in CIFAR-10.

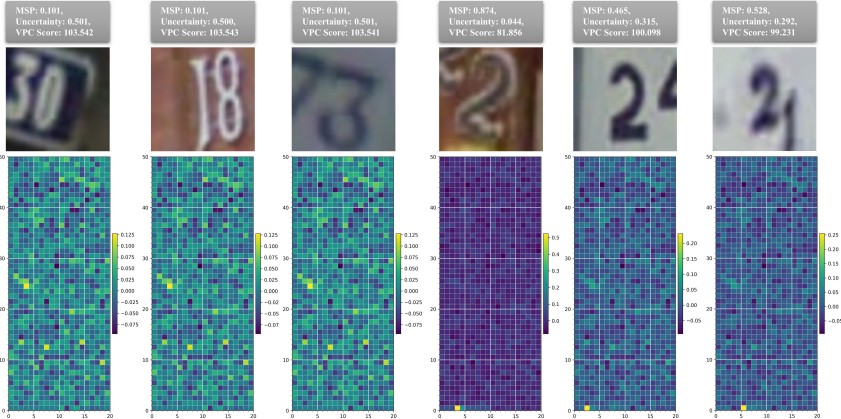

Figure 20: Visualization of OOD:SVHN detection using different scoring functions in CIFAR-10.

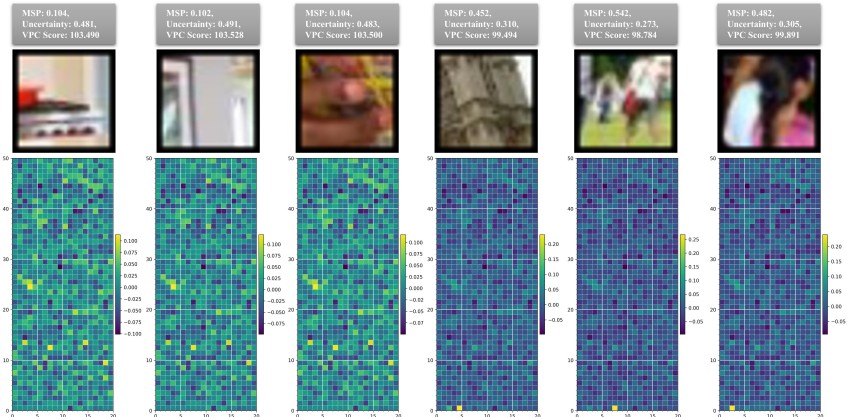

Figure 21: Visualization of OOD:LSUN detection using different scoring functions in CIFAR-10.

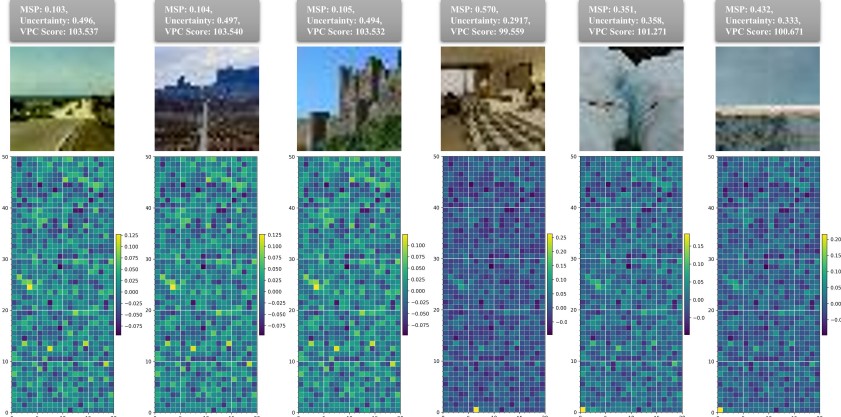

Figure 22: Visualization of OOD:iSUN detection using different scoring functions in CIFAR-10.

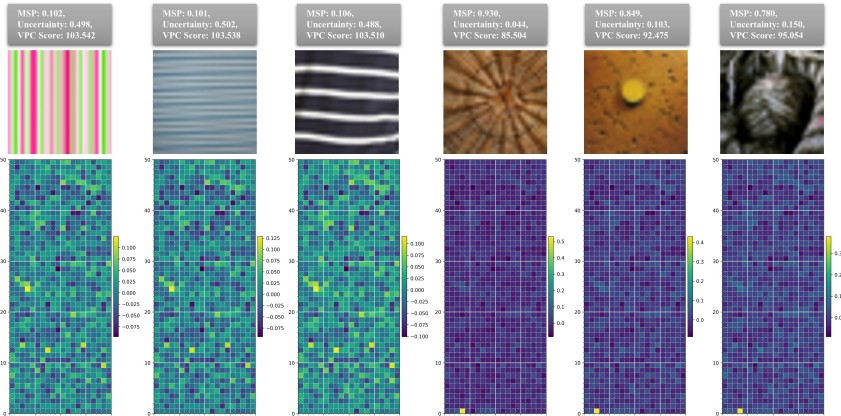

Figure 23: Visualization of OOD:Texture detection using different scoring functions in CIFAR-10.

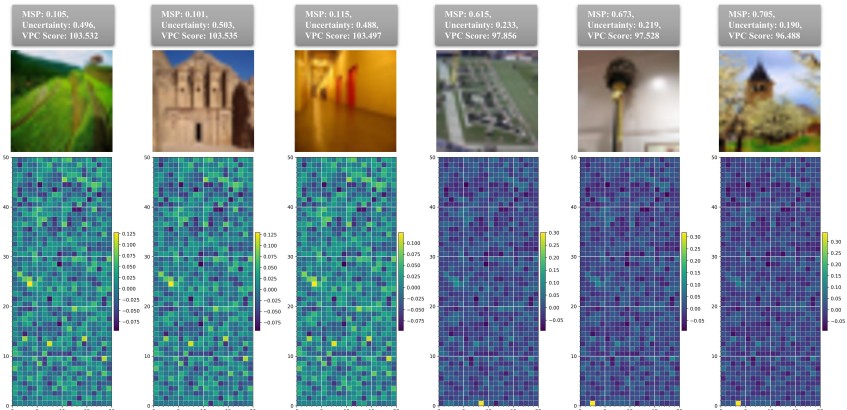

Figure 24: Visualization of OOD:Place365 detection using different scoring functions in CIFAR-10.

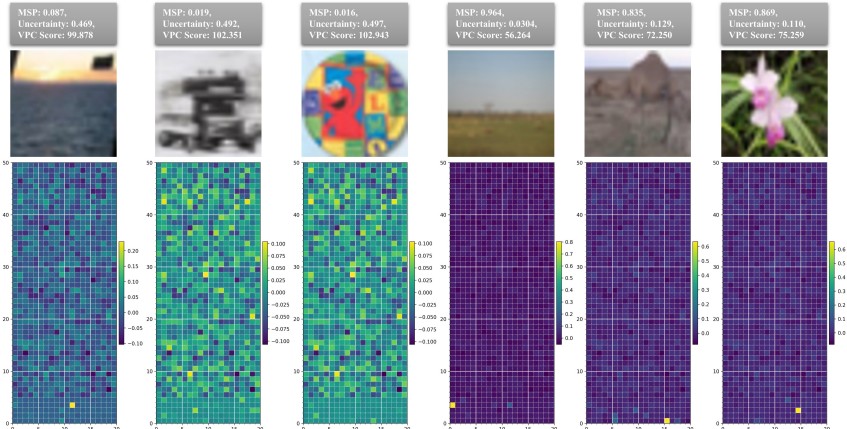

Figure 25: Visualization of ID detection using different scoring functions in CIFAR-100 and CIFAR-10. The left three images show failure cases for ID detection in CIFAR-100; the right three images show failure cases in CIFAR-10.

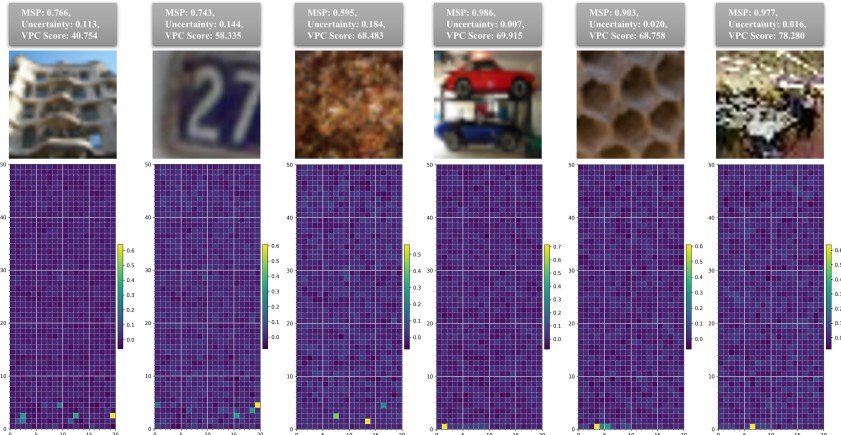

Figure 26: Visualization of OOD detection using different scoring functions in CIFAR-100 and CIFAR-10. The left three images show failure cases for OOD detection in CIFAR-100; the right three images show failure cases in CIFAR-10.

