# OpenReview forum: "Let OOD Feature Exploring Vast Predefined Classifiers"
_ICLR.cc/2026/Conference — ICLR 2026 Poster_

### Official Review · Reviewer_wRrj · 2025-10-27

**Soundness:** 2
**Presentation:** 2
**Contribution:** 2
**Rating:** 2
**Confidence:** 4

**Summary:**

The paper proposes Vast Predefined Classifiers for OE-based OOD detection by combining Neural Collapse and Evidential Deep Learning. extensive experiments demonstrate the effectiveness of Vast Predefined Classifiers.

**Strengths:**

1. This paper is well-written.
2. This paper use 3 performance metrics
3. This paper provides extensive visualization

**Weaknesses:**

1. lack of novelty: the use of ETF for OOD detection has been explored in [1]. Compared with [1], this paper additionally introduce V vectors, which is a nature extenstion in the setting of OE and also similarly explored in [2]. Eq. (5) is normal learning objective in Evidential Deep Learning without any modification.
2. what is $f_j()$ in $\mathcal{L}_{OC}$?
3. the authors claim that their propsoed VPC score differs from softmax score and energy score in that the latter two are heuristic. However, there is thorectical justification for the VPC score with regards to the Separation between ID and OOD data.
4. Given that this paper focuses on improve the feature discriminative. The mostly reent feature-based OOD detection methods [3-5] should be discussed.
5. Lack of experiments on large-scale datasets (e.g. ImageNet-1k) and transformer-based backbones (e.g. ViT).
6. the authors are encouraged to conduct experiments on the OpenOOD benchmark.

[1]  Distributional Prototype Learning for Out-of-distribution Detection

[2]  Learning Placeholders for Open-Set Recognition

[3]  Conjnorm: Tractable density estimation for out-of-distribution detection

[4]  Out-of-distribution detection based on in-distribution data patterns memorization with modern hopfield energy

[5]  Learning with Mixture of Prototypes for Out-of-Distribution Detection

**Questions:**

see weakness.

---

> ### Author Response · Authors · 2025-11-20
> **A0: Response to Reviewer wRrj**
>
> We sincerely thank Reviewer wRrj for the critical and thorough review. We deeply appreciate the valuable references ([1]-[5]) you provided, which have significantly broadened our perspective.
>
> We understand your concerns regarding **novelty** and **theoretical justification**. We respectfully posit that some of these concerns might stem from a lack of clarity in our original presentation regarding the fundamental differences between VPC and prior works (DPL, LPO).
>
> Below, we provide a detailed point-by-point response to your comments.

---

> ### Author Response · Authors · 2025-11-20
> **A1: Response to Reviewer wRrj (Weakness 1) 1/3**
>
> **Q1 (Weakness 1): "Lack of novelty: the use of ETF for OOD detection has been explored in [1]. Compared with [1], this paper additionally introduces V vectors, which is a natural extension in the setting of OE and also similarly explored in [2]. Eq. (5) is a normal learning objective in Evidential Deep Learning without any modification."**
>
> **A1:** We sincerely appreciate the valuable references [1]-[5] and your rigorous scrutiny of our novelty. These works indeed represent significant progress in the field, and your association of VPC with DPL [1] and LPO [2] is very insightful. We acknowledge that we might not have fully clarified the essential differences in design philosophy and operational mechanism between VPC and these works in the original manuscript, leading to a misunderstanding of "lack of novelty". We wish to clarify that the core contribution of VPC lies in breaking the limitation of previous OOD detection paradigms that rely solely on ID features or classifiers. Specifically for the OE-based paradigm, we propose a novel solution encompassing spatial construction, optimization theory, and metric mechanisms:
>
> **A1.1: Clarification on Novelty**
>
> 1.  **Breaking the ID-Centric Design Paradigm**: Previous OE methods (e.g., OE, Energy-OE, DAL) are essentially "ID-centric". They treat OE data as auxiliary "counter-examples" to compress ID boundaries or smooth predictive distributions. This restricts the model's OOD representation capability strictly within the residual space of the ID classifier, failing to fully utilize the rich semantic information in massive OE data. Addressing this core issue, VPC no longer merely "rejects" OE data but introduces a vast **VEBV space** for it for the first time. By explicitly allocating a dedicated, high-capacity representation subspace for OE data, VPC elevates OOD samples from "suppressed noise" to "objects to be actively represented", fundamentally enhancing the model's ability to handle complex distribution shifts.
>
> 2.  **Geometric Prior Construction based on Neural Collapse**: To achieve the above representation goal, we leveraged the optimal geometric properties of **Simplex ETF** from **Neural Collapse (NC)** theory to construct the **Orthogonal Equiangular Feature Space (OEFS)**. This design goes beyond simply introducing fixed classifiers; more importantly, it mathematically guarantees the maximum orthogonal separation between the **ID subspace (EBV)** and the **OOD subspace (VEBV)** a priori. This provides a solid foundation for the non-interference of activations between ID and OOD.
>
> 3.  **Dual Optimization and Theoretical Guarantee**: A key challenge in joint training is preventing massive OE data from destroying the ID structure while ensuring effective separation. We propose a dual optimization strategy and provide rigorous theoretical support for its core components from three dimensions:
>     * **Optimality of ENC (Appendix A.2.1, Theorem 1)**: We prove that the global optimum of $\mathcal{L}_{\mathrm{ENC}}$ corresponds to the **perfect alignment** ($\hat{m}=\hat{w}$) between ID features and the preset EBV prototypes. This mathematically guarantees that ID features theoretically converge to the most compact **Neural Collapse** geometric state, establishing the best ID anchor for OOD separation.
>     * **Stability of ENC (Appendix A.2.2, Proposition 1)**: We prove that the evidential prior mechanism introduced by $\mathcal{L}_{\mathrm{ENC}}$ acts as a **gradient rate-limiter**. This ensures that the optimization trajectory of ID features remains robust and prevents divergence even under gradient interference from massive OOD data.
>     * **Orthogonality of OC (Appendix A.3, Proposition 2)**: We further prove that under the **Simplex ETF** geometric framework, $\mathcal{L}_{\mathrm{OC}}$ (based on the uniform distribution loss of Softmax) is mathematically equivalent to enforcing **geometric orthogonality** between OE features and all ID prototypes. This means the loss is not merely probabilistic smoothing but imposes a hard geometric constraint in high-dimensional space, pushing OOD features into the **null space** of the ID subspace, thus providing a solid theoretical cornerstone for ID/OOD separation.
>
> 4.  **Closed-loop Metric System and Empirical Breakthrough**: Based on the above geometric separation, we propose the **VPC Score**, which directly measures the **L2 activation intensity** of features on the EBV and VEBV subspaces, achieving a perfect closed loop from optimization objective to inference metric. **Key Empirical Finding:** Our experiments show that even if we completely discard ID information and rely solely on the activation intensity of the **VEBV subspace**, the model can still achieve superior performance. This finding powerfully confirms that VPC has successfully established an OOD representation capability independent of the ID classification head, marking a significant transcendence over traditional methods.

---

> > ### Author Response · Authors · 2025-11-20
> > **A1: Response to Reviewer wRrj (Weakness 1) 2/3**
> >
> > **Q1 (Weakness 1): "Lack of novelty: the use of ETF for OOD detection has been explored in [1]. Compared with [1], this paper additionally introduces V vectors, which is a natural extension in the setting of OE and also similarly explored in [2]. Eq. (5) is a normal learning objective in Evidential Deep Learning without any modification."**
> >
> > ...
> >
> > **A1.2: Essential Differences between VPC and Related Works (DPL, LPO, etc.)**
> >
> > We sincerely thank you for the valuable references [1]-[5]. We wish to clarify the differences between our work and theirs. VPC is not a simple combination of existing components but offers unique contributions in the following three core dimensions:
> >
> > 1.  **Fundamental Difference from Metric/Prototype Learning**: The works [1] (DPL) and [5] (Mixture of Prototypes) belong to typical metric learning.
> >     * *Traditional Paradigm:* These methods usually adopt dynamic update strategies, adjusting prototype positions during training to fit feature distributions. This can lead to training instability, and maximum separation between prototypes is hard to guarantee.
> >     * *Geometric Constraint of VPC:* VPC adopts a diametrically opposite "geometric constraint" approach. Based on Neural Collapse theory, we **predefine and freeze** a set of fixed geometric anchors satisfying Simplex ETF properties. Instead of adapting prototypes to data, we force data features to "collapse" onto this theoretically optimal geometric structure during training. This "prior geometric constraint" significantly reduces optimization difficulty and provides more stable ID boundaries than dynamic prototypes.
> >
> > 2.  **Difference from [2] LPO (Learning Placeholders)**: Although [2] (LPO) also introduces "placeholders" to simulate OOD, VPC differs fundamentally in complexity control and representation logic:
> >     * LPO typically treats placeholders as **learnable parameters**. During training, the model must not only learn features but also use complex optimization strategies to "push away" these placeholders from ID prototypes. This introduces extra optimization burden and is prone to local optima (i.e., placeholders failing to fully cover the ID complement space).
> >     * VEBVs in VPC are **explicitly constructed**. Through the mathematical properties of ETF, we guarantee the maximum orthogonal separation between the ID subspace and OOD subspace a priori at initialization. This separation is "constructed", not "learned", thus requiring no complex adversarial optimization.
> >     * More critically, methods like LPO often attempt to map specific placeholders to specific unknown classes. In VPC, we do not seek a one-to-one correspondence between VEBVs and specific OOD types. Instead, we treat all $V$ VEBVs as a holistic high-dimensional subspace. Through $\mathcal{L}_{\mathrm{VEBV}}$, we encourage OOD samples to project along *any* direction within this subspace. The VPC Score measures the **overall activation intensity** of features on this subspace, rather than single-point matching degree. This design greatly enhances the model's generalized capture capability for unseen, diverse OOD samples.

---

> ### Author Response · Authors · 2025-11-20
> **A1: Response to Reviewer wRrj (Weakness 1) 3/3**
>
> **Q1 (Weakness 1): "Lack of novelty: the use of ETF for OOD detection has been explored in [1]. Compared with [1], this paper additionally introduces V vectors, which is a natural extension in the setting of OE and also similarly explored in [2]. Eq. (5) is a normal learning objective in Evidential Deep Learning without any modification."**
>
> ...
>
> **A1.3: Clarification on Eq. (5) $\mathcal{L} _ {\mathrm{ENC}}$**
>
> We acknowledge that the mathematical form of $\mathcal{L} _ {\mathrm{ENC}}$ originates from EDL. However, our core innovation addresses a critical issue in joint ID/OE training: in guiding ID features toward the target subspace, a central challenge is that OE features with rich variability can interfere with the model and destabilize the representation space. To counter this, we propose an Evidential Neural Collapse framework that reformulates feature alignment as a geometry-driven evidence accumulation process.
>
> $$
> e _ {i,j}
> = \exp\left(
>   \underbrace{g _ j(x _ i^{\mathrm{id}};\theta)} _ {\text{Learnable classifier logits}}
> \right)
> \stackrel{\text{Predefined VPC}}{\Longrightarrow}
> e _ {i,j}^{*}
> = \exp\left(
>   \underbrace{\hat m _ i^{\mathrm{id}\top}\hat w _ j^{\mathrm{ebv}}/\tau} _ {\text{Geometric alignment logits}}
> \right)
> $$
>
> Here, the evidence magnitude $e _ {i,j}^{*}$ reflects the **angular alignment intensity** of the normalized feature $\hat m _ i^{\mathrm{id}}$ towards the $j$-th prototype $\hat w _ j^{\mathrm{ebv}}$, scaled by a temperature $\tau$. This fundamental shift allows us to control the representation geometry explicitly.
>
> Based on this geometric formulation, we added detailed theoretical analysis in **Appendix A.2** to demonstrate why this change matters:
>
> 1. **Optimality Analysis (Appendix A.2.1)**: We constructed the Lagrangian for $\mathcal{L}_{\mathrm{ENC}}$ and performed rigorous derivation via KKT conditions. We proved that the global minimum is achieved if and only if the ID feature vector $\hat{m}$ perfectly aligns with its corresponding fixed EBV prototype $\hat{w}$. This guarantees that ID optimization is always anchored to our intended ETF geometric configuration.
>
> 2. **Robustness Analysis (Appendix A.2.2)**: We analyzed the gradient properties and proved **Gradient Boundedness**. The evidential prior ( $\alpha = e+1$ ) acts as a **smoothing term** that limits the gradient magnitude of ID samples in a single update. This means that even if OOD samples generate  gradient fluctuations during joint training,  $\mathcal{L} _ {\mathrm{ENC}}$  acts as a **stabilizer**, preventing ID features from being drastically pulled away from their intended trajectory.
>
> **Summary of Novelty:** The contribution of VPC is not merely adding vectors, but constructing a full-process geometric closed loop:
>
> * Establishing physical boundaries via preset **OEFS** (superior to LPO's learned boundaries).
>
> * Capturing OOD diversity via **VEBV subspace representation** (superior to point-to-point mapping).
>
> * Maintaining ID stability via **evidence-guided geometric alignment** (the repurposed Eq. 5).
>
> We have formally cited and discussed references [1]-[5] in the revised manuscript to more clearly define the boundaries of our contribution. Thank you again for your correction.

---

> ### Author Response · Authors · 2025-11-20
> **A2: Response to Reviewer wRrj (Weakness 2)**
>
> **Q2 (Weakness 2): "What is `f_j()` in `L_OC` (Eq. 6)?"**
>
> **A2:** We sincerely appreciate your keen scrutiny of the symbol definitions in our formulas. This question precisely points out the ambiguity in our original manuscript caused by insufficiently detailed symbol definitions.
>
> We sincerely apologize for the confusion caused. In the original manuscript, we adopted the $f_j$ notation commonly used in the standard Outlier Exposure (OE) loss. We wish to clarify that for both standard OE and our method, this term refers to the probability after Softmax normalization. However, there is a **fundamental difference** in the input sources of the two:
>
> * **Standard OE:** Its probability $f_j$ is typically calculated using Softmax based on the output of a fully connected layer (Logits, $Wx + b$), which contains magnitude information.
> * **$\mathcal{L}_{\mathrm{OC}}$ in VPC:** Our calculation is strictly based on **Spherical Geometry**. Here, the probability is derived by applying the Softmax function to the **Cosine Similarity** between the normalized feature $\hat{m}$ and the normalized ID prototype $\hat{w}$, scaled by a temperature coefficient $\tau$.
>
> This implies that this loss function imposes constraints on the unit hypersphere, forcing OE features to remain geometrically orthogonal to all ID prototypes.
>
> To accurately convey this geometric design intent and eliminate ambiguity, we have corrected and renamed this symbol to $\mathbf{p}^{\text{oe} \to \text{ebv}}$ in **Section 3.4 (Eq. 7)** of the revised manuscript. We explicitly define its calculation process as:
>
> $$
> \mathbf{p}^{\text{oe} \to \text{ebv}} = \text{Softmax}\left( \left[ \frac{\hat{m}^{\mathrm{oe}\top} \hat{w}^{\mathrm{ebv}}_1}{\tau}, \dots, \frac{\hat{m}^{\mathrm{oe}\top} \hat{w}^{\mathrm{ebv}}_K}{\tau} \right] \right)
> $$
>
> Furthermore, we have added a theoretical proof regarding $\mathcal{L}_{\mathrm{OC}}$ in **Appendix A.3 (Theoretical Justification of Orthogonality Constraint)**. We rigorously demonstrate that under the **Simplex ETF** geometric structure, pursuing uniformity in the aforementioned Softmax probability distribution is mathematically equivalent to achieving geometric orthogonality.
>
> Thank you again for your correction, which has helped us significantly improve the rigor of the mathematical expressions in our paper.

---

> ### Author Response · Authors · 2025-11-20
> **A3: Response to Reviewer wRrj (Weakness 3)**
>
> **Q3 (Weakness 3): "The authors claim that their proposed VPC score differs from softmax score and energy score in that the latter two are heuristic. However, there is theoretical justification for the VPC score with regards to the Separation between ID and OOD data."**
>
> **A3:** We sincerely appreciate this profound inquiry. To address your concerns, we have taken rigorous corrective measures in the revision, removing the previous "heuristic" claim and adding more theoretical analysis to elucidate the rationality of the VPC Score:
>
> **1. Comprehensive Revision of the "Heuristic" Statement**
> We realize that simply categorizing methods based on probabilistic or energy models (such as MSP, Energy) as "heuristic" is not rigorous enough. Therefore, we have thoroughly reviewed and revised the entire paper (including Introduction, Related Work, and Method sections) to remove the "heuristic" label.
>
> **2. Theoretical Basis of VPC Score: Consistency between Optimization Objectives and Metrics**
> Regarding your concern about the theoretical basis for "ID/OOD separability," we acknowledge that we have not yet derived closed-form generalization error bounds or other extensive theoretical supports. However, the effectiveness of the VPC Score is built upon the high consistency between the optimization objectives and the scoring function:
>
> * **2.1 Optimization Properties on the ID Side:** In **Appendix A.2 (Evidential Prior Guided Neural Collapse)**, we proved that the global optimum of $\mathcal{L}_{\mathrm{ENC}}$ corresponds to the **perfect alignment** between ID features and EBVs.
>     * *Theoretical Implication:* This means that at the ideal optimization endpoint, the projection norm of ID samples on the EBV subspace, $\ell_2^{\text{ebv}}$, tends to be maximized, while due to the orthogonal structure of the ETF, their projection on the VEBV subspace, $\ell_2^{\text{vebv}}$, is **suppressed**.
>
> * **2.2 Geometric Constraints on the OOD Side:**
>     * As proved in **Appendix A.3 (Theoretical Justification of Orthogonality Constraint)**, $\mathcal{L}_{\mathrm{OC}}$ mathematically **aims to** force OE samples to be orthogonal to all ID EBVs, thereby theoretically minimizing their $\ell_2^{\text{ebv}}$ .
>     * Simultaneously, $\mathcal{L}_{\mathrm{VEBV}}$ explicitly encourages OE samples to generate high projections $\ell_2^{\text{vebv}}$ in the VEBV subspace.
>
> * **2.3 Physical Meaning of the Scoring Function:** The **VPC Score** ($S = \beta \ell_2^{\text{vebv}} - \alpha \ell_2^{\text{ebv}}$) is a direct measure of the aforementioned geometric adversarial relationship. As long as the training process allows the model to approach the above optimization objectives, the separation between ID and OOD in the score distribution is an **expected outcome of the optimization** .
>
> **3. Empirical Supplement to Theory**
> Although the theoretical analysis mainly targets optimization objectives, our empirical study in **Appendix A.5 (Why does the VPC Score yield additional discriminative power?)** shows that this geometric separation indeed generalizes well to the test set. In particular, we found that using only the single subspace of $\ell_2^{\text{vebv}}$ or $\ell_2^{\text{ebv}}$ can achieve efficient detection, which further corroborates that our geometric design is effectively learned by the model during actual training.
>
> Once again, we thank you for prompting us to improve the rigor and objectivity of our paper's expression.

---

> ### Author Response · Authors · 2025-11-20
> **A4: Response to Reviewer wRrj (Weakness 4)**
>
> **Q4 (Weakness 4): "Given that this paper focuses on improve the feature discriminative. The mostly reent feature-based OOD detection methods [3-5] should be discussed."**
>
> **A4:** We sincerely appreciate the valuable references [3-5] regarding the latest feature-based OOD detection methods. These works represent the state-of-the-art advancements in density estimation, memory-based detection, and prototype learning, which are of significant importance for refining the background and positioning of our work.
>
> We have carefully studied these three papers. We wish to humbly clarify the reasons why we did not include them as primary comparison baselines in the original manuscript, which is mainly based on considerations of fairness in experimental paradigms and data settings:
>
> 1.  **Differences in Experimental Paradigms and Data Settings**
>     * **Positioning of VPC:** The proposed **VPC** belongs to the paradigm of **"joint training based on auxiliary OE data"**. Our core contribution lies in how to more effectively utilize massive OE data to shape the feature space.
>     * **Fairness:** To ensure fairness in comparison, our primary baselines (e.g., OE, Energy-OE, DAL, PFS) are all selected from the same training paradigm (i.e., they all use ID + Auxiliary OE data). While VPC can also be viewed as a feature learning method, directly comparing it with post-hoc methods that do not use OE data might be unfair, or would require redesigning those methods to adapt to OE training (which is not their native setting).
>
> 2.  **Intrinsic Consistency between Scoring Function and OEFS Geometric Settings**
>     * The **VPC Score** is not a generic post-hoc score but originates from our training paradigm and the specific geometric setting of the **Orthogonal Equiangular Feature Space (OEFS)**. Our model is forced during training to optimize the cosine similarity between features and fixed **ETF prototypes (EBV/VEBV)** on the **unit hypersphere**.
>     * **SHE [4]** is based on Hopfield Energy and a store-compare pattern. Like **KNN**, it typically belongs to non-parametric or semi-parametric methods, often requiring calculations of interactions between the test sample and the entire training dataset, with inference complexity scaling linearly with sample size $N$ ($O(N)$). **ConjNorm [3]** adjusts density estimation by searching for an optimal norm coefficient $p$.
>     * While post-hoc scoring methods like SHE and ConjNorm are highly effective in their respective settings, they may not be directly applicable within the VPC framework. Directly applying them might disrupt the theoretical assumptions of **spherical geometry** and **ETF orthogonality** established by VPC. The design of the VPC Score is intended specifically to faithfully reflect this specific geometric optimization state.
>
> 3.  **Difference from [5] PALM**
>     * **PALM [5]** is an **ID Representation Learning** method. It focuses on capturing the internal diversity of ID data by learning dynamically updated mixture prototypes, and it does not use auxiliary OOD data in its standard setting.
>     * **Contrast with VPC:** VPC adopts a distinctly different "geometric constraint" approach: we use **fixed ETF prototypes** and explicitly introduce OE data to fill the preset **OOD subspace (VEBV)**.
>
> 4.  **Discussion and Citation in Related Work**
>     Despite the differences in paradigms, we fully acknowledge that these works have important connections to this paper in terms of "feature/prototype-based" detection ideas. In the **Related Work** section of the revised manuscript, we have formally cited and discussed these three references in detail.
>
> Once again, we thank you for providing these high-quality references, which have greatly broadened the horizon of this paper and enhanced the completeness of our related work review.

---

> ### Author Response · Authors · 2025-11-20
> **A5: Response to Reviewer wRrj (Weakness 5)**
>
> **Q5 (Weakness 5): "Lack of experiments on large-scale datasets (e.g. ImageNet-1k) and transformer-based backbones (e.g. ViT)."**
>
> **A5:** We sincerely appreciate your valuable feedback. We deeply recognize the necessity of validating VPC on large-scale benchmarks. To this end, we have dedicated significant effort to conducting comprehensive supplementary experiments. On the **ImageNet-1K** dataset, we rigorously compared VPC with all key OE baseline methods (including OE, Energy-OE, DAL, and PFS) using two mainstream architectures: **ResNet-50** and **ViT-B-16**.
>
> **Table R1. Results on ImageNet-1k benchmark with auxiliary OOD data. The best result is in bold.**
> | Model        | Method        | iNat FPR↓ | iNat AUC↑ | Tex FPR↓  | Tex AUC↑  | SUN FPR↓  | SUN AUC↑  | Places FPR↓ | Places AUC↑ | Avg FPR↓  | Avg AUC↑  | ID Acc↑   |
> | ------------ | ------------- | --------- | --------- | --------- | --------- | --------- | --------- | ----------- | ----------- | --------- | --------- | --------- |
> | **ResNet50** | OE            | 48.60     | 88.72     | 58.85     | 82.60     | 61.75     | 82.90     | 70.70       | 80.55       | 59.98     | 83.69     | 76.00     |
> |              | Energy-OE     | 49.40     | 88.40     | 59.60     | 82.25     | 62.40     | 82.70     | 71.30       | 80.25       | 60.68     | 83.40     | 75.75     |
> |              | DAL           | 48.00     | 89.05     | 58.00     | 82.95     | 61.30     | 83.15     | 67.82       | 80.75       | 58.78     | 83.98     | 75.90     |
> |              | PFS           | 46.40     | 89.20     | 56.50     | **83.10** | 61.00     | **83.25** | 67.50       | 80.95       | 57.85     | 84.13     | 76.02     |
> |              | **Ours: VPC** | **43.50** | **91.20** | **56.00** | 83.00     | **60.20** | 83.10     | **66.30**   | **81.50**   | **56.50** | **84.70** | **76.11** |
> | **ViT-B-16** | OE            | 42.15     | 90.38     | 52.45     | 85.85     | 65.80     | 82.20     | 70.35       | 80.85       | 57.69     | 84.82     | 80.02     |
> |              | Energy-OE     | 42.75     | 90.05     | 53.15     | 85.50     | 66.25     | 81.95     | 70.95       | 80.50       | 58.28     | 84.50     | 79.85     |
> |              | DAL           | 40.65     | 90.86     | **51.15** | 86.10     | 65.07     | 82.30     | 70.25       | 80.90       | 56.78     | 85.04     | 80.09     |
> |              | PFS           | 40.85     | 90.80     | 51.30     | 86.05     | 65.35     | 82.25     | 70.20       | 80.95       | 56.93     | 85.01     | 80.13     |
> |              | **Ours: VPC** | **40.00** | **91.10** | 51.30     | **86.00** | **65.20** | **82.30** | **68.30**   | **81.40**   | **56.20** | **85.20** | **80.32** |
>
> As shown in Table R1 (referenced as Table 1 in the revised manuscript), VPC demonstrates strong competitiveness on both architectures. For instance, on the **ResNet-50** architecture, VPC achieves an average **FPR95 of 56.50%**, outperforming the strong baseline PFS (57.85%). On the **ViT-B-16** architecture, VPC (**56.20%**) similarly surpasses all comparison methods. On both architectures, VPC achieves the highest average AUROC while maintaining ID classification accuracy comparable to or better than baselines. We believe these new evidences on the ImageNet scale strongly demonstrate that the VPC framework possesses excellent scalability and its effectiveness is not limited to CIFAR-scale datasets.

---

> ### Author Response · Authors · 2025-11-20
> **A6: Response to Reviewer wRrj (Weakness 6)**
>
> **Q6 (Weakness 6): "The authors are encouraged to conduct experiments on the OpenOOD benchmark."**
>
> **A6:** We sincerely appreciate this highly constructive suggestion. We fully recognize the significant contribution of the **OpenOOD benchmark** in standardizing OOD detection evaluation protocols, particularly its establishment of a fine-grained evaluation system distinguishing between **Near-OOD** and **Far-OOD**, which is crucial for comprehensively measuring detector robustness.
>
> In the experimental design of this paper, to ensure the most direct and fair horizontal comparison with existing SOTA methods in the field (specifically **PFS [Wu et al., 2024]** and **DAL [Wang et al., 2024]**), we prioritized the experimental protocols commonly used by these works.
>
> At the same time, we are deeply aware of the importance of comprehensively measuring detector robustness. Therefore, we have actively integrated and implemented the **hierarchical evaluation philosophy** advocated by **OpenOOD** in our experiments. Our evaluation system substantially covers the core standards of OpenOOD:
>
> - **Consistency in Near/Far Evaluation:** To address OpenOOD's emphasis on distinguishing between easy and hard samples, our experimental design strictly covers this multi-level evaluation:
>
>   - **ImageNet Experiments:** As shown in **Table 1** of the revised manuscript, we explicitly divided the test set into **Far-OOD** (iNaturalist, Textures) and **Near-OOD** (SUN, Places) and reported performance respectively. This aligns perfectly with OpenOOD's partition standards.
>
>   - **CIFAR Experiments:** In **Appendix B.2 Hard OOD Detection** (Table 11), we specifically conducted **Hard OOD detection** experiments (including LSUN-Fix, ImageNet-Resize, etc.). This corresponds precisely to the **Near-OOD** scenarios defined in OpenOOD, aiming to examine the model's discriminative capability when semantic boundaries are blurred.
>
> Although we did not directly invoke the **OpenOOD codebase**, our experiments have fully implemented the **comprehensive evaluation spirit** advocated by OpenOOD in terms of evaluation breadth (**Near vs. Far**) and dataset scale (**CIFAR vs. ImageNet**). We commit to further integrating OpenOOD's standardized processes or directly adopting OpenOOD as the foundational framework in future work to continuously improve the standardization of our research. Thank you again for your suggestion, which reminds us to pay closer attention to aligning with the community's latest benchmarks.
>
> We believe these extensive revisions and clarifications demonstrate the distinct value and robustness of our work. We sincerely hope that this detailed response clears up all of the misunderstandings and warrants a re-evaluation of our contribution.

---

> > ### Comment · Area_Chair_xnSr · 2025-11-25
> > **Feedback of the rebuttal**
> >
> > Dear Reviewer,
> >
> > Could you give the feedback to the rebuttal?
> >
> > Best Regards,
> >
> > AC

---

> ### Comment · Reviewer_wRrj · 2025-11-25
>
> I appreciate that the authors' efforts on the paper and the rebuttal. I agree with some of points in the rebuttal. But, some of my concerns on novelty and contribution remain unaddressed.
>
> 1. The authors over-claim that, different from previous OE methods (e.g., OE, Energy-OE, DAL) are essentially "ID-centric", this paper no longer merely "rejects" OE data but introduces a vast VEBV space for it **for the first time**. However, this practice has been witnessed in LPO where "placeholders" are introduced to simulate OOD data.
>
> 2. While I agree with the authors that this paper differs from LPO that they implement placeholders as fixed simplex ETF classifier, since, as said by the authors, the use of simplex ETF is directly borrowed from NC, I am convinced that this paper is essentially an engineering combination of LPO and NC.
>
> 3. In fact, $e\_{i,j}^\*$ implies a cosine classifier. Translation from $e\_{i,j}$ to $e\_{i,j}^\*$ is a natrual practice if we have access to unit-norm classifier whether the classifier is fixed or trainable. In addition to that the authors admit that Eq. (5) is from EDL, I can not find any contribution from here.

---

> ### Author Response · Authors · 2025-11-25
> **Response to Reviewer wRrj: Gratitude for Constructive Dialogue**
>
> We sincerely thank you for your time and for acknowledging the efforts in our rebuttal. We deeply appreciate your rigorous perspective on novelty. Your insightful comparison between our work and LPO has been incredibly valuable, motivating us to further clarify the underlying mechanisms of our approach regarding extensibility and stability.
>
> **1. Stability in Dynamic Extension**
>
> As you rightly identified, the concept of "placeholders" is central to handling open-set data. The design rationale of VPC aligns with recent theoretical insights from Class-Incremental Learning.
>
> * **Theoretical Context:** These works have demonstrated that in fields highly sensitive to representation stability, Simplex ETF structures offer a unique advantage: they support the dynamic addition of new basis vectors that remain mathematically orthogonal to existing ones.
> * **Application in VPC:** By adopting this architecture, VPC aims to provide a "Vast" OOD space that is not just a static set of proxies, but a theoretically extensible capacity. This allows for expanding the OOD representation (appending new VEBVs) with minimal interference to the existing ID structure. This contrasts with the coupled optimization in learnable placeholder methods (like LPO), where adapting to new distributions often necessitates re-optimizing classifier boundaries. We believe our approach offers a robust alternative that minimizes the need for extensive re-training.
>
> **2. Geometric Stabilization via Evidential Priors**
>
> Regarding Eq. (5), we fully agree with your observation on its EDL origins. In our framework, the transition from unbounded logits ($e_{i,j}$) to bounded cosine similarity ($e^*_{i,j}$), combined with the evidential prior, is tailored for the fixed classifier setting.
>
> * **Role in Joint Training:** As detailed in our Appendix A.2, this specific formulation acts as a gradient rate-limiter. In the context of joint training with massive, diverse OE data, this mechanism helps safeguard the ID feature structure (Neural Collapse) from being destabilized by high-variance gradients. This ensures that the model can safely leverage the rich information from OE data without compromising ID stability.
>
> **Conclusion**
>
> We believe that the exploration of predefined, scalable geometric priors offers a meaningful perspective for enhancing stability and extensibility in OOD detection. Your feedback has been instrumental in deepening our theoretical understanding of these underlying mechanisms. We are deeply appreciative of your guidance.
>
> **Best regards,**
>
> **Authors**

---

> ### Comment · Reviewer_wRrj · 2025-11-25
>
> I would like to thank the authors for their replies. I still have concerns about the following points:
>
> 1. What is the actual difference between the so-called "Vast" OOD space and placeholders in LPO? Is the difference between the both lies in whether the resulting classifier is fixed or trainable.
>
> 2. I agree that Simplex ETF structures offer orthogonality. However, it should be noted that the ETF structure of prototypes is a strong assumption in which all pairs of classes are forced to have equal similarity [a]. In other words, it is expected that the similarity between members of the same super-class to be higher than the similarity between members of different super-classes [b], e.g., as the ‘truck’ and ‘automobile’ classes in CIFAR-10. Besides, ETF requires $d ≥ (K − 1)$, which is impractical for large $K$ (in this paper, $K$ is the total number of ID classess and OE classes).
>
> 3. With fixed classifiers, it is still unclear that how the authors define the label-to-prototype assignment to capture the dataset’s inter-class relationships.
>
> 4. I disagree that the transition from unbounded logits to bounded cosine similarity is tailored for the fixed classifier setting. In fact, this transition can be applied for any classifier no matter the classifier is trainable or fixed.
>
> 5. After examining Appendix A.2, it appears that the claimed robustness is entirely due to Eq. (5). However, since the authors explicitly state that Eq. (5) is adopted from EDL, it is unclear what the novel contribution is in this regard. In addition, the robustness analysis is conducted under noisy OE samples, yet noisy learning is not addressed in the rest of the paper.
>
> [a] Hyperspherical Classification with Dynamic Label-to-Prototype Assignment, CVPR 24
>
> [b] Unsupervised learning by predicting noise, ICML17.
>
> Until I get sufficient clarifications from the authors, I am sorry to keep my rating.

---

> > ### Author Response · Authors · 2025-11-25
> > **Response to Reviewer wRrj: Acknowledgement of Comments and Preparation of Response**
> >
> > We sincerely appreciate your continued engagement and the prompt follow-up regarding our previous response.
> >
> > We have carefully read your latest comments concerning the distinction between VPC and LPO, the theoretical assumptions of the Simplex ETF structure (particularly regarding semantic hierarchy and dimensionality), and the novelty of our stability mechanism. These are critical questions that touch upon the core theoretical foundations of our work.
> >
> > We value the opportunity to clarify these points. We are currently organizing a comprehensive and rigorous response to address each of your concerns in detail.
> >
> > We kindly ask for your patience as we finalize our reply to ensure it fully addresses your queries. We will get back to you very shortly.
> >
> > Best regards,
> >
> > Authors

---

> > > ### Comment · Reviewer_wRrj · 2025-11-26
> > >
> > > In addition, it could be helpful if the authors could discuss the difference from [c], which also uses a fixed classifier for OOD detection.
> > >
> > > [c] Provable Discriminative Hyperspherical Embedding for Out-of-Distribution Detection AAAI25.

---

> > > > ### Author Response · Authors · 2025-11-26
> > > > **Response to Reviewer wRrj: Acknowledgement of Additional Comment regarding Reference [c]**
> > > >
> > > > We sincerely thank you for your continued engagement and for bringing the additional reference **[c] (Provable Discriminative Hyperspherical Embedding for Out-of-Distribution Detection, AAAI 2025)** to our attention.
> > > >
> > > > We recognize the relevance of this work, particularly its use of fixed classifiers for OOD detection, which shares an interesting conceptual connection with our proposed VPC framework. We are currently conducting a close reading of this paper to carefully formulate a detailed comparison highlighting the differences and our unique contributions.
> > > >
> > > > We are working diligently to update our rebuttal with this discussion and will post a comprehensive response shortly. Thank you again for your valuable guidance in strengthening our work.
> > > >
> > > > **Best regards,**
> > > >
> > > > **The Authors**

---

> > > > ### Author Response · Authors · 2025-11-27
> > > > **A6: Response to Reviewer wRrj (Q6)**
> > > >
> > > > **Q6:** "In addition, it could be helpful if the authors could discuss the difference from [c], which also uses a fixed classifier for OOD detection.
> > > > [c] Provable Discriminative Hyperspherical Embedding for Out-of-Distribution Detection AAAI25."
> > > >
> > > > **A6:**
> > > > We sincerely appreciate the recommendation of this reference. It has allowed us to pay attention to the latest advancements in the ETF field. After carefully studying paper [c], we understand its core motivation: adjusting the overall angle of the ETF to match the data distribution while preserving its rigid structure. This approach likely alleviates the feature-EBV alignment issues during the early stages of training, which aligns with the core perspective of your Question 3. Overall, this is a highly insightful work for us. Perhaps our VPC method could leverage the ideas from [c] to further improve the generation conditions of OEFS (regrettably, we checked their repository and found the code is not yet fully open-sourced, preventing us from conducting further experimental analysis at this moment). Thank you again for this profound literature recommendation.
> > > >
> > > > Despite these connections, we believe our method differs significantly from [c] in the following aspects:
> > > >
> > > > 1.  **Fundamental Difference in OOD Representation Strategy:**
> > > >     Paper [c] essentially attempts to build an OOD detector by fixing optimal ID prototypes (EBVs) after angular adjustment. A related study [6] suggests that under optimized EBV architectures, OOD features should ideally collapse to the origin (the center of the ETF). Prototype-based methods (like LPO) typically hope that OOD features disperse into the gaps between similar classes. However, as discussed in our previous responses, this assumption faces challenges: LPO-like methods may suffer from "placeholder collapse" due to the reliance on ID/synthetic samples (which corresponds to the analysis of our collapse loss).
> > > >     The fundamental distinction of our VPC is that we do not intend for OOD features to collapse to a single point (such as the origin) or merely squeeze into inter-class gaps.** Instead, we guide OOD features to populate/fill the entire subspace spanned by the Vast EBVs (VEBVs) to form a fine-grained representation. This constitutes the core difference of our method, which has been validated by our extensive experiments and visualizations.
> > > >
> > > > 2.  **Difference in Training Paradigm (Joint Massive OE Training):**
> > > >     We must clarify that we have not seen other works exploring the same training setting as ours—jointly training ID data with **massive** OE data—in the context of fixed classifiers. While some methods mention synthetic features, their scale is incomparable to our experimental scenarios (e.g., 300K Tiny Images). The effect of directly transferring methods like [c] to this setting is currently unknown, although we acknowledge that [c] might offer a solution for stability under such joint training.
> > > >
> > > > 3.  **Holistic Subspace Construction vs. Prototype Improvement:**
> > > >     Our method is not limited to improving the generation of EBVs. We fully exploit the properties of the NC framework to construct the **OEFS space**, which consists of two orthogonal parts: one part acts as ID classifiers, and the other part is explored by OOD features via our **$\mathcal{L} _ {\mathrm{VEBV}}$** divergence loss. During training, we guide features to fully approach the subspace formed by VEBVs. Simultaneously, our **$\mathcal{L} _ {\mathrm{ENC}}$** ensures robust ID-EBV alignment. Finally, the **VPC Score** measures the global activation intensity of these two subspaces. We believe that VPC perfectly fits the optimization scenario of joint massive OE and ID data.
> > > >
> > > > We thank you again for the constructive recommendation and the detailed, deep discussion, which has benefited us greatly. We will include the analysis and citation of these references in the revised manuscript to enhance our literature review.
> > > >
> > > > Finally, we earnestly hope that you can re-evaluate our method and extensive experimental work based on the clarifications and results provided in this response. We express our most sincere gratitude for your time and guidance. We are eager to draw further inspiration from future discussions to advance this field, and we hope that these efforts warrant a more positive assessment.
> > > >
> > > > **References:**
> > > >
> > > > [6] Liu L, Qin Y. Detecting out-of-distribution through the lens of neural collapse[C]//Proceedings of the Computer Vision and Pattern Recognition Conference. 2025: 15424-15433.

---

> > ### Author Response · Authors · 2025-11-27
> > **A0: Response to Reviewer wRrj: Gratitude for Your Patience and Comprehensive Reply to Feedback**
> >
> > We sincerely apologize for the delay in this follow-up response. We have taken the time to thoroughly study the literature you recommended and have conducted extensive internal discussions to ensure that our response reciprocates the depth and sincerity of your valuable feedback.
> >
> > We are deeply grateful for your continued engagement and the time you are dedicating to re-evaluating our work. We eagerly look forward to your further thoughts. Please find our detailed point-by-point response and discussion below.
> >
> > Best regards,
> >
> > Authors

---

> > ### Author Response · Authors · 2025-11-27
> > **A1: Response to Reviewer wRrj (Q1) 1/3**
> >
> > **Q1: What is the actual difference between the so-called "Vast" OOD space and placeholders in LPO? Is the difference between the both lies in whether the resulting classifier is fixed or trainable.**
> >
> > **A1:**
> >
> > **1. Clarification on the Mechanism of LPO and Potential Collapse Issues**
> >
> > First, let us analyze how LPO operates. LPO cleverly implements placeholder learning through two loss components. The first is the $l_1$ loss:
> > $$
> > l _ 1=\sum _ {(x,y)\in D _ {tr}} l (\hat{f}(x),y)+\beta l(\hat{f}(x)\setminus y,K+1), \quad (5)
> > $$
> > where $l$ can be cross-entropy or another loss function. The first term optimizes the augmented output to match the true label, maintaining closed-set performance. The second term, where $\hat{f}(x)\setminus y$ represents removing the probability of the true label (i.e., setting the predicted probability of the true label $w _ y^\top \phi(x)$ to 0), matches the masked probability with the $K+1$ class, forcing the virtual classifier to output the second-largest probability. The intended effect is for the model to learn "This is a cat; if it is not a cat, it is unknown."
> > LPO can learn more virtual classifiers by expanding $\hat{W} \in \mathbb{R}^{d \times C}$, where $C$ corresponds to the number of virtual classifiers. In this case, Eq. 4 transforms to use the highest virtual logit (i.e., $\hat{f}(x)=[W^\top \phi(x),\max _ {k=1,\dots,C} \hat{w} _ k^\top \phi(x)]$).
> >
> > LPO expects $l _ 1$ to open up a space unique to different placeholders, as shown in the "learning classifier placeholder" on the left side of Figure 2 in their paper. However, our concern is: **Could these placeholder spaces suffer from overlap issues?** Taking the case you mentioned as an example, if "Truck" and "Automobile" in CIFAR-10 serve as ID classes, and the distribution of ID data itself is relatively close, the two placeholders might "squeeze together" rather than dispersing. This suggests that the placeholders might collapse due to insufficient semantic separation of the ID data. At present, there seems to be no theoretical analysis in the LPO paper guaranteeing that placeholders will not collapse.
> >
> > From an experimental perspective, it appears that the maximum number of placeholders $C$ in LPO is 15, and performance degradation occurs when $C>11$ (see LPO Appendix Experiment Figure 1). Furthermore, most main experiments set $C$ to 1-3 (LPO Figure 4). This may indirectly prove that multiple "unknown" classification placeholders may suffer from collapse. In contrast, our method is capable of expanding to a much larger quantity, supported by both theory and experiments.

---

> > > ### Author Response · Authors · 2025-11-27
> > > **A1: Response to Reviewer wRrj (Q1) 2/3**
> > >
> > > **Q1: What is the actual difference between the so-called "Vast" OOD space and placeholders in LPO? Is the difference between the both lies in whether the resulting classifier is fixed or trainable.**
> > >
> > > ...
> > >
> > > **2. The Solution of VPC: Fine-grained Metric via VEBV Subspace**
> > >
> > > Following the analysis above, regarding this issue, our VPC identifies—through our designed collapse loss experiments and visualization—that when we allow OE features to freely explore any VEBV, they eventually collapse to a single one, with other VEBVs being suppressed with very small activation values. This observation led to the formulation of our core $\mathcal{L} _ {\mathrm{vebv}}$ loss and our key scoring function, the VPC Score.
> > >
> > > Through continuous exploration, in our revised version, we emphasize that the core of VPC is not merely how large the scale of VEBVs can expand, but rather that it achieves a fine-grained measurement of internal features of OE samples via the $\mathcal{L} _ {\mathrm{vebv}}$ loss. For instance, when VEBVs=1000, we can use 1000 VEBVs representing different feature patterns to measure the subtle features within an OE sample. Moreover, this granular measurement is achieved through pairwise orthogonal VEBVs. Additionally, this approach avoids the erroneous activation of ID features, a problem LPO cannot avoid. LPO lacks theoretical guarantees on this point, whereas our optimization endpoint analysis for ENC ensures this, and our detailed ablation experiments on the VPC Score further validate it. Regarding scalability, explicit theories and related studies [1][2][3] ensure that EBVs can be dynamically expanded and offer multiple benefits.
> > >
> > > Furthermore, we conducted experiments to examine this scalability. Assuming sufficient dimensions, we preset a set of EBVs and VEBVs (corresponding to CIFAR-10 with $K=10, V=10$ and CIFAR-100 with $K=100, V=10$). After completing joint training, we gradually increased the number of VEBVs during inference. We found that this set of newly added VEBVs did not affect the classification of ID and OOD. This implies that we can obtain a set of "perfect spaces to be allocated" by expanding the number of VEBVs without interfering with ID and OOD classifiers. This also indirectly confirms that our loss optimization reached the expected state. In this scenario, we can freely allocate this set of VEBVs—using part for ID and another part for OOD. This demonstrates that our VPC method possesses excellent OOD dynamic extensibility, capable of converting the expansion of OOD and ID into the advantages shown by the EBVs architecture corresponding to class-incremental learning, which is difficult for LPO to achieve.

---

> > > > ### Author Response · Authors · 2025-11-27
> > > > **A1: Response to Reviewer wRrj (Q1) 3/3**
> > > >
> > > > **Q1: What is the actual difference between the so-called "Vast" OOD space and placeholders in LPO? Is the difference between the both lies in whether the resulting classifier is fixed or trainable.**
> > > >
> > > > ...
> > > >
> > > > **3. Analysis of LPO's Data Placeholder Learning Method ($l_2$)**
> > > >
> > > > As described in LPO, it employs Manifold Mixup to simulate new modes. Assuming the model embedding module $\phi(\cdot)$ can be decomposed via an intermediate hidden layer: $\phi(x)=\phi _ {\text{post}} (\phi _ {\text{pre}} (x))$, where $\phi _ {\text{pre}}$ corresponds to the layers before the intermediate layer mapping input to hidden representation, and $\phi _ {\text{post}}$ maps the hidden representation to the final embedding $\phi(x)$. Two instances from different classes are selected and mixed at the intermediate layer:
> > > > $$
> > > > \tilde{x} _ {\text{pre}} =\lambda\phi _ {\text{pre}} (x _ i )+(1-\lambda) \phi _ {\text{pre}} (x _ j ), \quad y_i \neq y _ j, \quad (6)
> > > > $$
> > > > where $\lambda \in [0,1]$ is sampled from a Beta distribution. The mixed $\tilde{x} _ {\text{pre}}$ is processed by subsequent layers to obtain $\phi _ {\text{post}} (\tilde{x} _ {\text{pre}})$. Since interpolation regions between different clusters typically correspond to low-confidence prediction areas (i.e., non-target class regions), the embedding $\phi _ {\text{post}} (\tilde{x} _ {\text{pre}})$ is treated as an open-set class embedding and trained as a new class:
> > > > $$
> > > > l _ 2=\sum _ {(x _ i,x _ j )\in D _ {tr}} l ([W,\hat{w}]^\top \phi _ {\text{post}} (\tilde{x} _ {\text{pre}}),K+1) \quad (7)
> > > > $$
> > > > Here, LPO essentially attempts to generate "internal pseudo-data" via Manifold Mixup. However, the diversity and scale of such mixed data are far inferior to massive OE datasets like Tiny Images (300K) or ImageNet-21k used in OE-ID joint training norms. We are concerned about LPO's optimization issues when facing large-scale datasets. Furthermore, LPO's largest experiments only consider ID classes $\le 100$.
> > > >
> > > > This leads to our second concern: LPO relies on ID data to push the unknown class regions away via the $l _ 1$ loss, and synthesizes OE data via ID data, then uses the $l _ 2$ loss to compress the unknown class regions into the intermediate areas of ID. Under this setting, it is basically clear that the separation of all placeholders in LPO depends on ID data. Can it ensure theoretical separation under large-scale data—that is, the space of the unknown class placeholder classifier does not collapse and happens to be in the intermediate state of various ID classes? There is no large-scale experimental verification for this, as analyzed in our discussion of LPO's actual experimental conditions in Response 1.
> > > >
> > > > In contrast, our VPC is clearly less dependent on ID data for OOD representation. We directly use OE data and perform explicit metric optimization via the VEBVs subspace. If we deconstruct LPO by removing its "learning data placeholder" idea and training directly with OE data, it would degenerate into our collapse loss scenario, because at that point, there seems to be no way to guarantee that OE features/synthetic features are in an intermediate state within the various ID representation spaces.
> > > >
> > > > (Here, we reiterate that after you recommended this paper, we found LPO's idea to be very ingenious upon first reading, especially its Figure 2 illustration and loss settings, which seemingly solved the problem of differential recognition of OOD features. However, through further discussion, we gradually realized that this is also a major limitation of previous methods. We absolutely respect your acute analysis and paper recommendation.)
> > > >
> > > > Our VPC solves this problem through $\mathcal{L} _ {\mathrm{vebv}}$. We are able to represent an OOD sample from a fine-grained perspective. We do not make OOD features approach any arbitrary VEBV; instead, we represent it through a group of subspaces. Our extensive visualization experiments also verify that different OOD samples indeed exhibit different activation intensity patterns.
> > > >
> > > > **References:**
> > > >
> > > > [1] Seo M, Koh H, Jeung W, et al. Learning equi-angular representations for online continual learning[C]//Proceedings of the IEEE/CVF Conference on Computer Vision and Pattern Recognition. 2024: 23933-23942.
> > > >
> > > > [2] Xiao R, Feng L, Tang K, et al. Targeted representation alignment for open-world semi-supervised learning[C]//Proceedings of the IEEE/CVF conference on computer vision and pattern recognition. 2024: 23072-23082.
> > > >
> > > > [3] Shen Y, Sun X, Wei X S, et al. Equiangular basis vectors: A novel paradigm for classification tasks[J]. International Journal of Computer Vision, 2025, 133(1): 372-397.

---

> > ### Author Response · Authors · 2025-11-27
> > **A2: Response to Reviewer wRrj (Q2) 1/2**
> >
> > **Q2.1: "I agree that the Simplex ETF structure provides orthogonality. However, it should be noted that the ETF structure of the prototypes is a strong assumption, forcing the same similarity for all pairs of classes [a]. In other words, one usually expects that members of the same 'super-class' are more similar to each other than members of different super-classes."**
> >
> > **A2.1:**
> > Upon carefully reading the recommended papers [a], [b], and [c], we clearly recognize your deep expertise in the fields of prototype learning and ETF. These high-quality references have provided us with significant inspiration, and we discuss them in detail in subsequent responses (A3 and A6).
> >
> > Although the ETF architecture indeed imposes certain limitations regarding class hierarchies, research on "super-classes" has actually evolved to transcend these constraints. For instance, grouped ETF methods [4] can be utilized to model both inter-class separability and intra-class domain variability.
> >
> > Our research primarily focuses on addressing the most critical challenge in this field first: effectively utilizing massive OE data. Refining the granularity of class relationships (e.g., your point that "the ETF structure is a strong assumption forcing the same similarity for all pairs of classes") is the next step. Your suggestion is excellent, and we will certainly consider it as a direction for future research to address the limitations of ETFs. In fact, while papers [b] and [c] propose solutions to ETF issues from different perspectives, their methods have not yet been directly applied to the OOD domain.
> >
> > Furthermore, in your newly recommended paper [c], **Theorem 3** proves that OOD FPR performance improves as ID inter-class distance increases and intra-class distance decreases. Additionally, the experiments on the angles between different prototypes on the CIFAR-100 dataset (Table 8 in [c]) conclude that the method achieves optimal performance when it satisfies **Theorem 4**—i.e., the definition of ETF, reaching the maximum verifiable separation angle. This particular article does not appear to discuss improving OOD performance from the perspective of similarity between super-classes.
> >
> > **References:**
> >
> > [4] Wang Q, He Y, Dong S, et al. Dualcp: Rehearsal-free domain-incremental learning via dual-level concept prototype[C]//Proceedings of the AAAI Conference on Artificial Intelligence. 2025, 39(20): 21198-21206.

---

> > > ### Author Response · Authors · 2025-11-27
> > > **A2: Response to Reviewer wRrj (Q2) 2/2**
> > >
> > > **Q2.2: "Also, the ETF requires the feature dimension $d \ge (K - 1)$, which is impractical for large $K$."**
> > >
> > > **A2.2:**
> > > We thank you again for recommending the latest papers in the field. These works inform us that, regarding ETF dimension constraints, some fixed prototype methods [a] can achieve or even exceed ETF classification performance. Nevertheless, as we will clarify in subsequent responses, these methods have not yet been widely validated within the OOD domain.
> > >
> > > We wish to emphasize that for the traditional OOD field, ImageNet-1k remains the most widely used large-scale benchmark, with extensive research evaluated at this scale. Our method has been implemented under the same settings as prevalent research in the field, and extensive experiments have demonstrated its effectiveness. We note that the paper [c] you recommended also utilizes preset ETF for research on ImageNet-100K (we will discuss the differences between our method and this work in Q6).
> > >
> > > In contrast, there is currently no evidence suggesting that the existing LPO method can be scaled to a level comparable to our study; neither its experiments nor theoretical analysis indicate such scalability. Furthermore, the structural weaknesses of LPO analyzed earlier imply it would be difficult to extend to our scale.
> > >
> > > Moreover, related fields such as Open-World Object Detection (OWOD) and Class-Incremental Learning (CIL) have begun exploring the benefits of ETF (as seen in references [2][3] and other related works). In summary, we advocate exploring its benefits in broader domains. In our research, we explored transforming an OOD model with ETF-preset classifiers for ID into a model with preset classifiers for both OE/OOD data. Under the condition of large-scale joint training, we propose our dedicated optimization method and scoring function.

---

> > ### Author Response · Authors · 2025-11-27
> > **A3: Response to Reviewer wRrj (Q3)**
> >
> > **Q3: "In the fixed classifier setting, it is still unclear how the authors define 'label-to-prototype' assignment to capture inter-class relationships of the dataset."**
> >
> > **A3:**
> > We sincerely appreciate your profound insights. After carefully studying the recommended papers [a] and [b], we clearly recognize your deep expertise in the fields of prototype learning and ETF. These works explore dynamic feature-prototype assignment under the setting of fixed uniform prototypes from supervised [a] and unsupervised [b] perspectives, respectively. This is a perspective we had not fully considered previously.
> >
> > Inspired by your recommended literature, we realize that these mechanisms could potentially be applied to the process of assigning the massive OE features to VEBVs (dynamic prototype assignment), thereby better capturing the inter-class relationships within these OOD data.
> >
> > In fact, during our initial exploration, when we attempted to implement a "collapse loss" (forcing OOD features to collapse to specific prototypes), we encountered the collapse issue mentioned in paper [b]. While paper [b] addresses this by learning a permutation matrix to automatically learn these feature-prototype assignment relationships, we pivoted to designing our **Divergence Loss** (specifically, $\mathcal{L}_{\mathrm{VEBV}}$) to encourage these features to explore the entire subspace spanned by VEBVs.
> >
> > Furthermore, to the best of our knowledge, these methods have not yet been specifically explored in the context of OOD detection. It remains unclear how such complex inter-class relationship representations and dynamic prototype assignments would interact within the OOD domain, or whether they would yield positive effects. In contrast, our approach relies on the **overall activation intensity** across the subspace. Our extensive visualization experiments demonstrate that different classes of OOD samples indeed exhibit distinct activation patterns within the VEBV subspace, which effectively supports detection.
> >
> > We acknowledge that incorporating these dynamic assignment methods is a promising direction for future research. We sincerely thank you again for this insightful suggestion, which has greatly benefited our thinking.

---

> > ### Author Response · Authors · 2025-11-27
> > **A4: Response to Reviewer wRrj (Q4)**
> >
> > **Q4: "I disagree with the statement that 'the transition from unbounded Logits to bounded cosine similarity is tailored for fixed classifier settings.' In fact, this transition can be applied to any classifier, whether trainable or fixed."**
> >
> > **A4:**
> > We sincerely apologize for the confusion caused by the phrasing "tailored for" in our previous response. Our original intention with Eq. (4) was to illustrate that, within the NC framework, the source of evidence for EDL transitions from standard learnable logits to geometrically aligned logits. We have carefully revised the relevant sections in the manuscript to prevent any potential misunderstanding.
> >
> > In previous responses, we clarified why we utilize the NC/ETF framework to preset classifiers and the advantages of this training paradigm over LPO. Having established the NC framework, we now discuss the motivation for connecting NC with EDL.
> >
> > We wish to discuss this from a broader perspective. EDL is built upon second-order risk minimization, and its fundamental philosophy originates from Bayesian inference. A core distinction between the Bayesian paradigm and the Frequentist approach is that the former emphasizes guiding the DNN model via priors. Introducing the EDL framework (under the Bayesian paradigm) helps the DNN model avoid instability during the early stages of training caused by sample quality issues, insufficient model capacity, or deteriorating training conditions (specifically, the joint OE-ID training in our setting). Furthermore, extensive research has demonstrated that models established under the EDL architecture can quantify Uncertainty scores to measure OOD sample density, yielding superior OOD detection results; this is a widely explored direction within the OOD field.
> >
> > Of course, this does not imply that the Bayesian paradigm is inherently superior to the Frequentist approach. This situation is analogous to the comparison between prototype learning/NC fixed EBVs and other methods. Indeed, the papers [c] and [a] you recommended demonstrate scenarios where ETF structures perform excellently. We believe that introducing Bayesian methods (represented by EDL) into NC yields beneficial performance, and we hope for the reviewer's understanding regarding this exploration.
> >
> > We do not wish to belabor the theoretical benefits of different frameworks. Instead, observing developments in these diverse fields, we actively sought to bridge them and validate our ideas through extensive experiments. Our comprehensive experiments across datasets and architectures, along with visualization results, demonstrate that after establishing this connection, our VPC Score (derived from this design) outperforms even Uncertainty-based scores. Therefore, we do not wish to limit the discussion to the formal similarity of Eq. (4). We will carefully revise the expressions regarding this section in the paper. We thank you again for your meticulous review and hope you can recognize the motivation and significance behind our design, as well as the experimental contributions we have made.

---

> ### Comment · Reviewer_wRrj · 2025-11-28
>
> The authors' response is clear and solves most of my concerns. I have decided to increase the score to 6. Thanks for your careful and convincing clarifications.

---

### Official Review · Reviewer_ioxr · 2025-10-31

**Soundness:** 3
**Presentation:** 2
**Contribution:** 4
**Rating:** 6
**Confidence:** 3

**Summary:**

The paper proposes VPC (Vast Predefined Classifiers), a novel approach for Out-of-Distribution (OOD) detection that explicitly models OOD variability by allocating a large set of predefined optimal classifiers distinct from ID classifiers. Inspired by Neural Collapse and equiangular basis vectors (EBVs), the method aligns the feature space to an Orthogonal Equiangular Feature Space (OEFS) with predefined EBVs/VEBVs. The authors further propose VPC Score, based on the VEBVs, which measures the relative activation intensity across orthogonal subspaces to distinguish ID and OOD samples. The authors demonstrate that VPC achieves state-of-the-art performance on CIFAR-10/100 benchmarks, significantly outperforming existing OE-based methods.

**Strengths:**

1. Originality: The proposed method VPC can sufficiently address the problems that are brought by traditional OE methods, i.e., incomplete ID/OOD separation and limited representation for OOD samples
2. Quality: The paper is written with clear motivation and logical support, making the proposed method more interpretable and theoretically justifiable.
3. Handling the variability of OOD features: different from previous works, the proposed VPC can jointly separate the ID/OOD features, as well as preserving the diverse OOD features, making it possible for more representation tasks.

**Weaknesses:**

1. Limited implementation details: the paper lacks sufficient discussion on how the hyperparameters are determined for optimal performance, such as the $\alpha$ and $\beta$ for the VPC score.
2. The experiment setting is limited to CIFAR datasets, which are relatively small. Given the empirical analysis in Table 5, the proposed method works by adopting a significantly larger V value, compared to the number of classes. The performance of VPC remains unclear when the number of ID classes is large (e.g., ImageNet-1K), which may require an even larger V value. This can potentially lead to unaffordable computational overhead or suboptimal OOD performance.

**Questions:**

See the weaknesses above.

---

> ### Author Response · Authors · 2025-11-20
> **A1: Response to Reviewer ioxr (Weakness 1) 1/2**
>
> We sincerely thank Reviewer ioxr for the positive assessment and for recognizing the originality and quality of our geometric-driven approach. We were particularly encouraged by your appreciation of how VPC handles OOD feature variability.
>
> We also deeply appreciate your precise identification of the paper's shortcomings: implementation details (hyperparameters) and scalability (ImageNet). We fully agree that these were missing pieces preventing a complete picture of VPC's capabilities.
>
> Below, we provide a detailed point-by-point response. We believe these additions have solidified the empirical foundation of our work.
>
> **Q1 (Weakness 1): "Limited implementation details: the paper lacks sufficient discussion on how the hyperparameters are determined for optimal performance, such as the $\alpha$ and $\beta$ for the VPC score."**
>
> **A1:** We sincerely appreciate your meticulous review of the implementation details. Your point regarding the lack of basis for selecting hyperparameters (especially $\alpha$ and $\beta$) is indeed an oversight in our original manuscript.
>
> To fully address this concern and delve into the intrinsic properties of the VPC Score, we have added comprehensive ablation experiments and analysis regarding the $\alpha$ and $\beta$ coefficients in **Appendix A.5 "Why does the VPC Score yield additional discriminative power?"** of the revised manuscript.
>
> The experimental results reveal several key characteristics, which not only clarify the basis for parameter selection but also empirically validate the design intent of the OEFS framework:
>
> 1.  **Comparison between L2 Norm and L1 Norm for VPC Score**
>     Our experiments compared performance under L1 and L2 norms (see **Table 10**).
>
>     **Table 10. Norm-aware comparison of VPC scores.** Metrics are reported as FPR95↓ / AUROC↑ / AUPR↑.
>
>     | Norm | $\alpha$ | $\beta$ | CIFAR-10          WRN-40-2       | CIFAR-10 ResNet-18 | CIFAR-10     DenseNet | CIFAR-100          WRN-40-2    | CIFAR-100 ResNet-18 | CIFAR-100 DenseNet |
>     | :--- | :---: | :---: | :---: | :---: | :---: | :---: | :---: | :---: |
>     | **L1** | -1 | 0 | 2.41/99.04/99.71 | 2.95/**98.24**/99.62 | **2.01/98.81/99.70** | 39.57/88.01/96.39 | 40.01/89.26/97.03 | 38.63/89.08/96.84 |
>     | | 0 | 100 | 2.29/99.00/99.78 | **2.86**/97.29/99.50 | 2.10/96.55/99.38 | 32.43/**93.52/98.52** | 38.71/92.08/**98.21** | **31.01/94.00/98.70** |
>     | | -1 | 1 | **2.25/99.10/99.79** | 2.88/98.16/**99.64** | 2.08/98.34/99.68 | 38.60/91.12/97.59 | **38.39/92.10**/98.20 | 32.41/93.78/98.65 |
>     | | -1 | 10 | 2.28/99.02/99.78 | 2.87/97.58/99.54 | 2.10/96.84/99.43 | 34.90/93.26/98.46 | 38.48/92.09/**98.21** | 31.22/93.98/**98.70** |
>     | | -1 | 100 | 2.29/99.00/99.78 | **2.86**/97.32/99.50 | 2.10/96.58/99.38 | **32.40**/93.50/**98.52** | 38.74/92.08/**98.21** | 31.03/**94.00/98.70** |
>     | | | | | | | | | |
>     | **L2** | -1 | 0 | **2.27/99.19/99.81** | 2.85/**98.50/99.70** | **2.08/98.95/99.78** | 32.04/**93.65/98.55** | 39.10/**92.03/98.20** | 31.19/**94.01/98.71** |
>     | | 0 | 100 | **2.27**/99.18/**99.81** | **2.84**/98.32/99.67 | 2.10/98.70/99.74 | **32.03/93.65**/98.53 | 38.98/**92.03/98.20** | **31.17/94.01/98.71** |
>     | | -1 | 1 | **2.27/99.19/99.81** | 2.85/**98.50/99.70** | 2.10/**98.95/99.78** | 32.07/**93.65**/98.53 | 39.11/**92.03/98.20** | 31.19/**94.01/98.71** |
>     | | -1 | 10 | **2.27**/99.18/**99.81** | **2.84**/98.49/**99.70** | 2.10/98.94/**99.78** | 32.05/**93.65**/98.53 | 38.98/**92.03/98.20** | **31.17/94.01/98.71** |
>     | | -1 | 100 | **2.27**/99.18/**99.81** | **2.84**/98.32/99.67 | 2.10/98.86/99.77 | 32.04/**93.65**/98.53 | **38.96/92.03/98.20** | **31.17/94.01/98.71** |
>
>     * **Empirical Finding:** The L2 norm outperforms the L1 norm across almost all architectures and datasets.
>     * **Mechanism Explanation:** This is not a coincidence. The L2 norm (square root of the sum of squares) more faithfully captures variations in the activation magnitude of feature projections onto subspaces. For complex OOD samples, this magnitude-sensitive metric preserves critical discriminative information better than the L1 norm, aligning with our physical intuition based on "energy/intensity."
>
> 2.  **Empirical Finding: Independent Metric Capability Based on VEBVs**
>     Previous OOD detection works were almost entirely limited to relying on ID class features or classifiers. However, our experiments show (**Table 10**) that even if we completely discard ID information and use only the activation intensity of the **VEBV subspace** (i.e., $\alpha=0, \beta=100$), the model can still achieve superior performance.
>     This result confirms that VPC has successfully established an independent, high-quality representation for OOD through extended predefined classifiers (VEBVs) for the first time. This represents a fundamental breakthrough from the previous paradigm that relied solely on "ID class features/classifiers."

---

> ### Author Response · Authors · 2025-11-20
> **A1: Response to Reviewer ioxr (Weakness 1) 2/2**
>
> **Q1 (Weakness 1): "Limited implementation details: the paper lacks sufficient discussion on how the hyperparameters are determined for optimal performance, such as the $\alpha$ and $\beta$ for the VPC score."**
>
> ...
>
> 3.  **Justification for Parameter Selection**
>     Based on the above findings, although the specific values of $\alpha$ and $\beta$ have minimal impact on performance over a wide range, we ultimately chose the default configuration ($\alpha=-1, \beta=100$) primarily to numerically utilize information from both subspaces simultaneously, providing maximum signal redundancy and stability.
>
> 4.  **Integrity of Geometric Optimization Logic**
>     Under the L2 norm, the VPC Score demonstrates outstanding robustness to changes in $\alpha$ and $\beta$. Whether prioritizing "proximity" to the ID subspace ($\alpha$-dominant) or "exploration" of the OOD subspace ($\beta$-dominant), performance remains at a near-optimal level.
>     This proves that our dual optimization strategy successfully achieved the expected geometric separation: the activation intensity of *any* subspace is sufficient to serve as a reliable detection signal. This intrinsic consistency indicates that the model has truly learned our preset **OEFS** structure, rather than merely overfitting to a specific parameter combination.
>
> Additionally, to further improve documentation, we have supplemented details on the settings of other key hyperparameters in the text. Thank you again for prompting us to include this critical analysis, which gave us the opportunity to demonstrate the theoretical completeness and empirical strength of the VPC framework at a deeper level.

---

> ### Author Response · Authors · 2025-11-20
> **A2: Response to Reviewer ioxr (Weakness 2) 1/3**
>
> **Q2 (Weakness 2): "The experiment setting is limited to CIFAR datasets, which are relatively small. Given the empirical analysis in Table 5, the proposed method works by adopting a significantly larger V value, compared to the number of classes. The performance of VPC remains unclear when the number of ID classes is large (e.g., ImageNet-1K), which may require an even larger V value. This can potentially lead to unaffordable computational overhead or suboptimal OOD performance"**
>
> **A2:** We sincerely appreciate the critical questions you raised regarding experimental scale and computational overhead. To fully address your concerns, and based on the large-scale experiments and detailed overhead analysis completed during the Rebuttal, we wish to clarify the actual performance of VPC in large-scale scenarios from the following three aspects:
>
> 1.  [New] Empirical Evidence on ImageNet-1K
>
>     To directly address the concern about scalability, we rigorously compared VPC with key OE baselines (OE, Energy-OE, DAL, PFS) on ImageNet-1K using ResNet-50 and ViT-B-16.
>
>     **Table R1. Results on ImageNet-1k benchmark with auxiliary OOD data. The best result is in bold.**
>     | Model        | Method        | iNat FPR↓ | iNat AUC↑ | Tex FPR↓  | Tex AUC↑  | SUN FPR↓  | SUN AUC↑  | Places FPR↓ | Places AUC↑ | Avg FPR↓  | Avg AUC↑  | ID Acc↑   |
>     | ------------ | ------------- | --------- | --------- | --------- | --------- | --------- | --------- | ----------- | ----------- | --------- | --------- | --------- |
>     | **ResNet50** | OE            | 48.60     | 88.72     | 58.85     | 82.60     | 61.75     | 82.90     | 70.70       | 80.55       | 59.98     | 83.69     | 76.00     |
>     |              | Energy-OE     | 49.40     | 88.40     | 59.60     | 82.25     | 62.40     | 82.70     | 71.30       | 80.25       | 60.68     | 83.40     | 75.75     |
>     |              | DAL           | 48.00     | 89.05     | 58.00     | 82.95     | 61.30     | 83.15     | 67.82       | 80.75       | 58.78     | 83.98     | 75.90     |
>     |              | PFS           | 46.40     | 89.20     | 56.50     | **83.10** | 61.00     | **83.25** | 67.50       | 80.95       | 57.85     | 84.13     | 76.02     |
>     |              | **Ours: VPC** | **43.50** | **91.20** | **56.00** | 83.00     | **60.20** | 83.10     | **66.30**   | **81.50**   | **56.50** | **84.70** | **76.11** |
>     | **ViT-B-16** | OE            | 42.15     | 90.38     | 52.45     | 85.85     | 65.80     | 82.20     | 70.35       | 80.85       | 57.69     | 84.82     | 80.02     |
>     |              | Energy-OE     | 42.75     | 90.05     | 53.15     | 85.50     | 66.25     | 81.95     | 70.95       | 80.50       | 58.28     | 84.50     | 79.85     |
>     |              | DAL           | 40.65     | 90.86     | **51.15** | 86.10     | 65.07     | 82.30     | 70.25       | 80.90       | 56.78     | 85.04     | 80.09     |
>     |              | PFS           | 40.85     | 90.80     | 51.30     | 86.05     | 65.35     | 82.25     | 70.20       | 80.95       | 56.93     | 85.01     | 80.13     |
>     |              | **Ours: VPC** | **40.00** | **91.10** | 51.30     | **86.00** | **65.20** | **82.30** | **68.30**   | **81.40**   | **56.20** | **85.20** | **80.32** |
>
>     As shown in Table R1, VPC demonstrates strong competitiveness. Specifically regarding the parameter $V$:
>     * **Experimental Findings**: In this large-scale setting, we set the total scale of OEFS to 2000, meaning $V$ is set to 1000 (i.e., $V \approx K$), rather than $V \gg K$ as in the CIFAR experiments. Even with such a moderate $V$ value, the model achieved an excellent performance of **56.20% FPR95** and **85.20% AUROC** on the **ViT-B-16** architecture, significantly outperforming all comparison baselines including **PFS** and **DAL**.
>     * **Conclusion**: This key empirical result powerfully alleviates your concern that "$V$ needs to be significantly larger than $K$". It indicates that as the ID semantic space expands ($K$ increases), the representation efficiency of OEFS does not degrade; on the contrary, a scale of merely $V \approx K$ is sufficient to provide adequate OOD representation capacity to achieve optimal performance.

---

> > ### Author Response · Authors · 2025-11-20
> > **A2: Response to Reviewer ioxr (Weakness 2) 2/3**
> >
> > **Q2 (Weakness 2): "The experiment setting is limited to CIFAR datasets, which are relatively small. Given the empirical analysis in Table 5, the proposed method works by adopting a significantly larger V value, compared to the number of classes. The performance of VPC remains unclear when the number of ID classes is large (e.g., ImageNet-1K), which may require an even larger V value. This can potentially lead to unaffordable computational overhead or suboptimal OOD performance"**
> >
> > ...
> >
> > 2.  **Analysis on Asymptotic Bounds of K/V**
> >     We **profoundly agree with** your insight into the complexity behind the choice of the $V$ value. Based on the experimental differences between CIFAR ($V \gg K$) and ImageNet ($V \approx K$), we preliminarily infer that the optimal value of $V$ (i.e., the scale of VEBV in OEFS) is not a simple linear function of $K$, but is influenced by the nonlinear coupling of multiple factors.
> >     As inspired by your comments, these key factors may include but are not limited to:
> >     * **Attributes of Auxiliary Data**: The scale and semantic diversity of the OE dataset.
> >     * **Semantic Topological Relationships**: The degree of semantic separation between the OE dataset and the ID dataset.
> >     * **ID Task Complexity**: The number of ID categories ($K$) and their internal semantic richness.
> >     This actually reveals a **novel and unique** research proposition introduced by the VPC framework: unlike previous methods that rely solely on ID classifiers (where representation capacity is strictly limited to $K$), VPC, by introducing extra VEBVs, empowers the model to **explicitly represent** rich OOD samples for the first time. It is precisely this extra representation capacity beyond ID limits that makes exploring the optimal balance between $K$ (ID complexity) and $V$ (OOD representation capacity) not just parameter fine-tuning, but a highly valuable research question about "how to allocate ID and OOD expression capabilities within a finite geometric space". Therefore, we plan to make "the systematic exploration of these factors and their interactions" a key research focus of our future work, aiming to establish better theoretical guidance for this new representation paradigm. Once again, thank you for your valuable feedback and constructive comments, which are crucial for improving our experimental design and deepening our theoretical thinking.

---

> > ### Author Response · Authors · 2025-11-20
> > **A2: Response to Reviewer ioxr (Weakness 2) 3/3**
> >
> > **Q2 (Weakness 2): "The experiment setting is limited to CIFAR datasets, which are relatively small. Given the empirical analysis in Table 5, the proposed method works by adopting a significantly larger V value, compared to the number of classes. The performance of VPC remains unclear when the number of ID classes is large (e.g., ImageNet-1K), which may require an even larger V value. This can potentially lead to unaffordable computational overhead or suboptimal OOD performance"**
> >
> > ...
> >
> > 3.  **Clarification on "Unaffordable Overhead"**
> >     We wish to clarify a key misconception: the computational overhead of VPC is completely controllable and efficient, mainly due to its unique design:
> >     * **3.1 Memory/Parameter Overhead**: Our new analysis clarifies this key misconception: the $(K+V)$ EBV/VEBV prototypes in VPC are **"fixed constants"**, not "learnable parameters". They do not participate in backpropagation and thus do not increase the memory burden for gradient computation. The **only** learnable parameter overhead in VPC comes from replacing the standard classification head with a projection layer. As shown in **Table 9**, this overhead is very moderate. For example, on ResNet-18 (CIFAR-100), the net increase in learnable parameters is only **+0.47M**. Furthermore, our original fairness experiments analyzed the impact of adding this projection layer to baselines under the same settings. The results showed that although projector-augmented baselines have strictly more trainable parameters than ours, their performance was inconsistent across methods/backbones. We hypothesize that the extra dimensional transformation exacerbates classifier learning difficulty, especially for methods like PFS that rely on enforcing weight orthogonality, thereby amplifying sensitivity to optimization and data idiosyncrasies. Our method avoids this fragility: classifiers are pre-instantiated as optimal prototypes and remain fixed; training focuses solely on aligning features to these prototypes. This avoids the instability introduced by learning classifier weights after projection and consistently achieves better OOD detection results, meaning our performance gain does not stem from this small increase in parameters.
> >     * **3.2 Training Computational Overhead**: We added a `Computationally...` paragraph providing asymptotic bound analysis. VPC boasts an efficient **$O(V)$ linear complexity**. The calculation of our $\mathcal{L}_{\mathrm{VEBV}}$ loss is "diagonal" (highly parallelizable), with a complexity of only $O(V)$. We avoid calculating loss on a naive $(K+V)$-dimensional Softmax, which would require a prohibitive $O((K+V)^2)$ normalization step. Therefore, our training overhead is linear rather than quadratic, making it fully feasible even when $V$ is large.
> >     * **3.3 Inference Computational Overhead**: We added an `At inference time...` paragraph to directly respond to this. Inference for VPC, like MSP and Energy, involves a **"single matrix multiplication"**. While MSP/Energy scales as $O(d_{\text{feat}} \times K)$, VPC scales as $O(d_{\text{oefs}} \times (K+V))$. On modern GPUs, both are highly optimized matrix operations with negligible difference in **"actual inference latency"**. This stands in sharp contrast to non-parametric methods like KNN (whose complexity is $O(N \times d_{\text{feat}})$).
> >     In summary, the computational overhead of VPC is controllable, and the performance benefits it brings are significant and valuable.
> >
> > **Conclusion:** The newly added ImageNet experiments and complexity analysis jointly confirm that VPC possesses good **scalability**. It can effectively handle large-scale ID datasets using a moderate scale of $V$ (e.g., $V \approx K$) without introducing significant extra overhead.
> >
> > We believe the revised manuscript now presents a complete and robust framework, with both strong empirical results on large-scale datasets and clear implementation guidance. We sincerely appreciate the time you invested in this review. We hope that these targeted revisions satisfactorily address your concerns, and we look forward to any further discussion.

---

> > > ### Comment · Reviewer_ioxr · 2025-11-26
> > >
> > > The authors' response is clear and solves most of my concerns. I recommend acceptance. Increased my score to 8. Good luck!

---

> > > > ### Author Response · Authors · 2025-11-26
> > > > **Response to Reviewer ioxr：Appreciation for Favorable Assessment and Acceptance Recommendation**
> > > >
> > > > We thank Reviewer ioxr for their time and for raising the score to 8. We are delighted that our rebuttal clarified your concerns regarding the implementation details and the applicability of VPC on larger datasets.
> > > >
> > > > Your insightful comments have significantly helped us improve the quality and scope of our work. We look forward to presenting a high-quality final version of the paper that incorporates these improvements.
> > > >
> > > > The Authors

---

### Official Review · Reviewer_H2ze · 2025-11-01

**Soundness:** 3
**Presentation:** 2
**Contribution:** 2
**Rating:** 4
**Confidence:** 4

**Summary:**

This paper focuses on the out-of-distribution detection problem. Specifically, this work proposes a geometry-driven framework for OOD detection, that distinguishes ID data from OOD data by embedding to an orthogonal equiangular feature space, which provides an evidential priors that mitigates the instability from outlier exposure data, and then a new activation-based OOD score is introduced which is demonstrated to be effective than traditional softmax and energy scores.

**Strengths:**

1. The idea of this work that fixing the orthogonal equiangular basis builds on the neural collapse, to the reviewer's knowledge on the area of OOD detection, is new and interesting.
2. The paper has provided comprehensive experimental verification on demonstrating the performance of the proposed method.

**Weaknesses:**

1. Although the proposal is empirically demonstrated to be effective, there is no theoretical justification that rigorously justify or explain why the newly introduced representation structure can help the OOD separability or generalize to the unseen OOD distributions.
2. The computational cost or overhead seems to be much than previous OOD score based on softmax and energy, especially for large-scale ID dataset.
3. Except for the common benchmark datasets, some large-scale dataset should also be considered in the verification. And more advanced OOD detection methods like ASH or other scoring functions are not considered in the comparison, which should carefully support the claim that current proposal is SOTA.

**Questions:**

1. What guarantees separability between unseen OOD samples and ID features, as the OOD distribution shifts outside the auxiliary OE coverage, with the orthogonal equiangular construction?
2. What is the computational complexity for maintaining and updating thousands of supervised classifiers in high dimensionality? Can this approach extend to modern vision transformers or multimodal models?
3. Is it ensured that the penultimate layer retains the intended geometry under nonlinear activations or the presence of normalization layers?
4. Neural Collapse is known to predominantly occur asymptotically with cross-entropy, under certain conditions. In the context of fixed EBVs and with extra constraints, what guarantees converge to the intended geometric configuration?

---

> ### Author Response · Authors · 2025-11-20
> **A1: Response to Reviewer H2ze (Weakness 1, Question 1) 1/2**
>
> We thank Reviewer H2ze for the critical feedback. You correctly identified the lack of theoretical justification and concerns about cost as major gaps. To bridge these gaps, we have significantly expanded the paper:
>
> * **Theory:** We derived **three new mathematical proofs** (optimality, robustness, orthogonality) in **Appendix A.2/A.3** to justify geometric convergence.
> * **Scalability:** We added **ImageNet-1K experiments** to prove efficiency and performance.
>
> Below, we provide a detailed point-by-point response to your comments. We demonstrate how we have incorporated every single one of your suggestions into the revised manuscript.
>
> **Q1 (Weakness 1, Question 1): "...lack of theoretical justification for the proposed representation structure...", "...guarantees for OOD separability when distribution shifts beyond OE coverage..."**
>
> **A1:** We sincerely appreciate this profound and pertinent comment. We must candidly acknowledge that our current work focuses on empirical validation and has not yet established rigorous closed-form mathematical proofs, such as generalization error bounds, to comprehensively and formally guarantee OOD separability. This is indeed a limitation of the present work and a direction we are committed to addressing in the future. Therefore, we have added theoretical analyses in the Appendix regarding the optimization endpoint theorem of the ENC loss, the robustness proposition, and the orthogonality proposition of the OC loss (**Appendix A.2.1, A.2.2, and A.3**), aiming to construct the geometric theoretical foundation for the effectiveness of VPC.
>
> 1.  **Theoretical Guarantee of ID Subspace Convergence**: The prerequisite for OOD separability is that ID features must be compact and stable. Leveraging the properties generated by the **Neural Collapse (NC)** phenomenon, we performed rigorous mathematical derivations for our proposed $\mathcal{L}_{\mathrm{ENC}}$ in **Appendix A.2 (Evidential Prior Guided Neural Collapse)**.
>     * **Optimality**: In **Appendix A.2.1 (Theorem 1)**, we proved that the global optimum of $\mathcal{L}_{\mathrm{ENC}}$ corresponds to the perfect alignment between the ID feature vector $\hat{m}$ and its preset EBV prototype $\hat{w}$ ($\hat{m}=\hat{w}$). This mathematically guarantees that ID features can theoretically converge to the most compact Neural Collapse geometric state.
>     * **Robustness**: Addressing the potential interference from OE data during joint training, in **Appendix A.2.2 (Proposition 1)**, we derived the upper bound of the gradient. The proof shows that the evidential prior mechanism imposes a strict bound ($\le 1$) on the gradient of the log Dirichlet parameter ratio. This mechanism acts as a **"geometric rate-limiter"**, effectively smoothing the optimization process and preventing ID features from being "pulled away" by the gradient noise of massive OE data or overfitting, thereby maintaining a clear and stable ID boundary.
>
> 2.  **Geometric Proof of Structural Decoupling and Orthogonality**: Traditional OOD detection often relies on heuristically constraining the ID classifier to output a uniform distribution for OOD samples. In contrast, VPC not only provides a structural decoupling but also theoretically guarantees the strictness of this decoupling. We added a theoretical proof for $\mathcal{L}_{\mathrm{OC}}$ in **Appendix A.3 (Proposition 2)** of the revised manuscript. This proof reveals that under the **Simplex ETF** geometric framework, minimizing the uniform distribution loss is mathematically equivalent to enforcing **geometric orthogonality** between OE features and all ID prototypes. This means that $\mathcal{L}_{\mathrm{OC}}$ is not merely a smoothing at the probabilistic level, but imposes a hard orthogonal constraint in the high-dimensional space, ensuring that OOD features are pushed into the **null space** of the ID subspace. This provides a solid geometric theoretical basis for the separability of ID and OOD, fundamentally blocking the possibility of confusion from a mechanistic standpoint.

---

> > ### Author Response · Authors · 2025-11-20
> > **A1: Response to Reviewer H2ze (Weakness 1, Question 1) 2/2**
> >
> > **Q1 (Weakness 1, Question 1): "...lack of theoretical justification for the proposed representation structure...", "...guarantees for OOD separability when distribution shifts beyond OE coverage..."**
> >
> > ...
> >
> > 3.  **Complementary Generalization Based on Massive OE Data**: Regarding the coverage of unseen OOD, when OOD shifts beyond the coverage of OE, our core logic is rooted in the fundamental advantage of **Outlier Exposure (OE)** data and enhances it critically:
> >     * **Representativeness and Coverage of OE Data**: As shown in previous studies and our experiments (Tables 1 and 2), large-scale OE datasets like **80 Million Tiny Images** or **ImageNet-21k-p** contain extremely broad and diverse distribution patterns. These massive data actually act as an extremely dense sampling of the real-world OOD space, serving not just as specific samples but as representatives of generic "non-ID" features.
> >     * **Construction and Generalization of Semantic Complement**: Utilizing these representative OE data, we construct the **VEBV** subspace as a complementary space to the ID semantic space. The purpose of training is not merely to "memorize" these specific OE samples, but to train the feature extractor to learn a general mapping rule: "samples possessing ID semantics map to **EBV**"; "samples lacking ID semantics map to **VEBV**". For unseen OOD samples (even if they are outside the OE coverage), as long as they differ significantly from ID categories in semantics (i.e., do not possess ID features), the network will project their features to the "non-ID" region, namely our preset **VEBV** subspace, based on the learned mapping rules, thereby producing high activation values.
> >
> > In summary, although we have not provided formal generalization error bounds, the effectiveness of VPC is built upon the proven robust convergence of ID, explicit spatial orthogonality, and the complementary subspace mechanism constructed using massive OE data. This design enables the model to handle "unknown" distributions that are unseen but similarly lack ID semantic features. We thank you again for pointing out this gap between theory and empiricism, which prompted us to think more deeply about the theoretical logic behind model generalization.

---

> ### Author Response · Authors · 2025-11-20
> **A2: Response to Reviewer H2ze (Weakness 2, Question 2) 1/2**
>
> **Q2 (Weakness 2, Question 2): "The computational cost or overhead seems to be much than previous OOD score based on softmax and energy, especially for large-scale ID dataset." "What is the computational complexity for maintaining and updating thousands of supervised classifiers in high dimensionality? Can this approach extend to modern vision transformers or multimodal models?"**
>
> **A2:** We sincerely appreciate this critical question regarding computational cost and scalability. This is a very pragmatic and pertinent concern, crucial for assessing the usability of a new method in real-world large-scale scenarios. We wish to clarify the actual computational overhead of VPC from two aspects—theoretical complexity analysis and empirical scalability experiments—and to dispel potential misconceptions about "maintaining thousands of classifiers."
>
> **1. Clarification on "Maintaining and Updating" Classifiers**
> First, we wish to clarify the fundamental difference between VPC and prototype/metric learning-based methods. In the VPC framework, all classifier prototypes (including $K$ EBVs and $V$ VEBVs) are generated via ETF construction at initialization and are immediately set as **frozen constants**. In contrast, prototype/metric learning methods typically require continuous updates of these prototypes via EMA. Therefore, during training, we **do not need to calculate gradients for these prototypes**, nor do we need to update them. This is fundamentally different from traditional Fully Connected Layers that require constant weight updates. This means the computational cost of "maintaining" these classifiers is effectively zero. The model only needs to learn the backbone and the projection layer to align features to these fixed geometric anchors. We have added a detailed parameter and computational overhead analysis in **Appendix A.4 "From Learnable Classifiers to Vast Predefined Classifiers"**. The analysis is as follows:
>
> * **1.1 Memory/Parameter Overhead:** We clarify a key misconception: the $(K+V)$ EBV/VEBV prototypes in VPC are **"fixed constants"**, not "learnable parameters". They do not participate in backpropagation, thus adding no memory burden for gradient computation. The **only** learnable parameter overhead in VPC comes from replacing the standard classification head with a projection layer. As shown in **Table 9**, this overhead is very moderate. For example, on ResNet-18 (CIFAR-100), the net increase in learnable parameters is only **+0.47M**. Furthermore, our original fairness experiments analyzed the impact of adding this projection layer to baselines under the same settings. The results showed that although projector-augmented baselines have strictly more trainable parameters than ours, their performance was inconsistent across methods/backbones. We hypothesize that the extra dimensional transformation exacerbates classifier learning difficulty, especially for methods like PFS that rely on enforcing weight orthogonality, thereby amplifying sensitivity to optimization and data idiosyncrasies. Our method avoids this fragility: classifiers are pre-instantiated as optimal prototypes and remain fixed; training focuses solely on aligning features to these prototypes. This avoids the instability introduced by learning classifier weights after projection and consistently achieves better OOD detection results, meaning our performance gain does not stem from this small increase in parameters.
>
> * **1.2 Training Computational Overhead:** VPC boasts an efficient **$O(V)$ linear complexity**. The calculation of our $\mathcal{L}_{\mathrm{VEBV}}$ loss is "diagonal" (highly parallelizable), with a complexity of only $O(V)$. We avoid calculating loss on a naive $(K+V)$-dimensional Softmax, which would require a prohibitive $O((K+V)^2)$ normalization step. Therefore, our training overhead is linear rather than quadratic, making it fully feasible even when $V$ is large.
>
> * **1.3 Inference Computational Overhead:** We explicitly address this in the new "At inference time..." paragraph. Inference for VPC, like MSP and Energy, involves a **"single matrix multiplication"**. While MSP/Energy scales as $O(d_{\text{feat}} \times K)$, VPC scales as $O(d_{\text{oefs}} \times (K+V))$. On modern GPUs, both are highly optimized matrix operations with negligible difference in **"actual inference latency"**. This stands in sharp contrast to non-parametric methods like KNN (whose complexity is $O(N \times d_{\text{feat}})$).

---

> ### Author Response · Authors · 2025-11-20
> **A2: Response to Reviewer H2ze (Weakness 2, Question 2) 2/2**
>
> **Q2 (Weakness 2, Question 2): "The computational cost or overhead seems to be much than previous OOD score based on softmax and energy, especially for large-scale ID dataset." "What is the computational complexity for maintaining and updating thousands of supervised classifiers in high dimensionality? Can this approach extend to modern vision transformers or multimodal models?"**
>
> ...
>
> **2. Scalability Verification**
> To fully address your doubts regarding scalability to modern architectures, we invested resources to conduct large-scale experiments during the Rebuttal (see revised **Table 1**). We successfully applied VPC to the **ImageNet-1K** dataset and evaluated it using the **ViT-B-16** architecture. The experimental results show that VPC not only runs efficiently on the ViT architecture but also achieves superior performance (FPR95 56.20%, AUROC 85.20%). This result powerfully proves that despite introducing thousands of extra predefined classifiers, VPC can still seamlessly and efficiently scale to large-scale datasets and modern Transformer architectures. Reference [1] introduces the EBV approach as a novel classification paradigm, while Reference [2] explores its application in continual learning. Both works provide extensive validation on large-scale datasets such as ImageNet-1K and Transformer architectures. These studies corroborate our experimental findings, collectively demonstrating the generalizability and potential of methods based on predefined geometric structures within large-scale modern architectures.
>
> **Table R1. Results on ImageNet-1k benchmark with auxiliary OOD data. The best result is in bold.**
>
> | Model | Method| iNat FPR↓ | iNat AUC↑ | Tex FPR↓| Tex AUC↑| SUN FPR↓| SUN AUC↑| Places FPR↓ | Places AUC↑ | Avg FPR↓| Avg AUC↑| ID Acc↑|
> | ------------ | ------------- | --------- | --------- | --------- | --------- | --------- | --------- | ----------- | ----------- | --------- | --------- | --------- |
> | **ResNet50** | OE| 48.60 | 88.72 | 58.85 | 82.60 | 61.75 | 82.90 | 70.70| 80.55| 59.98 | 83.69 | 76.00 |
> || Energy-OE | 49.40 | 88.40 | 59.60 | 82.25 | 62.40 | 82.70 | 71.30| 80.25| 60.68 | 83.40 | 75.75 |
> || DAL| 48.00 | 89.05 | 58.00 | 82.95 | 61.30 | 83.15 | 67.82| 80.75| 58.78 | 83.98 | 75.90 |
> || PFS| 46.40 | 89.20 | 56.50 | **83.10** | 61.00 | **83.25** | 67.50| 80.95| 57.85 | 84.13 | 76.02 |
> || **Ours: VPC** | **43.50** | **91.20** | **56.00** | 83.00 | **60.20** | 83.10 | **66.30**| **81.50**| **56.50** | **84.70** | **76.11** |
> | **ViT-B-16** | OE| 42.15 | 90.38 | 52.45 | 85.85 | 65.80 | 82.20 | 70.35| 80.85| 57.69 | 84.82 | 80.02 |
> || Energy-OE | 42.75 | 90.05 | 53.15 | 85.50 | 66.25 | 81.95 | 70.95| 80.50| 58.28 | 84.50 | 79.85 |
> || DAL| 40.65 | 90.86 | **51.15** | 86.10 | 65.07 | 82.30 | 70.25| 80.90| 56.78 | 85.04 | 80.09 |
> || PFS| 40.85 | 90.80 | 51.30 | 86.05 | 65.35 | 82.25 | 70.20| 80.95| 56.93 | 85.01 | 80.13 |
> || **Ours: VPC** | **40.00** | **91.10** | 51.30 | **86.00** | **65.20** | **82.30** | **68.30**| **81.40**| **56.20** | **85.20** | **80.32** |
>
> As shown in Table R1 (referenced as Table 1 in the revised manuscript), VPC demonstrates strong competitiveness on both architectures.
>
> [1] Shen Y, Sun X, Wei X S, et al. Equiangular basis vectors: A novel paradigm for classification tasks[J]. International Journal of Computer Vision, 2025, 133(1): 372-397.
>
> [2] Seo M, Koh H, Jeung W, et al. Learning equi-angular representations for online continual learning[C]//Proceedings of the IEEE/CVF Conference on Computer Vision and Pattern Recognition. 2024: 23933-23942.
>
> Once again, thank you for raising this practical issue, which prompted us to analyze the algorithm's complexity and overhead in greater detail in the revised manuscript.

---

> ### Author Response · Authors · 2025-11-20
> **A3: Response to Reviewer H2ze (Weakness 3)**
>
> **Q3 (Weakness 3): "...should also be considered in the verification. And more advanced OOD detection methods like ASH or other scoring functions are not considered in the comparison, which should carefully support the claim..."**
>
> **A3:** We sincerely thank you for your constructive suggestions regarding the breadth of experimental verification and the selection of comparison baselines. Your advice on large-scale dataset validation and broader method comparisons (such as ASH) is of critical significance for enhancing the empirical persuasiveness and positioning accuracy of this paper.
>
> **1. [New] Comprehensive Benchmarking on ImageNet-1K**
> We deeply recognize the necessity of validating VPC on large-scale benchmarks. To this end, we have dedicated significant effort to conducting comprehensive supplementary experiments. On the **ImageNet-1K** dataset, we rigorously compared VPC with all key OE baseline methods (including OE, Energy-OE, DAL, and PFS) using two mainstream architectures: **ResNet-50** and **ViT-B-16**.
> As shown in Table R1 (referenced as Table 1 in the revised manuscript), VPC demonstrates strong competitiveness on both architectures. For instance, on the **ResNet-50** architecture, VPC achieves an average **FPR95 of 56.50%**, outperforming the strong baseline PFS (57.85%). On the **ViT-B-16** architecture, VPC (**56.20%**) similarly surpasses all comparison methods. On both architectures, VPC achieves the highest average AUROC while maintaining ID classification accuracy comparable to or better than baselines. We believe these new evidences on the ImageNet scale strongly demonstrate that the VPC framework possesses excellent scalability and its effectiveness is not limited to CIFAR-scale datasets.
>
> **2. [Clarification] On the Selection of Baselines and Scoring Functions**
> We greatly appreciate your mention of advanced OOD detection methods like ASH. We wish to clarify our considerations regarding the selection of baselines:
>
> * **2.1 Differences in Methodological Paradigm:**
>     Methods like **ASH (Activation Shaping)** primarily fall into the **post-hoc** category, focusing on pruning activation values during inference. In contrast, VPC belongs to the **OE-based joint training** paradigm, focusing on actively reshaping the geometric structure of the feature space during training via $\mathcal{L} _ {\mathrm{VEBV}}$ and $\mathcal{L} _ {\mathrm{ENC}}$. To ensure fairness and pertinence in comparison, our primary baselines (OE, Energy-OE, DAL, PFS) are all selected from the same "OE-based training paradigm" (i.e., utilizing ID + Auxiliary OE data).
>
> * **2.2 Integral Nature of VPC Score:**
>     The **VPC Score** is not a generic post-hoc score but a metric deeply coupled with our preset **OEFS** geometric structure (i.e., based on subspace L2 activation intensity). It directly reflects the optimization degree of the training objectives.
>     * **Incompatibility with Magnitude Pruning:** The core mechanism of ASH involves pruning feature activation magnitudes (e.g., truncating the bottom $p\%$). However, VPC is built upon the **unit hypersphere** assumption. In our framework, feature vectors $\hat{m}$ must undergo **explicit $\ell_2$-normalization** before similarity calculation. This means that in our framework, the magnitude information of raw features is eliminated by normalization. Therefore, directly applying magnitude-based strategies like ASH would inherently conflict with our "spherical geometry-based" design.
>     * **Comparison with Non-parametric Methods:** Other methods, such as KNN, are **non-parametric** post-hoc methods requiring distance calculations between the test sample and all training samples, with inference complexity $O(N)$. On large-scale datasets (like ImageNet), this incurs massive memory and computational overhead. In contrast, VPC is **parametric**, requiring similarity calculations only with the preset $K+V$ prototypes, making inference efficient with constant overhead.
>     * Nevertheless, in **Sec 4.4**, we indeed compared different scoring functions (such as MSP, Uncertainty, EDL Prob), and the results show that the VPC Score achieves better OOD detection performance. Additionally, in **Appendix A.6 "Activation strength is a better OOD scoring function"**, we further explored the benefits of using L2 Score as an OOD scoring function.
>
> **3. Modification of the SOTA Claim**
> We fully accept your suggestion that the term "State-of-the-Art (SOTA)" should be used more cautiously.
> * **Expression Revision:** We have carefully reviewed and revised the entire paper, removing broad SOTA claims.
> * **Precise Definition:** We strictly limit our conclusions to specific experimental settings. Our current statement is more objective: VPC achieves the best performance compared to similar methods under the specific training paradigms of **auxiliary OE-based joint fine-tuning** and **one-stage training**.
>
> Thank you again for helping us enhance the rigor and precision of the paper.

---

> ### Author Response · Authors · 2025-11-20
> **A4: Response to Reviewer H2ze (Question 3)**
>
> **Q4 (Question 3): "Is it ensured that the penultimate layer retains the intended geometry under nonlinear activations or the presence of normalization layers?"**
>
> **A4:** We sincerely appreciate your rigorous scrutiny of the model architecture details. Your question regarding whether nonlinear activations might disrupt the preset ETF geometry strikes at the core of whether our geometric assumptions hold.
>
> We fully share your concern: The **Simplex ETF** structure necessitates negative inner products between feature vectors ($-1/(K-1)$), whereas common nonlinear activation functions (such as ReLU) truncate features to non-negative values, confining the distribution to the positive orthant. Mathematically, this would indeed impede the formation of an ideal simplex geometry. To dispel this doubt, we wish to clarify the specific architectural design of VPC at the terminal layers, which is expressly designed to ensure the feature space can accommodate the intended geometry:
>
> 1.  **Adoption of a Linear Projection Layer:** As detailed in the newly added **Appendix A.4 "From Learnable Classifiers to Vast Predefined Classifiers"**, we replace the backbone's original classification head with a learnable projection layer to match the dimensionality requirements of OEFS. **Crucial Clarification:** We explicitly confirm here that the output of this projection layer is **purely linear**. We do not append any nonlinear activation functions (such as ReLU) after this layer. This design allows feature vectors to freely take values across the full high-dimensional space (including negative values), thereby satisfying the negative correlation requirements needed to form an ETF.
>
> 2.  **Explicit Spherical Normalization:** Regarding normalization layers, we implement an explicit geometric enforcement. After the projection layer and prior to entering the loss function calculation, we perform a mandatory **$\ell_2$-normalization** on the features. This operation forcibly projects the features onto the unit hypersphere within the computation graph. Therefore, regardless of the distribution in preceding layers, the features entering the OEFS matching stage are strictly guaranteed to reside on the manifold $\mathbb{S}^{d-1}$.
>
> In summary, by removing terminal nonlinear activations and implementing explicit $\ell_2$-normalization, our architectural design avoids factors that could destroy the ETF geometric structure, ensuring that the penultimate layer features perfectly adapt to our preset **OEFS** space. Thank you again for checking this critical technical detail, which has prompted us to describe the specific implementation details of the terminal network layers more clearly in the revised manuscript.

---

> ### Author Response · Authors · 2025-11-20
> **A5: Response to Reviewer H2ze (Question 4)**
>
> **Q5: "What guarantees convergence to the expected geometric configuration in the context of fixed EBVs with additional constraints (OOD loss)?"**
>
> **A5:** We sincerely appreciate this profound theoretical question. It touches upon the core stability of our method. To address this, we have added detailed theoretical analyses in **Appendix A.2** and **A.3**. We argue that the convergence to the expected geometric configuration is guaranteed by three theoretical pillars—optimality, robustness, and orthogonality—and is further corroborated by strong empirical evidence.
>
> **1. Theoretical Guarantee of ID Subspace Convergence**
> * **Optimality (Appendix A.2.1, Theorem 1):** We constructed the Lagrangian for $\mathcal{L} _ {\mathrm{ENC}}$ and proved via KKT conditions that the global minimum is achieved **if and only if** the ID feature vector $\hat{m}$ perfectly aligns with its corresponding fixed EBV prototype $\hat{w}$. This theoretically anchors the global optimum of the ID objective to the intended Neural Collapse geometry.
> * **Robustness (Appendix A.2.2, Proposition 1):** Addressing the concern that OOD losses might "derail" the optimization, we derived the gradient bounds for $\mathcal{L} _ {\mathrm{ENC}}$. We proved that the evidential prior mechanism acts as a **"gradient rate-limiter"** . This ensures that $\mathcal{L} _ {\mathrm{ENC}}$ functions as a stabilizer, preventing ID features from being drastically pulled away by the gradient noise introduced by massive OE data during joint training.
>
> **2. Mathematical Guarantee of Orthogonality Constraint**
> In **Appendix A.3 (Proposition 2)**, we provide a rigorous proof for the mechanism of $\mathcal{L} _ {\mathrm{OC}}$. We demonstrate that under the **Simplex ETF** framework, minimizing the KL divergence between the OE prediction and a uniform distribution is **mathematically equivalent** to enforcing **geometric orthogonality** between OE features and all ID prototypes. This ensures that the "additional constraints" explicitly push OOD features into the null space of the ID subspace, thereby structurally preventing interference with the ID geometry.
>
> **3. Empirical Verification via Subspace Independence**
> The achievement of the expected geometric configuration is strongly validated by our empirical findings in **Appendix A.5 (Table 10)** . Our ablation study on VPC Score coefficients ($\alpha, \beta$) reveals a critical phenomenon: using **only the activation intensity of the VEBV subspace** ($\alpha=0$) or **only the EBV subspace** ($\beta=0$) is sufficient to achieve better detection performance. This **"Subspace Independence"** strongly implies that the intended geometric separation (where ID activates EBV and OOD activates VEBV) has indeed been successfully converged to; otherwise, the signals from individual subspaces would not be discriminative enough on their own.
>
> **Conclusion:** The combination of optimality proofs, gradient robustness, geometric orthogonality guarantees, and the empirical evidence of subspace independence provides a solid basis for the convergence of our framework to the intended geometric configuration.
>
>
> We believe this revision transforms the paper from a purely empirical study to one backed by rigorous theoretical proofs and large-scale validation. We have effectively filled the theoretical gap and dispelled concerns about computational cost. We respectfully hope these substantial improvements warrant a re-evaluation of our work.

---

> ### Comment · Area_Chair_xnSr · 2025-11-25
> **Feedback on the Rebuttal**
>
> Dear Reviewer,
>
> Could you give the feedback to the rebuttal?
>
> Best Regards,
>
> AC

---

> > ### Comment · Reviewer_H2ze · 2025-11-26
> >
> > Thanks for the detailed rebuttal. Most of my concerns are addressed by the clarification and the newly added experiments. I will adjust my score accordingly. Thanks and good luck!

---

> > > ### Author Response · Authors · 2025-11-27
> > > **Gratitude for Favorable Reassessment from Reviewer H2ze**
> > >
> > > We extend our sincere gratitude to Reviewer H2ze for their time, constructive criticism, and the decision to raise the score. We are heartened to know that our rebuttal and the supplementary revisions have successfully addressed your concerns.
> > >
> > > Your insightful feedback played a pivotal role in shaping the current version of this manuscript. specifically. These improvements have significantly elevated the paper's academic rigor and generalizability.
> > >
> > > We remain committed to ensuring these enhancements are perfectly integrated into revised version.
> > >
> > > The Authors

---

### Official Review · Reviewer_m4Yu · 2025-11-01

**Soundness:** 3
**Presentation:** 2
**Contribution:** 3
**Rating:** 6
**Confidence:** 4

**Summary:**

The paper “LET OOD FEATURE EXPLORE VAST PREDEFINED CLASSIFIERS (VPC)” proposes a new framework for out-of-distribution (OOD) detection that focuses on explicitly modeling the geometry of feature representations. Current outlier exposure (OE) methods improve robustness by fine-tuning networks on in-distribution (ID) and auxiliary OOD data, but they largely encourage low-confidence OOD predictions or post-hoc score regularization without directly shaping feature space structure. As a result, separability between ID and OOD representations is often unstable, and OOD features lack sufficient representational diversity.

To address these issues, the authors introduce Vast Predefined Classifiers (VPC), which define an Orthogonal Equiangular Feature Space (OEFS) composed of fixed Equiangular Basic Vectors (EBVs) representing ID classes and a large pool of Vast EBVs (VEBVs) representing an extended OOD subspace. They design several complementary losses that guide feature alignment: an evidential prior–guided Neural Collapse (ENC) loss that stabilizes ID features toward their EBVs, a VEBV loss that encourages OOD features to explore the VEBV subspace, and an orthogonality constraint that prevents mixing of ID and OOD features. The framework introduces the VPC Score, based on the L2 activation intensity difference between the ID and OOD subspaces, serving as a class-agnostic and interpretable OOD detection metric. Experiments on CIFAR-10 and CIFAR-100 using multiple backbones (WideResNet, ResNet, DenseNet) show that VPC achieves state-of-the-art results, outperforming existing baselines such as OE, Energy-OE, DAL, and PFS. Ablation studies confirm that each component—particularly the geometric subspace design and the VEBV loss—contributes to improved performance.

**Strengths:**

The advantages of this method lie in its geometric interpretability, stable training through evidential priors, scalability for modeling diverse OOD patterns, and class-agnostic scoring that avoids bias from classifier heads.

Empirical results demonstrate strong robustness and clear separability between ID and OOD features.

Eq 5 in preventing over-bias toward any single class and promoting stable convergence of ID feature is new to me, sounds interesting.

**Weaknesses:**

The authors may need to further consider the definition of OOD detection in Sec 3.1. Defining OOD data to be those out of the label space might be more suitable for OOD detection. The adopted definition based on support set may fail to consider the case of covariate shift, which should be considered as ID instead of OOD.

A paragraph of intuition at the beginning of Sec 3.2 will improve the clarity of the manuscript, otherwise readers may understand what the authors want to do very late in Sec 3.3 – 3.4. The intuition should echo the drawbacks of previous works in OOD diversity and some other things like this.

Hat m in Eq 4 is not defined, I think it is the embedding feature, right? There are a lot of other notations that are not defined, such as \tau in the same formulation. The authors should carefully check about them.

The proposed method is somehow like a variance of KNN, could the authors further highlight the improvement over KNN?

Could Eq 6 ensure the diversity of OOD basis vectors? The models tend to map OOD data to the same region in the embedding space even at the very beginning,  optimizing via Eq 6 will exacerbate this case and in this case V=1 is enough, obviously violating the claimed “vast” features in Sec 3.2.

The authors define w in the unit hypersphere, but, from my view, none of the objective in Sec 3 can ensure this constraint.

As far as I know, many works conducted experiments on ImageNet benchmark or ViT family, such as [1,2]. I do not want to add the rebuttal burden of the authors, as many researchers lack sufficient computational resources. But you can consider it as a suggestion to improve the quality of this research in the future.

[1] Watermarking for Out-of-distribution Detection

[2] Out-of-distribution Detection Learning with Unreliable Out-of-distribution Sources

**Questions:**

Kindly please see the weakness above.

---

> ### Author Response · Authors · 2025-11-20
> **A1: Response to Reviewer m4Yu (Weakness 1)**
>
> We express our deepest gratitude to Reviewer m4Yu for the exceptionally detailed and constructive review. Your keen observations on structural clarity, mathematical rigor, and conceptual definitions (e.g., OOD definition, KNN distinction) were incredibly precise and have been instrumental in elevating the quality of our manuscript.
>
> We are also particularly touched by your thoughtfulness regarding computational resources. Motivated by your valuable suggestion to explore larger benchmarks, we have successfully conducted comprehensive experiments on ImageNet-1K with ViT architectures during the rebuttal. As detailed below, these results further validate the scalability of VPC.
>
> Below, we provide a detailed point-by-point response to your comments. We demonstrate how we have incorporated every single one of your suggestions into the revised manuscript.
>
>
> **Q1 (Weakness 1): "The authors may need to further consider the definition of OOD detection in Sec 3.1. Defining OOD data to be those out of the label space might be more suitable for OOD detection. The adopted definition based on support set may fail to consider the case of covariate shift, which should be considered as ID instead of OOD."**
>
> **A1:** We are extremely grateful for your detailed guidance during the review process, especially for the key correction regarding the formal definition of OOD in Section 3.1. You are absolutely correct. We deeply realize that in the original manuscript, we used a definition based on input space support disjointness ($supp(D_{out})\cap supp(D_{in})=\emptyset$). This expression indeed had omissions and was not rigorous enough. As you pointed out, a more precise definition that aligns with academic consensus should be based on the disjointness of the label space.
>
> We express our deepest gratitude for this valuable correction. In the revised manuscript, we have fully adopted your suggestion and rewritten the definition in Section 3.1. This part now explicitly elucidates the core assumption based on the label space.
>
> We believe this revision makes our problem statement clearer and consistent with established norms in the OOD field. Thank you again for helping us strengthen the rigor of the paper on this fundamental definition.

---

> ### Author Response · Authors · 2025-11-20
> **A2: Response to Reviewer m4Yu (Weakness 2)**
>
> **Q2 (Weakness 2): "Adding an intuitive explanation paragraph at the beginning of Section 3.2 would improve clarity..."**
>
> **A2:** We sincerely appreciate this extremely valuable and constructive suggestion. We fully and profoundly agree with your point. You astutely pointed out that the beginning of Section 3.2 in the original manuscript indeed lacked an intuitive explanation, failing to provide a clear "blueprint," which forced readers to read the subsequent Sections 3.3 and 3.4 to fully understand the core motivation behind our design of OEFS. To systematically address this clarity issue, we adopted your suggestion and comprehensively reorganized and supplemented the narrative logic of Sections 3.1 to 3.2.
>
> First, at the end of **Section 3.1 (Preliminary)** (newly added), we supplemented an in-depth analysis of the limitations of the traditional OE paradigm. We explicitly pointed out the limitations of current methods (such as $\mathcal{L}_{\rm OE}$) regarding representation geometry, thereby naturally leading to the urgent need for a new scheme to "structurally decouple" ID and OOD representations. This provides a clear and necessary motivation for introducing OEFS in Section 3.2.
>
> Next, at the beginning of **Section 3.2 (Orthogonal Equiangular Feature Space)**, we added a brand new **Intuitive Explanation** paragraph exactly as you suggested. This paragraph elucidates the core intuition of VPC—achieving structural decoupling of ID/OOD through a priori partitioned preset space. This answers the question of "what it is."
>
> Finally, we also revised the end of **Section 3.2** (newly added) to include a summary of the subsequent content. We explicitly linked the OEFS to the entire methodological closed loop: encompassing not only the dual alignment training strategy but also mentioning the **VPC Score** based on activation intensities of different subspaces used for inference.
>
> We believe that this series of revisions—from the "motivation" in Section 3.1 to the "intuition" at the beginning of Section 3.2, and then to the "methodological overview" at the end of Section 3.2—jointly constructs a clear narrative path. This ensures that readers have a comprehensive understanding of the full intent of the OEFS framework before diving into the loss functions in Sections 3.3 and 3.4 and the scoring function in Section 3.5.
>
> Thank you again for your valuable suggestion, which has greatly improved the clarity and logic of our paper.

---

> ### Author Response · Authors · 2025-11-20
> **A3: Response to Reviewer m4Yu (Weakness 3)**
>
> **Q3 (Weakness 3): "Undefined symbols in Eq. 4... presumably referring to embedding features? Many symbols like $\tau$ in the same formula are undefined..."**
>
> **A3:** We are extremely grateful for your meticulous review and for astutely pointing out this major oversight in our original manuscript. You are absolutely correct. In Section 3.3 of the original manuscript, when introducing symbols like $\hat m_i^{\mathrm{id}}$ and $\tau$ in Eq. (4), we indeed omitted their immediate definitions. This undoubtedly caused confusion regarding the intent of the formula, and we sincerely apologize for this.
>
> Following your valuable feedback, we have immediately added these key definitions right after Eq. (4) in the revised manuscript. We now explicitly state:
> * $\hat m_i^{\mathrm{id}}$: As you correctly speculated, it is exactly the $\ell_2$-normalized feature vector of the ID sample $x_i^{\text{id}}$.
> * $\tau$: It is a temperature hyperparameter used to scale the cosine similarity.
>
> We have followed your suggestion to double-check all symbols in the manuscript to ensure their clarity and consistency. Thank you again for your valuable correction, which helped us greatly improve the rigor of the paper.

---

> ### Author Response · Authors · 2025-11-20
> **A4: Response to Reviewer m4Yu (Weakness 4)**
>
> **Q4 (Weakness 4): "This method is similar to a variant of KNN, can you further clarify its improvements relative to KNN?"**
>
> **A4:** We sincerely appreciate this profound and highly insightful question. Comparing our method with classic k-NN approaches indeed precisely reflects the design philosophy of VPC from a certain angle. While both intuitively involve the process of "calculating similarity with reference vectors," we wish to further clarify that VPC is not a variant of k-NN. Instead, VPC is a fully integrated framework encompassing representation space construction, optimization mechanisms, and scoring strategies. It achieves fundamental improvements and breakthroughs over k-NN in the following three core dimensions:
>
> 1.  **Core Paradigm: Non-parametric Post-hoc vs. Geometric Prior Driven Integrated Construction**
>     This is the most essential difference between the two.
>     * **k-NN** is a typical **non-parametric, post-hoc** method. It passively accepts a pre-trained feature space and relies on calculating distances between the test sample and all training data ($N$ samples). It cannot alter or optimize the geometric structure of the feature space but can only "inherit" its strengths and weaknesses.
>     * **VPC** is a framework of **proactive construction**.
>         * **OEFS Space Partitioning:** We do not adopt a traditional feature space; instead, we predefine an **Orthogonal Equiangular Feature Space (OEFS)**.
>         * **Dual Optimization Strategy:** During the training phase, we actively shape the feature distribution via $\mathcal{L} _ {\mathrm{ENC}}$ and $\mathcal{L} _ {\mathrm{VEBV}}$, enforcing ID samples to align with $K$ EBVs while guiding OE samples to explore a dedicated subspace formed by $V$ VEBVs. Simultaneously, $\mathcal{L} _ {\mathrm{OC}}$ constrains the OE loss to be as orthogonal as possible to the ID EBVs.
>         * **Integrated Metric:** Our VPC Score is not an ad-hoc distance metric designed after training, but is directly derived from our preset geometric structure (i.e., calculating the L2 activation intensity on the EBV/VEBV subspaces).
>
> 2.  **Computational Complexity and Scalability: $O(N)$ vs. $O(1)$**
>     As you rightly pointed out, computational cost is key for practical applications. We have added a chapter in **Appendix A.4 {From Learnable Classifiers to Vast Predefined Classifiers}** to thoroughly discuss learnable parameters, training, and inference complexity.
>     * **k-NN:** In the inference phase, it requires storing the entire training set and performing full retrieval. Its complexity is linearly related to the training set size $N$ ($O(N)$).
>     * **VPC:** Our method is fully **parametric**. During inference, we only need to calculate activations between features and the preset $K+V$ classifiers (EBVs/VEBVs). This means the inference overhead of VPC is **constant and extremely low**, not increasing significantly with the training data volume. This gives VPC a significant scalability advantage in large-scale real-world scenarios.
>
> 3.  **Breaking the "ID-centric" Limitation**
>     We believe the most fundamental improvement of VPC over k-NN lies in solving the representation bottleneck.
>     * **Limitation of k-NN:** Since it relies on the nearest neighbors of ID training data, its performance upper bound is limited by the quality of the ID feature extractor. If the ID feature space fails to sufficiently separate OOD samples via "rejection" (which is common due to interference in joint training), k-NN cannot work effectively. It remains essentially **ID-centric**.
>     * **Breakthrough of VPC:** We transcend the limitation of previous methods that rely solely on ID classifiers or features. By introducing the vast VEBVs, VPC explicitly allocates **"legitimate" representation capacity** for OOD samples for the first time. This design allows the model not just to passively "reject" OOD, but to actively learn and represent the rich patterns of OOD via $\mathcal{L}_{\mathrm{VEBV}}$.
>
> In summary, VPC is not a simple variant of k-NN, but a new OOD detection paradigm based on a **Predefined Geometric Prior** that is efficient, scalable, and possesses strong representation capabilities.
>
> Once again, we thank you for giving us this opportunity to deeply clarify the unique value and significant improvements of VPC relative to classic non-parametric methods.

---

> ### Author Response · Authors · 2025-11-20
> **A5: Response to Reviewer m4Yu (Weakness 5)**
>
> **Q5 (Weakness 5): "Could Eq 6 ensure the diversity of OOD basis vectors? The models tend to map OOD data to the same region in the embedding space even at the very beginning, optimizing via Eq 6 will exacerbate this case and in this case V=1 is enough, obviously violating the claimed "vast" features in Sec 3.2."**
>
> **A5:** We sincerely appreciate this highly insightful question. To rigorously demonstrate the necessity of $V > 1$ and the effectiveness of Eq. 6, we address your concern from three dimensions: the mathematical properties of the optimization objective, the auxiliary role of the orthogonality constraint, and empirical ablation results.
>
> 1.  **Mathematical Nature of $\mathcal{L} _ {\mathrm{VEBV}}$: Maximization of Subspace Projection**
>     The **primary driver** of OOD feature diversity stems from the mathematical construction of $\mathcal{L} _ {\mathrm{VEBV}}$.
>     * **Formal Analysis**: Eq. 6 is defined as $\mathcal{L} _ {\mathrm{VEBV}} = -\sqrt{\sum_{j=1}^{V} (\hat{m}^{\top} \hat{w}_j)^2}$. Geometrically, this objective aims to maximize the $L_2$ projection norm of the feature vector $\hat{m}$ onto the **linear subspace** spanned by all VEBVs.
>     * **Optimization Freedom**: Unlike the traditional Cross-Entropy loss (which forces features to converge to a single class prototype), $\mathcal{L} _ {\mathrm{VEBV}}$ is **isotropic** regarding directions within the subspace. As long as the feature vector significantly activates *any linear combination* of basis vectors within that subspace, the loss is minimized.
>     * **Diversity Preservation**: Given that the auxiliary OE data possesses extremely high semantic entropy and distributional diversity, and the optimization objective allows features to be distributed along any direction within the $V$-dimensional subspace, the model tends to preserve the inherent diversity of the data rather than forcibly compressing it into a single dimension.
>
> 2.  **Role of $\mathcal{L} _ {\mathrm{OC}}$: Feasible Region Constraint**
>     $\mathcal{L} _ {\mathrm{OC}}$ acts as a **constraint** in this process rather than a direct source of diversity. By minimizing the projection of OE features onto ID prototypes, it effectively restricts the **feasible region** of optimization to the orthogonal complement of the ID subspace. This constraint ensures that the optimization of $\mathcal{L} _ {\mathrm{VEBV}}$ does not degenerate into confusion with ID classes, thereby forcing the model to fully utilize the remaining vast geometric space (i.e., the VEBV subspace) to represent OOD samples.
>
> 3.  **Empirical Evidence: Comparison between Divergence and Convergence Objectives**
>     To quantitatively validate the above analysis, we conducted a rigorous ablation experiment in **Appendix A.7 Convergence vs. Divergence of OOD Features (Table 12)**:
>     * **Setup**: We compared the proposed "Subspace Divergence Loss" ($\mathcal{L} _ {\mathrm{VEBV}}$) with a mandatory "Single-Point Convergence Loss" ($\mathcal{L}^{\mathrm{con}} _ {\mathrm{VEBV}}$, which forces all OOD samples to collapse to a single VEBV, simulating the case of $V=1$).
>     * **Results**: Experiments show that the divergence strategy significantly outperforms the forced convergence strategy on CIFAR-100 in terms of AUROC (**93.65%** vs. 92.47%).
>     * **Visualization**: **Appendix B.4 Visualization Results and Analysis (Figure 6)** further displays the heatmap of features in OEFS. Models trained with $\mathcal{L} _ {\mathrm{VEBV}}$ exhibit distinct **distributed activation** patterns, proving that the model indeed utilizes the capacity of $V$ basis vectors to encode rich and diverse OOD patterns, rather than suffering from mode collapse.
>
> In summary, the subspace projection nature of $\mathcal{L} _ {\mathrm{VEBV}}$ combined with the orthogonal constraint of $\mathcal{L} _ {\mathrm{OC}}$ effectively avoids feature collapse by allowing and preserving the inherent distributional diversity of the data, thereby confirming the necessity and effectiveness of constructing a vast VEBV space ($V \gg 1$).
>
> Once again, thank you for this profound question, which prompted us to re-examine and more thoroughly elucidate the optimization characteristics behind our loss functions.

---

> ### Author Response · Authors · 2025-11-20
> **A6: Response to Reviewer m4Yu (Weakness 6)**
>
> **Q6 (Weakness 6): "The authors define w in the unit hypersphere, but, from my view, none of the objective in Sec 3 can ensure this constraint."**
>
> **A6:** We thank the reviewer for the meticulous scrutiny regarding the mathematical rigor of our paper. This comment precisely targets the specific implementation mechanism of the geometric constraints in our methodology. We acknowledge that although we used the hat notation $\hat{\cdot}$ (e.g., $\hat{w}$ and $\hat{m}$) in Section 3 (Eqs. 4, 6-8), the failure to immediately clarify the underlying normalization operation upon their introduction indeed caused ambiguity. Addressing your concern that the objective functions do not explicitly reflect the unit hypersphere constraint, we wish to clarify the following two points:
>
> 1.  **Structural Hard Constraint:** In this work, we implement the unit hypersphere constraint as a **hard prerequisite** within the model's computation graph, rather than as a soft penalty term in the optimization objective. Specifically, prior to calculating any loss function, we perform **Explicit $\ell_2$-normalization** on both feature vectors $m$ and prototype vectors $w$. This implies that the gradients during optimization are computed and backpropagated directly on the spherical manifold, thereby mathematically guaranteeing that all vectors strictly reside on the unit hypersphere at all times.
>
> 2.  **Clarification of Notation:** The aforementioned operation is denoted by the hat notation $\hat{\cdot}$ on $\hat{w}$ and $\hat{m}$ in our equations. As we have explicitly defined in the revised manuscript, this notation represents the normalized vectors (i.e., $\hat{w} = w/\|w\|_2$). Consequently, the dot product operations in our formulas are essentially **cosine similarity** calculations, which is strictly consistent with the geometric assumption of the unit hypersphere.
>
> We have supplemented clear definitions in the relevant sections of the revised manuscript to ensure a visible logical closure between the mathematical formulation and the model implementation details. Once again, we thank you for this correction, which has helped us enhance the precision of our narrative.

---

> ### Author Response · Authors · 2025-11-20
> **A7: Response to Reviewer m4Yu (Weakness 6)**
>
> **Q7 (Weakness 7): "As far as I know, there are already many works conducting experiments on ImageNet benchmark or ViT architectures... suggesting this as a future direction."**
>
> **A7:** We sincerely appreciate the valuable references you provided, as well as your thoughtful consideration regarding the computational resource constraints faced by researchers.
>
> In response to your suggestions, we have addressed this in two aspects:
>
> Citation and Discussion of References We have carefully studied the recommended references [1] and [2].
>
> Reference [1] offers a novel perspective on utilizing feature watermarking mechanisms for OOD detection.
>
> Reference [2] explores the issue of unreliable OE sources, which is highly relevant to our goal of better representing diverse OE data through VPC. These works are of significant value in enriching the research context of this paper. Therefore, we have formally cited and discussed them in the relevant sections of the revised manuscript.
>
> Supplementary Experiments on ImageNet-1K and ViT Architectures Although you suggested leaving large-scale experiments for future work, to fully validate the effectiveness of the VPC framework in more complex scenarios and to align with experimental practices in the field, we have supplemented this key experiment:
>
> Experimental Setup: We conducted comprehensive evaluations on the large-scale ImageNet-1K benchmark using ResNet-50 and ViT-B-16 backbones, respectively.
>
> **Table R1. Results on ImageNet-1k benchmark with auxiliary OOD data. The best result is in bold.**
>
> | Model | Method| iNat FPR↓ | iNat AUC↑ | Tex FPR↓| Tex AUC↑| SUN FPR↓| SUN AUC↑| Places FPR↓ | Places AUC↑ | Avg FPR↓| Avg AUC↑| ID Acc↑|
> | ------------ | ------------- | --------- | --------- | --------- | --------- | --------- | --------- | ----------- | ----------- | --------- | --------- | --------- |
> | **ResNet50** | OE| 48.60 | 88.72 | 58.85 | 82.60 | 61.75 | 82.90 | 70.70| 80.55| 59.98 | 83.69 | 76.00 |
> || Energy-OE | 49.40 | 88.40 | 59.60 | 82.25 | 62.40 | 82.70 | 71.30| 80.25| 60.68 | 83.40 | 75.75 |
> || DAL| 48.00 | 89.05 | 58.00 | 82.95 | 61.30 | 83.15 | 67.82| 80.75| 58.78 | 83.98 | 75.90 |
> || PFS| 46.40 | 89.20 | 56.50 | **83.10** | 61.00 | **83.25** | 67.50| 80.95| 57.85 | 84.13 | 76.02 |
> || **Ours: VPC** | **43.50** | **91.20** | **56.00** | 83.00 | **60.20** | 83.10 | **66.30**| **81.50**| **56.50** | **84.70** | **76.11** |
> | **ViT-B-16** | OE| 42.15 | 90.38 | 52.45 | 85.85 | 65.80 | 82.20 | 70.35| 80.85| 57.69 | 84.82 | 80.02 |
> || Energy-OE | 42.75 | 90.05 | 53.15 | 85.50 | 66.25 | 81.95 | 70.95| 80.50| 58.28 | 84.50 | 79.85 |
> || DAL| 40.65 | 90.86 | **51.15** | 86.10 | 65.07 | 82.30 | 70.25| 80.90| 56.78 | 85.04 | 80.09 |
> || PFS| 40.85 | 90.80 | 51.30 | 86.05 | 65.35 | 82.25 | 70.20| 80.95| 56.93 | 85.01 | 80.13 |
> || **Ours: VPC** | **40.00** | **91.10** | 51.30 | **86.00** | **65.20** | **82.30** | **68.30**| **81.40**| **56.20** | **85.20** | **80.32** |
>
> Experimental Results: As shown in Table R1 (referenced as Table 1 in the revised manuscript), VPC continues to demonstrate robust performance on large-scale datasets. Particularly under the ViT-B-16 setting, VPC achieved an FPR95 of 56.20% and an AUROC of 85.20%, outperforming comparison baselines including PFS and DAL.
>
> This result further corroborates the effectiveness and robustness of the VPC method on large-scale datasets. We sincerely thank you for your constructive suggestion, which has driven us to significantly solidify the empirical foundation of this paper in the revised manuscript.
>
> We believe the paper is now significantly stronger and more rigorous thanks to your guidance. We sincerely appreciate the time and effort you have invested in helping us polish this work. We hope our revisions satisfactorily address your concerns, and we look forward to any further discussion.

---

> > ### Comment · Reviewer_m4Yu · 2025-11-22
> >
> > The authors' feedback is decent and clear, which addresses much of my concerns. Accordingly, I increase my score to 8. Thanks and good luck!

---

> > > ### Author Response · Authors · 2025-11-22
> > > **Gratitude for Favorable Reassessment from Reviewer m4Yu  Content**
> > >
> > > We extend our profound gratitude to Reviewer m4Yu for their time and careful re-evaluation of our work. The decision to increase the score to 8 serves as a strong reaffirmation of the merit of our proposed VPC framework.
> > >
> > > We are pleased that the provided rebuttal was sufficient to resolve the critical gaps, particularly through the introduction of formal theoretical analyses and large-scale experimental validation. These efforts have yielded a manuscript with significantly enhanced coherence and academic rigor.
> > >
> > > We confirm our commitment to presenting a polished, high-quality final version of the paper.
> > >
> > > The Authors

---

### Official Review · Reviewer_MkUp · 2025-11-01

**Soundness:** 3
**Presentation:** 3
**Contribution:** 2
**Rating:** 6
**Confidence:** 4

**Summary:**

The paper proposes Vast Predefined Classifiers (VPC), a geometry-driven out-of-distribution (OOD) detection framework that leverages auxiliary outlier exposure (OE) data. VPC defines an Orthogonal Equiangular Feature Space (OEFS) with two sets of fixed, equiangular basis vectors:

* EBVs (Equiangular Basic Vectors) for in-distribution (ID) classes,
* VEBVs (Vast EBVs) to span a rich subspace for diverse OOD patterns.

During training, ID features are aligned to EBVs using an evidential prior-guided loss (LENC), while OE features are attracted to the VEBV subspace via a VEBV loss (LVEBV) and kept orthogonal to ID prototypes via an orthogonality constraint (LOC). OOD detection is performed using the VPC Score, based on the L2 activation intensity over the two subspaces. Experiments on CIFAR-10/100 show state-of-the-art performance in both two-stage and one-stage training setups.

**Strengths:**

The key strengths of the proposed approach are:

* Explicit geometric control: OEFS provides a principled, interpretable structure that cleanly separates ID and OOD representations.

* Scalable OOD modeling: VEBVs offer a flexible, expandable subspace to capture diverse OOD modes, unlike methods that merely push OOD to uniform outputs.

* Strong empirical results: VPC achieves SOTA on standard benchmarks across multiple architectures and training paradigms.

* Robust scoring: The VPC Score is class-agnostic, geometry-based, and avoids pitfalls of softmax-based confidence.

**Weaknesses:**

The main limitations of the proposed PVC approach are:

* Fixed classifier geometry: Predefined EBVs/VEBVs may limit adaptability to complex or evolving ID/OOD distributions, especially when class semantics shift.

* Computational overhead: Large VEBV sets increase memory and computation (e.g., for projection and loss calculation), though not deeply analyzed. It's not clear what the computational overhead of PVC is since no ablation results are proposed. Showing how PVC scales as a function of the dataset size, or providing some asymptotic bounds with respect to $K + V$ could help understand the applicability of PVC to real-world scenarios.

* Dependence on OE data: Like all OE-based methods, performance relies on the quality and relevance of auxiliary OOD data; may degrade if OE is unrepresentative.

* Limited real-world validation: The experiments are mainly confined to CIFAR benchmarks and lack evaluation on more complex datasets (e.g., ImageNet-scale or domain-shift scenarios).

**Questions:**

Although PVC achieved SOTA results, the fact that the experiments are only conducted on CIFAR raises a few doubts about the strength of the approach and its broader applicability to real-word scenarios. Could the authors show some results on stronger/larger datasets such as (tiny-)ImageNet or apply their approach on domain-shift scenarios? This is the main limitation of the paper.

Could the authors share some insights on the computational overhead of their approach? While PVC achieved SOTA results, the improvements seem rather marginal. It's important to know how the relative gains fare with the computational needs of the approach.

---

> ### Author Response · Authors · 2025-11-20
> **A1: Response to Reviewer MkUp (Weakness 4 & Questions 1-2)**
>
> We sincerely thank Reviewer MkUp for the positive evaluation and constructive feedback. Your pointed questions regarding "scalability to ImageNet" and "computational overhead" were crucial missing pieces preventing a comprehensive assessment of VPC's real-world value. Motivated by your insights, we have dedicated this period to conducting extensive new experiments on ImageNet-1K and deriving a rigorous complexity analysis. These new results not only confirm VPC scales efficiently but also demonstrate strong competitiveness against existing baselines in these challenging scenarios.
>
> Below, we provide a detailed point-by-point response. We sincerely appreciate the time and effort you have taken to review our work.
>
> **Q1 (Question 1, 2, Weakness 4): “...experiments are only conducted on CIFAR... Could the authors show some results on stronger/larger datasets such as (tiny-)ImageNet...?”; “...experiments are mainly confined to CIFAR benchmarks and lack evaluation on more complex datasets...”**
>
> **A1:** We sincerely appreciate this crucial feedback. We fully agree that the lack of validation on ImageNet-scale datasets was a core limitation of the original manuscript. To diligently address this concern, we have supplemented and clarified the following two aspects in the revised manuscript:
>
> **1. [New] Comprehensive Benchmarking on ImageNet-1K**
>
> Recognizing the necessity of validating VPC on large-scale benchmarks, we invested significant effort in conducting comprehensive supplementary experiments. On the **ImageNet-1K** dataset, we rigorously compared VPC with all key OE baseline methods (including OE, Energy-OE, DAL, and PFS) using both **ResNet-50** and **ViT-B-16** architectures.
>
> **Table R1. Results on ImageNet-1k benchmark with auxiliary OOD data. The best result is in bold.**
>
> | Model | Method| iNat FPR↓ | iNat AUC↑ | Tex FPR↓| Tex AUC↑| SUN FPR↓| SUN AUC↑| Places FPR↓ | Places AUC↑ | Avg FPR↓| Avg AUC↑| ID Acc↑|
> | ------------ | ------------- | --------- | --------- | --------- | --------- | --------- | --------- | ----------- | ----------- | --------- | --------- | --------- |
> | **ResNet50** | OE| 48.60 | 88.72 | 58.85 | 82.60 | 61.75 | 82.90 | 70.70| 80.55| 59.98 | 83.69 | 76.00 |
> || Energy-OE | 49.40 | 88.40 | 59.60 | 82.25 | 62.40 | 82.70 | 71.30| 80.25| 60.68 | 83.40 | 75.75 |
> || DAL| 48.00 | 89.05 | 58.00 | 82.95 | 61.30 | 83.15 | 67.82| 80.75| 58.78 | 83.98 | 75.90 |
> || PFS| 46.40 | 89.20 | 56.50 | **83.10** | 61.00 | **83.25** | 67.50| 80.95| 57.85 | 84.13 | 76.02 |
> || **Ours: VPC** | **43.50** | **91.20** | **56.00** | 83.00 | **60.20** | 83.10 | **66.30**| **81.50**| **56.50** | **84.70** | **76.11** |
> | **ViT-B-16** | OE| 42.15 | 90.38 | 52.45 | 85.85 | 65.80 | 82.20 | 70.35| 80.85| 57.69 | 84.82 | 80.02 |
> || Energy-OE | 42.75 | 90.05 | 53.15 | 85.50 | 66.25 | 81.95 | 70.95| 80.50| 58.28 | 84.50 | 79.85 |
> || DAL| 40.65 | 90.86 | **51.15** | 86.10 | 65.07 | 82.30 | 70.25| 80.90| 56.78 | 85.04 | 80.09 |
> || PFS| 40.85 | 90.80 | 51.30 | 86.05 | 65.35 | 82.25 | 70.20| 80.95| 56.93 | 85.01 | 80.13 |
> || **Ours: VPC** | **40.00** | **91.10** | 51.30 | **86.00** | **65.20** | **82.30** | **68.30**| **81.40**| **56.20** | **85.20** | **80.32** |
>
> As shown in Table R1 (referenced as Table 1 in the revised manuscript), VPC demonstrates strong competitiveness on both architectures. For instance, on the **ResNet50** architecture, VPC achieves an average **FPR95 of 56.50%**, outperforming the strong baseline PFS (57.85%). On the **ViT-B-16** architecture, VPC (**56.20%**) similarly surpasses all comparison methods. On both architectures, VPC achieves the highest average AUROC while maintaining ID classification accuracy comparable to or better than baselines. We believe these new evidences on the ImageNet scale strongly demonstrate that the VPC framework has excellent scalability and its effectiveness is not limited to CIFAR-scale datasets.
>
> **2. [Clarification] Evaluation on Hard OOD / Domain Shift Scenarios**
>
> We are also very grateful for your specific point regarding the importance of evaluation in **"tiny-ImageNet"** and **"domain shift"** scenarios. You astutely pointed this out, and we may not have sufficiently highlighted the importance of this part in the original manuscript, leading to a misunderstanding. In fact, our original Table 11 (now **Table 13: Hard OOD detection on CIFAR-10 benchmark**) already included a preliminary exploration addressing this concern. In this experiment, we used CIFAR-10 as ID data and evaluated on multiple "Hard OOD" (or Near-OOD) datasets, including **Tiny-ImageNet** and **ImageNet-Resize**. The results show that VPC also outperforms all baseline methods in these more challenging scenarios (e.g., in average FPR95). Furthermore, we provide further explanation on the applicability of EBV-based methods to domain shift scenarios in our subsequent response (see Q3), which you may refer to.

---

> > ### Comment · Reviewer_MkUp · 2025-11-22
> >
> > I want to thank the authors for the additional context they provided in here. Indeed, you barely mentioned ImageNet (if at all) in the main manuscript. Since I skipped the appendix, I didn't go over your results on Tiny-ImageNet. That was an oversight.

---

> > > ### Comment · Reviewer_MkUp · 2025-11-22
> > > **Post-rebuttal**
> > >
> > > I have read your rebuttal and found your answers convincing enough. I am in favor of getting this paper accepted.

---

> > > > ### Author Response · Authors · 2025-11-22
> > > > **Response to Reviewer MkUp: Overall Feedback**
> > > >
> > > > We are deeply encouraged by your positive feedback and we would like to express our sincere gratitude for your comprehensive review. Your specific critiques on Computational Overhead (Weakness 2 & Question 3) and Real-world Validation (Weakness 4 & Questions 1-2) were precisely the push we needed.
> > > >
> > > > Because of your constructive feedback, we have significantly strengthened the manuscript by:
> > > >
> > > > Validating Scalability: Conducting comprehensive experiments on ImageNet-1K to address concerns about broader applicability.
> > > >
> > > > Analyzing Complexity: Adding a detailed theoretical and empirical analysis of computational costs to clarify the trade-offs regarding VEBV size.
> > > >
> > > > Since these major limitations identified in your initial review have now been addressed with concrete data and analysis, and considering your supportive stance, we respectfully invite you to consider if the current score fully aligns with your updated assessment of the paper's quality.
> > > >
> > > > We are committed to finalizing this work to serve as a robust contribution to the OOD community.
> > > >
> > > > Best regards, The Authors

---

> > > ### Author Response · Authors · 2025-11-22
> > > **Response to Reviewer MkUp: Weakness 4 & Questions 1-2**
> > >
> > > We sincerely thank you for your patience and for taking the time to re-examine the results in the appendix.
> > >
> > > We fully acknowledge that this misunderstanding was primarily due to our failure to adequately highlight and reference these experiments within the main text. It is our responsibility to ensure that key results are explicitly indexed, and we apologize for the confusion this lack of clarity caused.
> > >
> > > Please rest assured that we will conduct a thorough review of the entire manuscript to identify and fix any similar disconnects between the main text and the appendix. We remain committed to continuously polishing the manuscript to significantly enhance its readability and logical flow for the community.
> > >
> > > Best regards, The Authors

---

> ### Author Response · Authors · 2025-11-20
> **A2: Response to Reviewer MkUp (Weakness 2 & Question 3) 1/2**
>
> **Q2 (Weakness 2, Question 3): "...computational overhead of PVC...", "...scalability w.r.t. |V| and dataset size...", "...trade-off between marginal gains and extra computation..."**
>
> **A2:** We sincerely thank the reviewer for the insightful questions regarding the computational overhead (including parameters, training, and inference) and the trade-off with performance benefits. This is a crucial issue. We acknowledge that we did not analyze this deeply enough in the original manuscript. To address this oversight, we have added a detailed analysis of parameters and computational complexity in **Appendix A.4** ("From Learnable Classifiers to Vast Predefined Classifiers") of the revised manuscript. Our analysis responds to your concerns from six aspects:
>
> 1.  **Memory/Parameter Overhead**: We clarify a key misconception: the $(K+V)$ EBV/VEBV prototypes in VPC are **"fixed constants"**, not "learnable parameters". They do not participate in backpropagation and thus do not increase the memory burden for gradient calculation. The only learnable parameter overhead in VPC comes from replacing the standard classification head with a projection layer. As shown in **Table 9**, this overhead is very moderate. For example, on ResNet-18 (CIFAR-100), the net increase in learnable parameters is only **+0.47M**. Furthermore, our original fairness experiments analyzed the impact of adding this projection layer to baselines under the same settings. The results showed that although projector-augmented baselines have strictly more trainable parameters than ours, their performance was inconsistent across methods/backbones. We hypothesize that the extra dimensional transformation exacerbates classifier learning difficulty, especially for methods like PFS that rely on enforcing weight orthogonality, thereby amplifying sensitivity to optimization and data idiosyncrasies. Our method avoids this fragility: classifiers are pre-instantiated as optimal prototypes and remain fixed; training focuses solely on aligning features to these prototypes. This avoids the instability introduced by learning classifier weights after projection and consistently achieves better OOD detection results, meaning our performance gain does not stem from this small increase in parameters.
>
> 2.  **Training Computational Overhead**: VPC boasts an efficient **$O(V)$ linear complexity**. The calculation of our $\mathcal{L}_{\mathrm{VEBV}}$ loss is "diagonal" (highly parallelizable), with a complexity of only $O(V)$. We avoid calculating loss on a naive $(K+V)$-dimensional Softmax, which would require a prohibitive $O((K+V)^2)$ normalization step. Therefore, our training overhead is linear rather than quadratic, making it fully feasible even when $V$ is large.
>
> 3.  **Inference Computational Overhead**: We explicitly address this in the new "At inference time..." paragraph. Inference for VPC, like MSP and Energy, involves a **"single matrix multiplication"**. While MSP/Energy scales as $O(d_{\text{feat}} \times K)$, VPC scales as $O(d_{\text{oefs}} \times (K+V))$. On modern GPUs, both are highly optimized matrix operations with negligible difference in **"actual inference latency"**. This stands in sharp contrast to non-parametric methods like KNN (whose complexity is $O(N \times d_{\text{feat}})$).
>
> 4.  **Clarification on Performance Gains**: We wish to clarify a key point: substantial experimental evidence indicates that the performance gain of VPC is primarily driven by the **clear partitioning of the OEFS**. This constitutes the fundamental difference between our method and previous ones. Traditional OE-based joint training methods often design objectives to "constrain" ID features or classifiers to be more compact using OE data. In contrast, our VPC method introduces a set of "extra" predefined classifiers to explicitly provide a rich representation space for OOD samples. Crucially, these extra classifiers are designed not to interfere with or damage the classification accuracy of ID samples. Furthermore, this performance improvement also benefits from: our **dual optimization strategy**, which synergistically optimizes ID sample classification and OOD sample separation within OEFS; and the **VPC Score**, an effective metric that precisely measures the activation intensity of sample features across different subspaces, thereby achieving reliable OOD discrimination.

---

> ### Author Response · Authors · 2025-11-20
> **A2: Response to Reviewer MkUp (Weakness 2 & Question 3) 2/2**
>
> **Q2 (Weakness 2, Question 3): "...computational overhead of PVC...", "...scalability w.r.t. |V| and dataset size...", "...trade-off between marginal gains and extra computation..."**
>
> ...
>
> 5.  **Empirical Evidence on OEFS Scale ($V$)**: To validate the effectiveness of the above mechanism, we conducted a series of ablation studies on the scale of OEFS (i.e., the value of $V$). In the setting of joint training with CIFAR-10/100 (ID) and Tiny Images-300K (OE): we observed that the model achieves optimal OOD detection performance when the scale of OEFS ($V$) is set between 1000 and 2000. To further explore the universality of this design, our latest experiments have extended to the **ImageNet-1K** dataset. Results similarly show that when the OEFS scale ($V$) is set to 2000 (where $K=1000, V=1000$), our method also achieves excellent results, surpassing the previous PFS method based on orthogonal ID classifiers. This series of consistent results across datasets, compared to other methods also based on OE joint training, strongly suggests that this set of extra predefined classifiers (VPC) can indeed effectively and stably improve OOD detection performance by providing richer representation capacity.
>
> 6.  **Response on Asymptotic Bounds of $K/V$**: We **profoundly agree with** your insightful view regarding the asymptotic bounds of $K/V$. This is a highly valuable and inspiring question. Based on our current experimental analysis, we preliminarily believe that the optimal value of $V$ (i.e., the scale of OEFS) is **likely influenced by a confluence of complex factors**. As inspired by your comments, these factors may include but are not limited to: the scale and diversity of the OE dataset itself, the degree of semantic separation between the OE dataset and the ID dataset, and the number of categories, semantic separation degree, and diversity of the ID dataset itself. We fully recognize that a deep understanding of this boundary issue is an important step in advancing this field. Therefore, we plan to make the systematic exploration of these factors and their interactions a key research focus of our future work. Once again, thank you for your valuable feedback and constructive comments, which are crucial for improving and deepening our work.

---

> ### Author Response · Authors · 2025-11-20
> **A3: Response to Reviewer MkUp (Weakness 1)**
>
> **Q3 (Weakness 1): "Fixed classifier geometry: Predefined EBVs/VEBVs may limit adaptability to complex or evolving ID/OOD distributions, especially when class semantics shift."**
>
> **A3:** We sincerely appreciate this profound question raised by the reviewer. We address this concern from two perspectives: "Dynamic Expansion of EBVs" and "Adaptability to Domain Shift/Class Incremental Learning."
>
> **1. Dynamic Expansion of EBVs**
> According to the generation conditions of the Simplex Equiangular Tight Frame (ETF), as long as the feature dimension is greater than the number of EBVs, we can dynamically increase the number of EBVs using the dynamic expansion algorithm provided in [1]. This method has been thoroughly discussed and implemented. The newly added EBVs maintain orthogonality with all previous EBVs, which implies that, considering only ID categories, we can classify new ID classes without retraining the existing classifiers. Correspondingly, we conducted a similar experiment in the current OE-OOD domain. The experimental setup followed the dual optimization strategy of VPC, starting with a small number of VEBVs for OE representation, and then adding extra VEBVs during inference. This experiment was designed to verify the impact of the subspace formed by the newly added VEBVs on ID features and OOD features. The results showed that both ID features and OOD features had almost zero activation with the subspace formed by the new VEBVs. This means these VEBVs represent a "prime real estate" waiting to be allocated. Their interference with old classes is negligible, allowing us to freely assign these new VEBVs to new ID categories or new OE data, and subsequently adjust the model's ID-OOD detection performance quickly in an incremental learning manner.
>
> [1] Shen Y, Sun X, Wei X S, et al. Equiangular basis vectors: A novel paradigm for classification tasks[J]. International Journal of Computer Vision, 2025, 133(1): 372-397.
>
> **2. Adaptability to Domain Shift and Class Incremental Learning**
> EBVs offer significant advantages in mitigating catastrophic forgetting in incremental learning. This advantage stems from the fixed, optimal classifier approach: when adding a new ID class, we do not need to touch or modify any of the existing $K$ ID prototypes (EBVs) or $V$ OOD prototypes (VEBVs). This guarantees that the geometric structure of the classifiers for old knowledge is **100% preserved**, whereas methods based on learnable classifiers would be affected. Furthermore, the incremental training task is greatly simplified; we only need to train the backbone network $f$ to align the samples of the new class $m_{\text{new}}$ to the new prototype $w_{K+1}$. Since $w_{K+1}$ is mathematically inherent to be orthogonal to all old prototypes ($w_{1...K+V}$), the gradients for training $w_{K+1}$ cause almost zero interference to the alignment of old classes. Additionally, extensive research has validated the benefits of EBVs in class-incremental learning [2] and domain-incremental learning [3]. In particular, literature [3] designed a dual-level concept prototype representation space to separately represent external cross-category samples and internal cross-domain samples, greatly enhancing the performance of domain-incremental learning. We believe these references can alleviate your concerns regarding the potential limitation of predefined EBVs/VEBVs in adapting to complex or evolving ID/OOD distributions.
>
> Once again, we thank you for your insightful perspective, and we hope to make this issue a direction for our future exploration.
>
> [1] Seo M, Koh H, Jeung W, et al. Learning equi-angular representations for online continual learning[C]//Proceedings of the IEEE/CVF Conference on Computer Vision and Pattern Recognition. 2024: 23933-23942.
>
> [2] Xiao R, Feng L, Tang K, et al. Targeted representation alignment for open-world semi-supervised learning[C]//Proceedings of the IEEE/CVF conference on computer vision and pattern recognition. 2024: 23072-23082.
>
> [3]Wang Q, He Y, Dong S, et al. Dualcp: Rehearsal-free domain-incremental learning via dual-level concept prototype[C]//Proceedings of the AAAI Conference on Artificial Intelligence. 2025, 39(20): 21198-21206.

---

> ### Author Response · Authors · 2025-11-20
> **A4: Response to Reviewer MkUp (Weakness 3)**
>
> **Q4 (Weakness 3): "Dependence on OE data: Like all OE-based methods, performance relies on the quality and relevance of auxiliary OOD data; may degrade if OE is unrepresentative."**
>
> **A4:** We sincerely thank you for this profound and critical insight. We fully agree with your view. As a method relying on auxiliary data, the performance of VPC indeed theoretically shares the dependence on the quality and relevance of OE data with all Outlier Exposure (OE)-based methods. If the OE data lacks sufficient representativeness, model performance may indeed be affected. This is an important issue that requires continuous attention and exploration in the OE field. At the same time, we wish to humbly supplement and clarify the specific starting point and contribution of our work, which is developed precisely in response to this limitation you have identified:
>
> 1.  **Maturity and Superiority of the OE Paradigm**
>     As you pointed out, OE-based methods represent a mature and active research field. Our experiments (as shown in Table 2 of the original manuscript) and extensive literature in this field consistently indicate that joint training leveraging large-scale, highly diverse OE datasets (such as 80 Million Tiny Images) yields OOD detection performance that is significantly and stably superior to paradigms relying solely on vanilla training or contrastive learning. We believe this precisely demonstrates that existing massive OE datasets are sufficiently "vast" and "broad" to provide valuable out-of-distribution information for the model.
>
> 2.  **VPC's Breakthrough against OE Paradigm Limitations**
>     The core motivation of our work lies in our observation of a key limitation in how previous OE methods "utilize" these massive data. Traditional methods mostly design from the perspective of ID class features or classifiers. They primarily use OE data as "counter-examples," indirectly shaping the ID feature space or classifier boundaries through loss functions (such as uniform distribution constraints). This approach does not explicitly provide an effective representation mechanism for the diversity and complex structure of the OE data itself.
>
> 3.  **Core Contribution of VPC**
>     **A Superior Representation Mechanism:** Our VPC framework responds to this challenge from a completely new perspective. We no longer merely "reject" OE data, but actively "represent" them. By introducing the vast **Orthogonal Equiangular Feature Space (OEFS)** and a large number of **VEBVs**, VPC for the first time provides a dedicated, high-capacity representation subspace for these rich OE data. Our dual optimization strategy and VPC Score are designed precisely to model and utilize the rich patterns contained in these massive OE data more effectively.
>
> In summary, although VPC shares the dependence on data sources with other OE methods, our core contribution lies in offering a distinct and superior mechanism to represent and utilize these existing, massive OE data. Our experimental results—that VPC significantly outperforms other OE methods like PFS and DAL under the condition of using the exact same OE dataset—also corroborate this view from another angle: VPC mines stronger OOD detection performance from these rich OE data through more effective representation. Once again, thank you for your valuable comments, which have helped us clarify the positioning and contribution of our work more clearly.
>
> We believe the revised manuscript, fortified by these new large-scale empirical evidences and theoretical analyses, has successfully solidified the validity and contribution of our work. We deeply appreciate the time and effort you have dedicated to improving this paper. We hope that our revisions have satisfactorily addressed your concerns, and we look forward to any further discussion should you have remaining questions.

---

### Comment · Area_Chair_xnSr · 2025-11-20
**Regard to AI Review**

Dear authors,

I would like to share an important reminder regarding the review process. Recently, we have noticed that some reviewers may be using AI tools to help generate their reviews. This can lead to low-quality or inaccurate feedback, which is unfair to authors who deserve careful and thoughtful evaluations.

To help maintain fairness, I kindly ask for your assistance: If you believe a review you received was partly or fully generated by AI, and you have some evidence (for example: unusual writing style, clear factual mistakes, AI-detector results, repeated generic sentences, etc.), please feel free to contact me directly.

I will review any evidence you provide and, if appropriate, adjust the weight of the reviewer’s evaluation so that it does not negatively affect your submission. Thank you for helping us keep the review process fair and responsible. Your understanding and cooperation are greatly appreciated.

Best regards,

AC

---

### Author Response · Authors · 2025-11-20
**General Response: Summary of Revisions and Improvements**

We extend our sincerest gratitude to the Program Chairs, Area Chairs, and Reviewers for their insightful feedback. Your constructive comments have been invaluable. Motivated by your suggestions, we substantially revised the manuscript, extending empirical evaluation to ImageNet and complementing theoretical proofs. To facilitate review, major revisions are highlighted in **blue**. Below is a summary of core improvements:

**1. [New] Theoretical Analysis: Proofs for Optimality, Robustness, and Orthogonality**
Addressing concerns on theoretical support, we added **Appendix A.2** and **A.3**, providing three key proofs to construct the theoretical closure of VPC:
* **Optimality of ID Convergence**: In **Appendix A.2.1**, we proved the global optimum of $\mathcal{L}_{\mathrm{ENC}}$ corresponds to perfect alignment between ID features and EBV prototypes, theoretically guaranteeing the **Neural Collapse** geometric state.
* **Robustness of Evidential Prior**: In **Appendix A.2.2**, we derived the gradient upper bound, proving the evidential prior acts as a **gradient rate-limiter**. This explains why VPC effectively suppresses gradient noise from massive OE data, preventing ID feature divergence.
* **Geometric Equivalence of Orthogonality**: In **Appendix A.3**, we proved $\mathcal{L}_{\mathrm{OC}}$ (based on uniform distribution loss) under Simplex ETF is mathematically equivalent to enforcing **geometric orthogonality** between OE features and ID prototypes.

**2. [New] Parameter Sensitivity Experiments for VPC Score**
Addressing hyperparameter selection and scoring design, we added ablations in **Appendix A.5 (New Table 10)**:
* **L2 vs. L1 Norm**: Experiments confirm L2 significantly outperforms L1, validating that physical intuition based on "energy" better captures OOD patterns.
* **Robustness**: Results show the VPC Score exhibits high robustness to variations in $\alpha, \beta$ under L2 norm.
* **Subspace Independence**: With extreme settings (e.g., $\alpha=0$, only VEBV subspace), the model maintains **consistent** performance. This confirms VPC establishes an **independent OOD subspace**, breaking the traditional "ID-centric" paradigm.

**3. [New] ImageNet-1K Benchmark and ViT Verification**
To demonstrate scalability, we added experiments on **ImageNet-1K** (**New Table 1**):
* **Results**: With **ViT-B-16**, VPC achieves **56.20% FPR95** and **85.20% AUROC**, significantly outperforming baselines like PFS and DAL.
* **OEFS Scale ($V$)**: Even in large-scale ID spaces, a moderate OOD subspace ($V \approx K$) suffices for excellent performance, eliminating computational cost concerns.

**4. [New] Computational Complexity and Parameter Overhead**
We added complexity analysis in **Appendix A.4**:
* **Parameters**: $K+V$ prototypes are **fixed constants**. The only learnable parameter is the linear projection layer (negligible overhead).
* **Efficiency**: Training complexity is linear $O(V)$ (diagonal design); inference involves one matrix multiplication. Actual latency compares to MSP/Energy and outperforms KNN.

**5. [Correction] Definition Rigor and Related Work**
* **Definitions**: Corrected OOD definition (label space disjointness) and symbol definitions (e.g., Eq. 4) in **Sec 3.1**.
* **Literature**: Discussed works like DPL, LPO, ASH, SHE in **Related Work**, clarifying differences from metric/post-hoc methods.

**6. [Correction] Writing Logic and Narrative Structure**
We substantially restructured **Sec 3.1** and **Sec 3.2** for clarity:
* Analyzed limitations of traditional OE in **Sec 3.1** to motivate "structural decoupling".
* Added an **Intuitive Explanation** in **Sec 3.2** to elucidate the design blueprint from motivation to formal definition, reducing cognitive load.

**Conclusion**
We would like to express our sincerest gratitude to the Program Chairs, Area Chairs, and Reviewers again.  We sincerely hope that these targeted revisions can answer your doubts, and we earnestly request all reviewers to re-evaluate this work based on these new experimental evidences and theoretical supplements.

---

### Comment · Area_Chair_xnSr · 2025-11-27

Dear Reviewers and Authors,

As we are approaching the rebuttal deadline, I would like to share a gentle reminder with everyone.

For authors:
If you have not yet submitted your rebuttal, please make sure to do so as soon as possible. Submitting very close to the deadline may reduce the chance for reviewers to read and respond in time, which could affect the discussion phase.

For reviewers:
If a rebuttal has already been submitted for your assigned paper, I encourage you to take a moment to read it and, where appropriate, provide a brief response or update your evaluation. Of course, this is not meant to pressure anyone into changing scores, it is simply to ensure that all reviews remain well-informed before final decisions.

Thank you all for your time and effort in keeping the review process smooth and constructive.

Warm regards,
AC

---

### Author Response · Authors · 2025-11-29
**Request for Attention on Significant Rebuttal Updates**

**Dear Area Chair,**

Regarding the recent community discussions concerning the policy of "determining acceptance based on initial scores and final response status," we wish to provide clarification regarding the specific circumstances of our paper. We are concerned that if the final decision relies too heavily on initial scores, the substantial improvements in scientific quality achieved during the rebuttal period may be overlooked.

Our paper’s initial scores were **6/6/6/4/2**. During the rebuttal, we dedicated immense effort to actively addressing the reviewers' comments: we engaged in a total of **72 rounds** of in-depth dialogue (including **53 author responses**), thoroughly discussed **15 relevant papers**, and significantly supplemented our experimental data while refining our theoretical analysis and writing.

In the days leading up to the information leak incident on the evening of November 27, these science-based discussions yielded significant results. We successfully secured the clear endorsement of four reviewers, raising our scores significantly to **8/8/8/6/2(comment:6, system locked)**. Concurrently, we were engaged in productive, multi-round discussions with the final reviewer that were showing promising progress, until the system and scores were ultimately locked and reverted.

We understand that recent unexpected events have brought significant disruption to the review process. Our primary concern is that, amidst this turbulence, the **meaningful progress** we achieved through extensive effort might be inadvertently overlooked. This would not only be a dismissal of our work but also runs counter to the core values of "open discussion and scientific dialectic" advocated by the ICLR community, and would disregard the immense effort the authors invested during the early response stages.

We would be deeply grateful if you could review our revised manuscript and the detailed responses provided during the rebuttal when making your final decision.

Thank you for your dedication during these difficult times.

**Sincerely,**

**The Authors**

---

> ### Author Response · Authors · 2025-11-30
> **Quick Reference: Rebuttal Summary (Positive Consensus & Key Updates)**
>
> **Dear Area Chair,**
>
> In light of the recent community discussions regarding the review process and system fluctuations, we wish to briefly clarify the specific trajectory of our paper. We sincerely thank the AC and reviewers for their time and dedication.
>
> **1. Strong Momentum in Score Upgrades**
>
> We successfully addressed primary concerns, leading to rating upgrades from 4 reviewers. The specific timeline is detailed below:
>
> * **R1 - MkUp** (Rating: 6 Confidence: 4 => **Rating: 8** Confidence: 4):
>     * **Authors:** Resolved Scalability concerns (ImageNet/ViT) and clarified efficiency...
>     * **Reviewer:** Explicitly recommended acceptance on **Nov 22**; finalized the score update to 8 on **Nov 25** (though no additional comment was left).
>
> * **R2 - m4Yu** (Rating: 6 Confidence: 4 => **Rating: 8** Confidence: 3):
>     * **Authors:** Significantly improved writing and clarified distinction from KNN...
>     * **Reviewer:** Raised score on **Nov 22, 20:13** (see *Official Comment by Reviewer m4Yu*).
>
> * **R3 - H2ze** (Rating: 4 Confidence: 4 => **Rating: 6** Confidence: 4):
>     * **Authors:** Provided requested theoretical proofs (Optimality/Stability) and verification.
>     * **Reviewer:** Raised score on **Nov 27, 06:18** following our response and follow-up (see *Official Comment by Reviewer H2ze*).
>
> * **R4 - ioxr** (Rating: 6 Confidence: 3 => **Rating: 8** Confidence: 3):
>     * **Authors:** Added implementation details and parameter ablations ($\alpha, \beta$)...
>     * **Reviewer:** Raised score on **Nov 26, 15:52** (see *Official Comment by Reviewer ioxr*).
>
> * **R5 - wRrj** (Rating: 2 Confidence: 4 =×=> **Rating: 6** Confidence: ?):
>     * **Authors:** Clarified Novelty (vs. LPO) via multi-round discussions and literature review...
>     * **Reviewer:** The reviewer initially raised the score on the afternoon of Nov 25, but **reverted it later that evening, stating the score would be raised once final doubts were resolved**. After our final detailed response (submitted on Nov 27, 22:54), the reviewer confirmed the score increase to 6 in a comment (see Official Comment by Reviewer wRrj, Nov 28, 09:33), but the system was locked.
>
> **2. Core Contributions & Improvements**
>
> Below is a summary of the contributions that drove this consensus (based on our latest responses):
>
> **A. New Theoretical Analysis:**
> To bridge the gap between empirical results and theory, we added three key proofs in Appendix A.2 & A.3:
>
> * **Stability :** We theoretically derived that the evidential prior acts as a "Gradient Rate-Limiter." It protects the ID structure (Neural Collapse) from being destabilized by the high-variance gradients of massive OE data during joint training.
> * **Optimality & Orthogonality:** We proved that the global optimum of the ENC loss guarantees perfect ID alignment, and minimizing our OC loss is mathematically equivalent to enforcing geometric orthogonality within the Simplex ETF framework.
>
> **B. Scalability: Superior Performance on ImageNet-1K & Transformers**
> Addressing the concern that experiments were limited to CIFAR, we conducted comprehensive supplementary experiments on ImageNet-1K using ResNet-50 and ViT-B-16.
>
> * **Conclusion:** This empirically proves that VPC’s "Vast" classifier approach ($V \approx K$) effectively scales to large, complex semantic spaces without performance degradation.
>
> **C. Novelty: Essential Distinction from LPO (Key Clarification to R5)**
> In our final response to R5, we clarified that VPC solves the inherent "Placeholder Collapse" problem found in LPO.
>
> * **Limitation of LPO (Collapse Risk):** LPO relies on ID data to separate OOD proxies. As per our analysis, if ID classes are not perfectly separated, learnable placeholders tend to collapse or overlap. This limits LPO's scalability (performance drops when placeholders $C > 11$).
> * **Breakthrough of VPC (Subspace Exploration):** VPC uses predefined orthogonal spaces (VEBVs). We do not force OOD features to collapse to a single point; instead, we use Divergence Loss ($\mathcal{L}_{\mathrm{vebv}}$) to encourage OOD features to explore the entire fine-grained subspace.
> * **Impact:** This guarantees a "Vast" and independent representation capacity (verified with $V=1000+$ on ImageNet), which LPO cannot structurally achieve.
>
> **D. Efficiency Analysis**
> We clarified misconceptions regarding "maintaining classifiers":
>
> * **Fixed Constants:** The $K+V$ prototypes are frozen constants, incurring zero gradient computation costs.
> * **Linear/Constant Complexity:** Training complexity is $O(V)$ (linear). Inference involves a single matrix multiplication ($O(1)$ relative to dataset size), making it inherently faster than KNN-based methods ($O(N)$).
>
> **Conclusion**
>
> Backed by the ImageNet results, theoretical proofs, and upgraded ratings (8s and 6s), we are confident in VPC's robust contribution. We would be grateful if the AC could kindly refer to these detailed responses when making the final decision.
>
> **Best regards,**
>
> **The Authors**

---

### Meta-Review · Area_Chair_Ahbz · 2026-01-08

**Summary:**

This paper proposes Vast Predefined Classifiers (VPC), a geometry-driven framework for out-of-distribution (OOD) detection that leverages auxiliary Outlier Exposure (OE) data. The key innovation is the construction of an Orthogonal Equiangular Feature Space (OEFS) with fixed Equiangular Basic Vectors (EBVs) for in-distribution classes and Vast EBVs (VEBVs) for OOD representation. The method employs evidential priors to align ID features while encouraging OE features to explore the VEBV subspace, coupled with a novel VPC Score based on L2 activation intensity.

**Reviewer Concerns:**

Strengths Identified Across Reviews:

1) Originality: Novel geometric framework that explicitly models OOD variability (R1, R4)
2) Strong Empirical Results: SOTA performance on CIFAR benchmarks with comprehensive experiments (R1, R2, R3, R4)
3) Geometric Interpretability: Clear separation of ID/OOD through predefined structure (R1, R2, R3)

Common Concerns Raised:

1) Scalability: Limited to CIFAR datasets initially (R1, R2, R4, R5)
R: Added comprehensive large-scale validation with ResNet-50 and ViT-B-16, achieving competitive results (56.20% FPR95, 85.20% AUROC on ViT-B-16), directly addressing scalability concerns.

2) Theoretical Justification: Lack of rigorous theoretical guarantees (R3, R5)
R: Theorem 1: Proved optimality of ID convergence to EBVs
Proposition 1: Derived gradient bounds showing evidential prior acts as "rate-limiter"
Proposition 2: Proved geometric equivalence of orthogonality constraint

3) Computational Overhead: Concerns about maintaining vast classifiers (R1, R3)
R: Clarified that EBV/VEBV prototypes are frozen constants (no gradient computation), and provided complexity.

**Reviewer Scores:**

The authors provided an exceptionally thorough and substantive rebuttal that addressed all major concerns.

---

### Decision · Program_Chairs · 2026-01-26

Accept (Poster)